# The regulatory landscape of the human HPF1- and ARH3-dependent ADP-ribosylome

Ivo A. Hendriks [1,4], Sara C. Buch-Larsen [1,4], Evgeniia Prokhorova [2], Jonas D. Elsborg [1], Alexandra K.L.F.S. Rebak[1], Kang Zhu[2], Dragana Ahel[2], Claudia Lukas [3], Ivan Ahel [2] & Michael L. Nielsen [1✉]

Despite the involvement of Poly(ADP-ribose) polymerase-1 (PARP1) in many important biological pathways, the target residues of PARP1-mediated ADP-ribosylation remain ambiguous. To explicate the ADP-ribosylation regulome, we analyze human cells depleted for key regulators of PARP1 activity, histone PARylation factor 1 (HPF1) and ADP-ribosylhydrolase 3 (ARH3). Using quantitative proteomics, we characterize 1,596 ADP-ribosylation sites, displaying up to 1000-fold regulation across the investigated knockout cells. We find that HPF1 and ARH3 inversely and homogenously regulate the serine ADP-ribosylome on a proteome-wide scale with consistent adherence to lysine-serine-motifs, suggesting that targeting is independent of HPF1 and ARH3. Notably, we do not detect an HPF1-dependent target residue switch from serine to glutamate/aspartate under the investigated conditions. Our data support the notion that serine ADP-ribosylation mainly exists as mono-ADP-ribosylation in cells, and reveal a remarkable degree of histone co-modification with serine ADP-ribosylation and other post-translational modifications.

[1] Proteomics Program, Novo Nordisk Foundation Center for Protein Research, Faculty of Health and Medical Sciences, University of Copenhagen, Blegdamsvej 3B, 2200 Copenhagen, Denmark. [2] Sir William Dunn School of Pathology, University of Oxford, Oxford OX1 3RE, UK. [3] Protein Signaling Program, Novo Nordisk Foundation Center for Protein Research, Faculty of Health and Medical Sciences, University of Copenhagen, Blegdamsvej 3B, 2200 Copenhagen, Denmark. [4] These authors contributed equally: Ivo A. Hendriks, Sara C. Buch-Larsen. ✉email: michael.lund.nielsen@cpr.ku.dk

ADP-ribosylation (ADPr) is catalyzed by poly-ADPr-polymerases (PARPs), also known as ADP-ribosyltransferases (ARTs). ADPr refers to the process where an ADP-ribose moiety is transferred from NAD+ to the amino acid side-chains of target proteins[1]. Although discovered more than 50 years ago[2], neither the biological functions of PARP1-catalyzed ADPr nor its contributions to cell biology are fully understood. ADPr imposes structural constraints and introduces negative charges to acceptor proteins, thereby altering protein localization, protein function, and interactions with other proteins and DNA[3]. Due to this, PARP1 has emerged as a master regulator in the DNA damage response through its crux position in the base excision repair pathway[1]. Given the importance of ADPr in the DNA repair process, inhibitors of catalytic PARP activity (i.e., PARP inhibitors) are widely used in the clinic to combat various cancers. Hence, additional insight into the enzymatic catalysis of ADPr advances our understanding of PARP biology, and refines current knowledge related to the mechanisms surrounding sensitivity and resistance to PARP inhibitors in cancer[4,5].

Despite the biomedical importance of PARP1-mediated ADPr, and inhibition thereof, the molecular details surrounding which amino acid residues are PARP1 targets has remained an analytical conundrum[6,7]. PARP1-catalyzed ADPr has historically been reported to target the side-chains of glutamate and aspartate, both as PARP1 auto-ADPr[8,9] and on downstream protein substrates[10–12]. Recently, serine residues were reported as the major target for PARP1-mediated ADPr in human cells during DNA damage[13–16], under physiological conditions[17], and in the context of PARP1 auto-ADPr[14,18]. While the mechanistic details surrounding the catalytic preferences of PARP1 remain perplexing, observations that PARP1/2 activity is regulated by Histone PARylation factor (HPF1)[19] have provided some biochemical and structural understanding to this ambiguity[20,21]. In vitro experiments support that availability of HPF1 changes the catalytic preference of PARP1 from glutamate and aspartate to serine residues[18,22,23]. However, an unbiased and in vivo investigation of the HPF1-dependent ADP-ribosylome, using MS-based proteomics, is currently lacking.

The catalytic activity of HPF1-PARP1/2 is counteracted by poly-ADP-ribose glycohydrolase (PARG)[24], and ADP-ribosylhydrolase 3 (ARH3)[25,26], providing an additional layer of complexity to the cellular ADPr dynamics. This leads to the high turnover of poly-ADP-ribosylation (PAR) and certain PARP1-catalyzed modifications existing as mono-ADPr (MAR) events in human cells[26,27]. With PARP1 described as a poly-ADPr (PAR) polymerase[1,28], the notion that PARP1 substrates primarily exist as MARylated entails fundamental biological insights and epitomizes the relevance of exploring the modularity of ADPr under in vivo conditions. However, at present, only a limited number of in vivo substrates have been described as MARylated. Hence, comprehensive and unbiased analyses are required to assess whether MARylation is a global phenomenon, considering that PARP1 is able to target hundreds of protein substrates[16].

To alleviate these knowledge gaps, while concomitantly providing a valuable resource to the community on the enzymatic modularity of the ADP-ribosylome, we employed an Af1521-based proteomic approach[29] to compare ADPr acceptor sites and protein substrates across human cells genetically depleted for HPF1 or ARH3[19,26]. Our results support that HPF1 and ARH3 are global regulators of serine ADPr, and unexpectedly we did not observe increased ADPr on glutamate and aspartate residues in the absence of HPF1, under the in vivo conditions we investigated. Collectively, our data summarizes the HPF1- and ARH3-mediated ADPr regulation at a proteome-wide scale, and provides

further insights into the cellular distribution of PARP1-catalyzed ADPr target residues.

## Results

**ETD analysis can detect ADPr on all residue types.** Over the last years, we have developed and optimized an Af1521-based mass spectrometry (MS) methodology, which relies on the non-ergodic fragmentation propensity of electron-transfer dissociation (ETD)[30] to allow for faithful localization of ADPr to the correct amino acid residue[14,16,17]. We previously found that supplemental activation via electron-transfer higher-collisional dissociation (EThcD) improves the ability to detect ADPr sites while not compromising localization accuracy[16]. Under physiological conditions and in response to DNA damage, we have observed ADPr on nine reactive amino acid residue types; C, D, E, H, K, R, S, T, Y, with the vast majority residing on serine residues across the cellular systems we have investigated. These observations stand in stark contrast to historical observations in the field, where predominant glutamate and aspartate ADPr have been observed[8,10,31].

We set out to ascertain the ability of EThcD to detect ADPr modification of any amino acid residue in an unbiased manner. As an initial control, we investigated the ability of HCD, ETD, and EThcD to detect ADPr-reactive amino acids in unmodified peptides. We performed triplicate analyses on HeLa total lysates and observed that the amino acid distribution of identified peptides was slightly different when using ETD fragmentation, with a preferential detection of basic residues (Supplementary Fig. 1). However, these differences were mitigated when using EThcD fragmentation, resulting in essentially the same identified amino acid distribution as when using HCD fragmentation. Thus, we conclude that EThcD fragmentation, which we use for our ADPr experiments, is not biased against the detection of any type of amino acid residue.

Next, we set up a series of in vitro ADP-ribosylation experiments, as in vitro experiments are more easily controlled compared to in vivo, and observations of E/D modification have canonically been derived from in vitro experiments[8]. We used ETD-based high-resolution MS to analyze tryptic digests of reactions with PARP1, PARP10, or PARP14. PARP1 reactions were optionally performed in the presence of H1.0, or both H1.0 and HPF1 (Fig. 1a and Supplementary Data 1), whereas PARP10 and PARP14 reactions were performed without substrates or co-factors.

Overall, we observed all nine reactive amino acids to be modified by ADPr across all in vitro experiments (Fig. 1b). Specifically, we observed mainly arginine residues to be auto-modified on PARP10, whereas chiefly lysine, histidine, and tyrosine residues were auto-modified in the case of PARP14. Aspartate ADPr was detected in low amounts on both PARP10 and PARP14. Notably, we did not observe serine ADPr auto-modification on PARP10 or PARP14 under the investigated in vitro conditions.

Auto-modification of PARP1 in the absence of HPF1 resulted in ADPr exclusively targeting glutamate residues, although we were unable to detect modification of H1.0 in this setting. With the addition of HPF1, we found PARP1 and H1.0 to be exclusively modified on serine residues (Fig. 1b). In vitro modification was consistently detected on S-204, S-499, S-507, S-519 on PARP1, and S-7, S-104, S-131, and S-171 on H1.0, in agreement with previous in vivo observations[14,27,32]. Strikingly, we noted the degree of serine modification to be ~50-fold more abundant in the presence of HPF1 as compared to glutamate modification in the absence of HPF1 (Fig. 1b). PARG treatment of samples is routinely applied to reduce PAR to MAR, as only

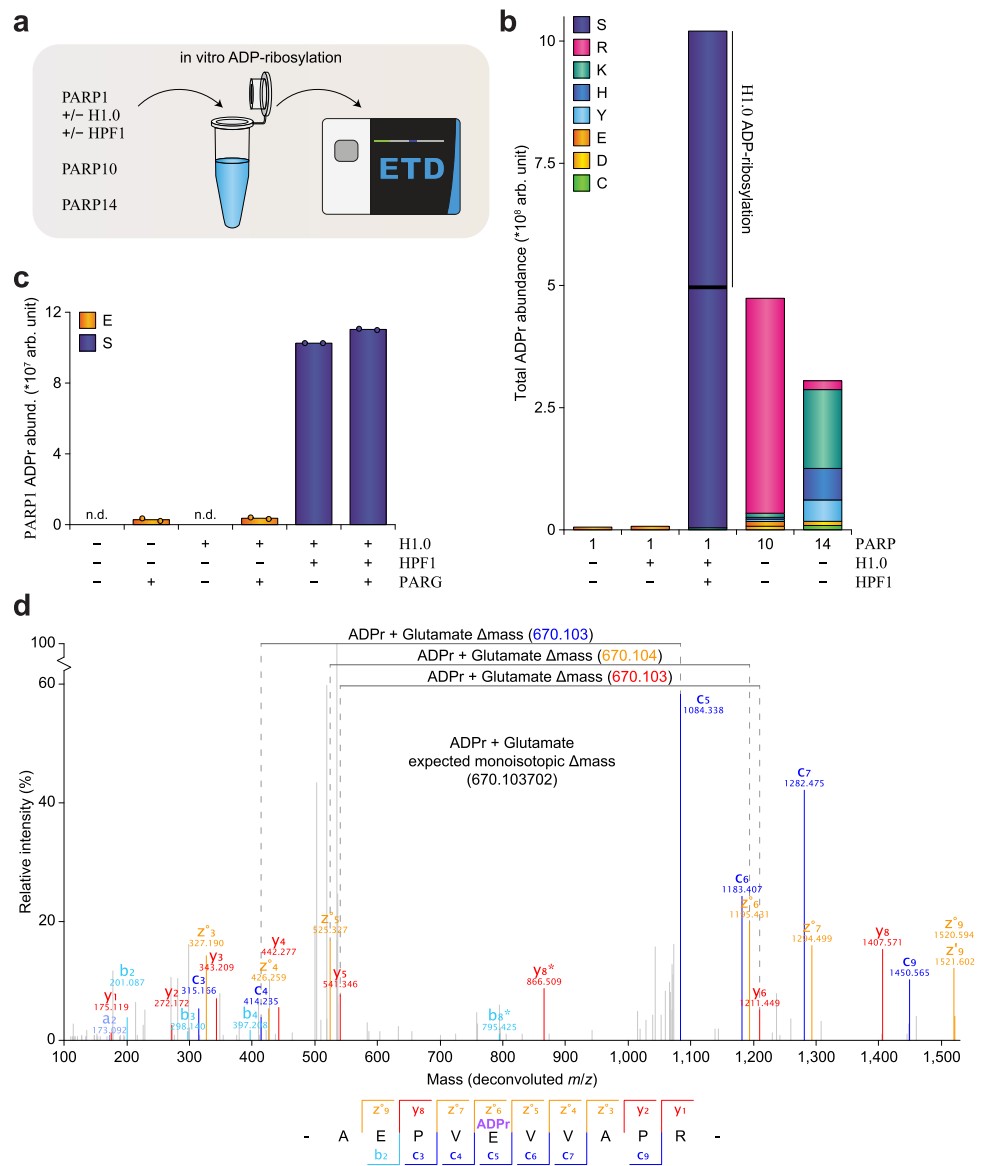

**Fig. 1 EThcD analysis of in vitro ADP-ribosylated PARPs. a** Overview of the experimental design. Recombinant PARP1, PARP10, or PARP14, were modified in vitro. For PARP1, the reaction was performed ±H1.0 and ±HPF1. Samples were trypsinized, and afterward mock-treated or treated with PARG, prior to analysis of the peptides by ETD-based MS. **b** Total abundance distribution of in vitro ADPr-modified amino acid residues. Abundance values ± PARG were summed. **c** Average abundance distribution of in vitro auto-modified residues on PARP1. $n = 3$ technical replicates. "n.d." not detected, data points not shown in case of n.d. **d** Fully annotated EThcD MS/MS spectrum, demonstrating confident localization of ADP-ribosylation to Glutamate-491 on PARP1. Light blue; b-ions, dark blue; c-ions, red; y-ions, orange; z-ions, gray; unassigned. Source data are provided as a Source Data file.

MAR modification of peptides is readily detectable by MS. Intriguingly, treatment of HPF1-mediated in vitro reactions with PARG did not notably change the amount of ADPr modification (Fig. 1c). Conversely, without HPF1 mediation of the in vitro reaction, PARG activity was necessary for detection of glutamate ADPr, suggesting that this modification occurs primarily as poly-ADPr, but overall in lower abundance compared to serine ADPr. Still, the most abundant glutamate ADPr site we observed on auto-modified PARP1 was E491 (Fig. 1d), which is consistent with the most predominant in vivo glutamate ADPr sites detected in previous proteomics screens[10,16,31].

Taken together, we observed vastly different profiles of ADPr-modified residue types across distinct in vitro reactions, supporting the absence of detection bias. Thus, we demonstrate that ETD-based MS is readily able to detect linkage of ADPr to all of the nine reactive amino acids.

**Mono-ADP-ribosylation is predominant in human cells**. With the Af1521 macrodomain able to bind both mono- and poly-ADP-ribose[33,34], we reasoned that moderations to our proteomics enrichment strategy would allow for assessing the global extent of MAR in human cells (Fig. 2a)[14]. While some macrodomains are able to hydrolyze ADP-ribose moieties[35], we previously demonstrated that the Af1521 macrodomain does not exert hydrolase activity[36]. Nonetheless, we wanted to ensure that our MS-based proteomics strategy using Af1521 enrichment preserves all ADPr and is unbiased with regard to the identification of amino acid acceptor sites. To this end, we compared Af1521 to two independent antibodies (from Cell Signaling Technology (CST)) raised against poly/mono-ADP-ribose, as antibodies should not possess hydrolase activity and thus allow unbiased enrichment of ADP-ribosylation for proteomics analysis (Fig. 2a). PARG is integral to our established protocol[16] but might exhibit off-target

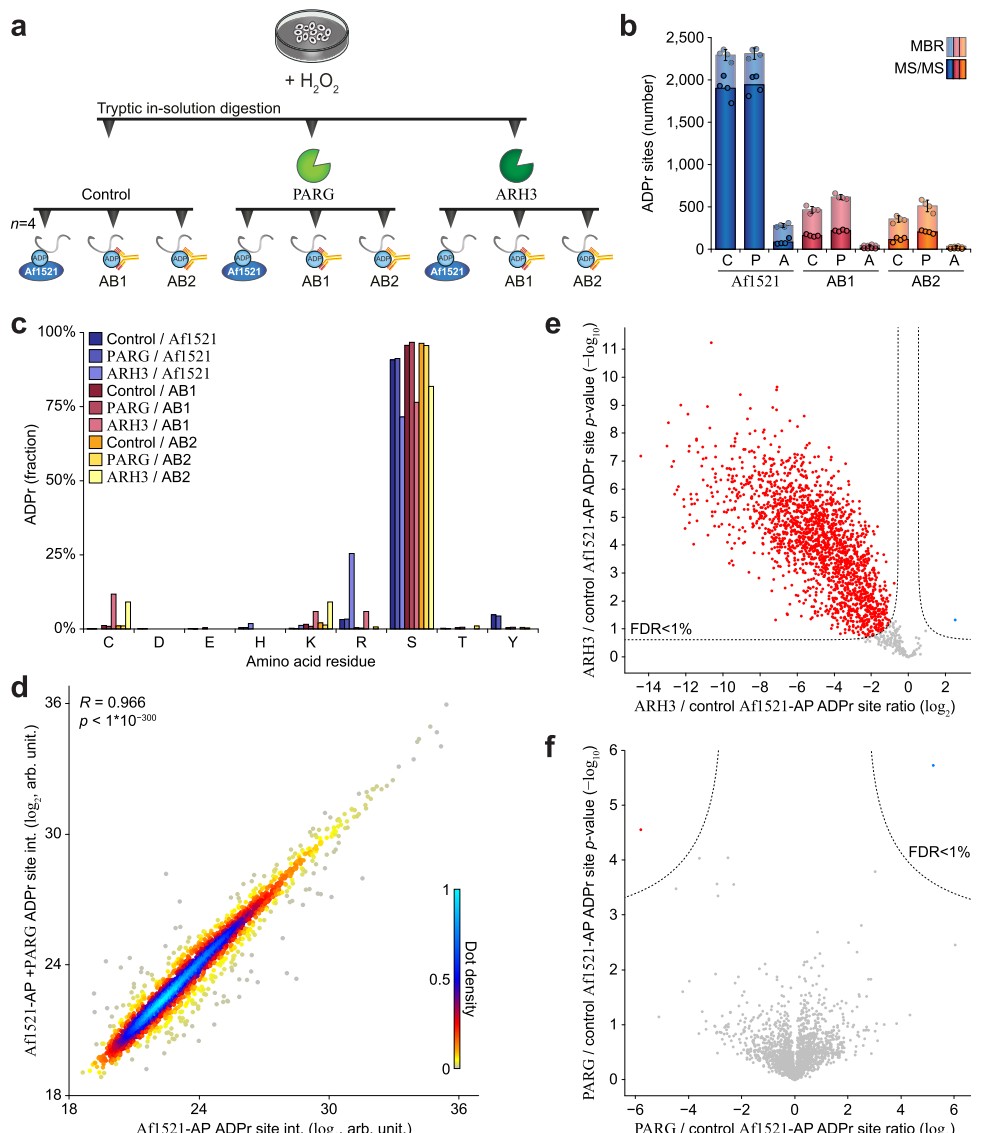

**Fig. 2 Evaluation of different erasers and enrichers for the purification of ADPr. a** Overview of the experimental design. HeLa cells were cultured, $H_2O_2$-treated at 1 mM for 10 min, lysed, and either mock treated, PARG-treated, or ARH3-treated. Quadruplicate ($n = 4$) ADPr peptide purifications were performed using either Af1521 macrodomain, E6F6A antibody (AB1), or D9P7Z antibody (AB2). ADPr peptides were analyzed using ETD-based high-resolution MS. **b** Overview of the number of ADPr sites identified and localized (>90% probability). Data are presented as mean values ± SD, $n = 4$ purification replicates. "C"; control, "P"; PARG, "A"; ARH3, "MBR"; matching between runs. **c** Visualization of the abundance fraction of ADPr as distributed across different amino acid residue types. **d** Scatter plot analysis demonstrating the correlation between mock-treated and PARG-treated ADPr sites. "R" indicates Pearson correlation, $p$-value determined via linear regression $t$ test. **e** Volcano plot analysis comparing ARH3-treated versus mock-treated Af1521-enriched ADPr sites. Red and blue dots indicate significantly down- and upregulated sites, respectively. Significance was determined via two-tailed Student's $t$ testing, with permutation-based FDR-control applied with an $s0$ fuzz factor of 0.5 and 2500 rounds of randomization, to ensure a corrected $p$-value of <1%. **f** As **e**, but comparing PARG-treated versus mock-treated Af1521-enriched ADPr sites. Source data are provided as a Source Data file.

activity. To control for this, we evaluated purification of ADPr without the addition of PARG, or with the addition of ARH3, which should remove all serine ADPr[25] (Fig. 2a). We used our contemporary method to purify ADPr from quadruplicate HeLa cell cultures exposed to oxidative stress and analyzed the ADPr peptides by MS.

In total, we identified and localized 2758 ADPr sites (Supplementary Data 2), of which the majority was identified via the Af1521 purification (Fig. 2b and Supplementary Fig. 2A). Both antibodies allowed direct identification of ~200 ADPr sites, and up to ~500 by matching MS1-level evidence from Af1521 runs. With all three purification methods, ADPr was primarily detected on serine residues (Fig. 2c), supporting that the Af1521

methodology is unbiased. In line with expectations, in vitro ARH3 treatment of peptides prior to ADPr purification greatly reduced the number of identified ADPr sites (Fig. 2b), with non-serine ADPr becoming relatively more abundant after in vitro ARH3 treatment (Fig. 2c). Strikingly, omitting in vitro PARG treatment did not notably alter the number of ADPr sites detected (Fig. 2B), and we observed a strong correlation between ADPr site abundances from untreated and PARG-treated samples (Fig. 2d and Supplementary Fig. 2B). Furthermore, whereas we observed a significant decline in the majority of all ADPr sites upon in vitro ARH3 treatment (Fig. 2e), this was not the case after PARG treatment (Fig. 2f), indicating that in vitro PARG treatment was redundant for reduction of ADPr polymer length. Notably, PARG

inhibition of HeLa cells resulted in accumulation of ADPr in both the absence and presence of oxidative stress, indicating that PARG actively reverses PAR under the experimental conditions we investigated (Supplementary Fig. 2C). Collectively, we demonstrate that our Af1521 methodology is unbiased, and our data indicate that mono-ADPr is a global phenomenon in cells.

**ADPr-modified proteins are functionally similar across multiple proteomics strategies**. We established that EThcD fragmentation is able to detect ADPr on any residue type (Fig. 1), and our antibody-based experiments strengthen our observations using Af1521-based enrichment of ADPr-modified peptides (Fig. 2). Nonetheless, the relative lack of glutamate and aspartate ADPr in our datasets remains puzzling. The contemporary proteomics strategy for detection of glutamate and aspartate ADPr relies on the enrichment of ADPr-modified peptides using boronate affinity chromatography, followed by elution of peptides using hydroxylamine (HA). This strategy entirely removes the ADPr moiety from modified peptides and instead induces a hydroxamic acid (+15.0109 Da) derivative of glutamate and aspartate residues, to pinpoint which residues originally harbored the ADPr moiety[10].

Recently, an investigation of TMT data—where HA is used as part of the sample preparation workflow—revealed the second most common unexpected mass shift to be +15.0109 Da[37]. This mass shift could be explained via the conversion of carboxylic acid to hydroxamic acid, resulting in the addition of an NH group on glutamate and aspartate residues[37]. Consequently, we wondered whether the usage of HA during ADPr sample preparation similarly could result in a derivatization artifact of glutamate and aspartate residues towards hydroxamic acid, and to which extent this would occur. To this end, we prepared a technical standard using commercial bovine serum albumin (BSA), a routinely used quality control protein for mass spectrometric analysis, and a protein highly unlikely to be substantially ADP-ribosylated. We subjected BSA to an ADPr sample processing workflow essentially as described previously[38], while including or excluding the HA treatment step (Supplementary Fig. 3A). The resulting BSA peptides were analyzed using an Orbitrap Exploris mass spectrometer (MS) using higher-energy collisional dissociation (HCD) fragmentation.

Overall, in the absence of HA treatment, we did not observe any hydroxamic acid modification on BSA (Supplementary Fig. 3B–E and Supplementary Data 3). Strikingly, when BSA was treated with HA, we observed dozens of hydroxamic acid modifications on aspartate and glutamate residues, overall accounting for 8% of all identified spectra and 4% of total BSA abundance. Notably, when re-processing proteomics data derived from analyses of human breast cancer cell lines, wherein ADPr was enriched using a combination of boronate affinity and HA elution[31], we observed ~20% of the total signal corresponding to peptides modified by hydroxamic acid (Supplementary Fig. 3C, E and Supplementary Data 4). Concomitantly, we also observed a large induction of asparagine and glutamine oxidation after HA treatment of BSA (Supplementary Fig. 3F–I), which constitutes a known chemical reaction of primary amines by HA, referred to as transamidation[39], which is otherwise rarely observed in proteomics samples. This aberrant chemical modification was not observed for untreated BSA, although it was observed to a modest degree in the breast cancer data (Supplementary Fig. 3G, I). Methionine oxidation, which is a common side effect during electrospray ionization (ESI)[40], was unaltered across all investigated data, showing an absence of technical bias. Our sequence coverage of BSA was ~50%, and we found

that nearly half of all profiled aspartate and glutamate residues were modified by hydroxamic acid when BSA was treated with HA (Supplementary Fig. 3J).

Notwithstanding the potential for HA to induce hydroxamic acid modification on unmodified glutamate and aspartate residues, we endeavored to elucidate whether glutamate and aspartate ADPr could be more efficiently purified using boronate affinity chromatography[10]. However, HA derivatization prohibits detection of the intact ADPr moiety and prevents detection of ADPr on residue types other than glutamate and aspartate. Thus, we decided to implement the boronate affinity-based enrichment of ADPr-modified peptides but instead opted for acid elution of the peptides[41], thereby retaining the intact ADPr moiety on the analyzed peptides, while avoiding any potential chemical artifacts (Fig. 3a). The experiment was performed using quadruplicate cultures of HeLa cells exposed to oxidative stress and analyzed using ETD-based MS. In terms of peptide purity, enrichment using boronate was comparable to "AB2", and slightly less efficient compared to "AB1", with Af1521-based purification proving optimal at ~93% peptide purity (Fig. 3b). Although we observed some glutamate ADPr when using boronate-affinity purification (Fig. 3c and Supplementary Data 5), we observed serine residues as the main ADPr target, consistent with Af1521- and antibody-based strategies.

ADPr target proteins identified in this experiment were associated with terms including chromosome and DNA repair (Fig. 3d), and moreover, we observed a highly significant overlap with ADPr target proteins identified in several MS-based proteomics studies based on distinct techniques and from various labs[10,14,16,29,31,36,42]. The large majority of ADPr target proteins identified here using boronate affinity were previously identified using either Af1521-based purification with intact ADPr readout or boronate affinity-based purification with hydroxamic acid readout (Fig. 3e).

Collectively, we present three distinct biochemical methods validating the predominant nature of serine ADPr in the cellular conditions we investigated (Figs. 2 and 3). Importantly, most ADPr proteomics studies identify the same target proteins, regardless of highly distinct methods and varying observations at the site-specific level.

**HPF1 and ARH3 globally regulate a homogenous ADP-ribosylome**. The important and contrasting roles of HPF1 and ARH3 in serine ADPr homeostasis were previously elucidated[18,19,21,25,26]. However, insight into the systemic effect of these enzymes on global and site-specific ADP-ribosylation is currently lacking. To this end, we cultured either wild-type (control) U2OS cells or U2OS lacking HPF1 or ARH3 in quadruplicate. To capture the dependency of HPF1 and ARH3 across various biological processes, we analyzed cells both under oxidative stress known to induce ADP-ribosylation[43], and untreated steady-state conditions (Fig. 4a). We validated the absence of HPF1 or ARH3 via immunoblot analysis (Fig. 4b) and confirmed strong induction of ADPr in response to H₂O₂ treatment and ARH3 depletion (Fig. 4c). Subsequently, we purified ADPr sites from all replicate cultures using our Af1521 methodology and analyzed them via MS. Overall, we identified 1596 ADPr sites (Supplementary Data 6), corresponding to 799 ADPr target proteins (Supplementary Data 7). On average, the largest numbers of ADPr sites were observed in ARH3 KO cells (Fig. 4d), even in the absence of H₂O₂ treatment), which closely resembled immunoblot observations (Fig. 4c). In terms of ADPr abundance, HPF1 KO nearly abolished detectable ADPr, whereas ARH3 KO resulted in dramatic ADPr accumulation, with ~100- and

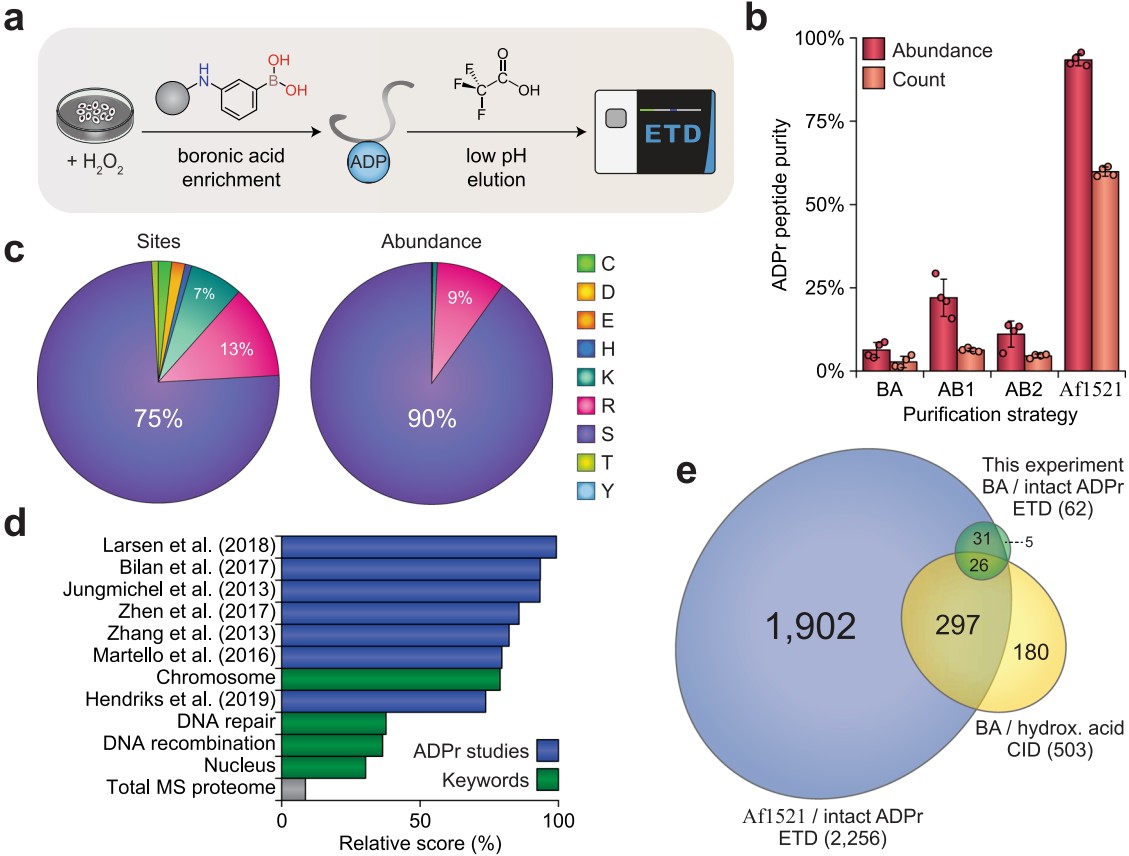

**Fig. 3 Integration of boronic acid enrichment with ETD-based MS. a** Overview of the experimental design. HeLa cells were cultured in quadruplicate ($n = 4$), mock-treated or $H_2O_2$-treated at 1 mM for 10 min, lysed, and trypsinized. ADPr-modified peptides were enriched using boronate-affinity beads, eluted using 0.15% trifluoroacetic acid to maintain the intact ADPr moiety, and analyzed using ETD-based high-resolution MS. **b** Comparison of all ADPr enrichment methods employed in this study, visualizing peptide purity based on either abundance or spectral counting. Data are presented as mean values ± SD, $n = 4$ cell culture replicates. "BA"; boronate affinity, "AB1"; E6F6A antibody, "AB2"; D9P7Z antibody. **c** Pie-chart analysis showing the distribution of ADPr sites across different amino acid residue types, either based on the number of sites or abundance. **d** Term enrichment analysis visualizing similarity of ADPr target proteins identified in this experiment to several other ADPr proteomics studies[10,14,16,29,31,36,42], to several cellular functions associated with ADPr, or to a deep total proteome study[80]. The relative score is based on the multiplication of logarithms derived from the enrichment ratio and the $q$-value. Terms were significant with $q < 0.02$, as determined through Fisher Exact Testing with Benjamini–Hochberg correction. **e** Scaled Venn diagram depicting the overlap between ADPr target proteins identified using either; boronate affinity enrichment and detection of intact ADPr using ETD-based fragmentation (this experiment), boronate affinity enrichment and detection of hydroxamic acid derivatives using collision-induced dissociation (CID) fragmentation[10,31], or Af1521 enrichment and detection of intact ADPr using ETD-based fragmentation[14,16]. CID fragmentation is comparable to HCD fragmentation. "BA"; boronate affinity, "hydrox. acid"; hydroxamic acid derivatives. Source data are provided as a Source Data file.

~1000-fold increases compared to control and HPF1 KO cells, respectively (Fig. 4e). We observed a high degree of replicate reproducibility (Fig. 4f, g), and a strong tendency for untreated control cells to resemble HPF1 KO cells, with $H_2O_2$-treated control cells resembling ARH3 KO cells. Indeed, $H_2O_2$ induction of ADPr was predominantly observed in wild-type cells (Fig. 4d–f, h, i), whereas the KO cell lines responded markedly less to $H_2O_2$ treatment. A considerable number of ADPr sites induced in response to $H_2O_2$ treatment in control cells were detectable in untreated ARH3 KO cells (Fig. 4f), and the majority of control ADPr sites were also detectible in the ARH3 KO cells (Fig. 4h, i). Homogeneity between untreated ARH3 KO cells and $H_2O_2$-treated control cells was supported by Pearson correlation analysis of identified ADPr modification sites (Supplementary Fig. 4A), and term enrichment analysis on ADPr target proteins detected under both conditions revealed many canonical features associated with ADP-ribosylation (Supplementary Fig. 4B), such as nuclear and chromosomal localization and modification of proteins involved in RNA metabolism and DNA repair. Overall, we find that ARH3 KO results in the accumulation of vast

amounts of ADPr, while remaining functionally homogenous with the majority of ADPr usually observed in response to oxidative stress.

**ARH3 and HPF1 primarily modulate serine ADPr**. We have previously demonstrated that serine residues are the primary target of ADPr in cultured cells[14], and the predominance of lysine-directed serine ADPr in the form of the lysine–serine (KS) motif[13,16]. We scrutinized the prevalence of these phenomena in the context of HPF1 KO and ARH3 KO and found that serine ADPr was overall predominant (Fig. 5a), with an increase to >99% abundance in ARH3 KO cells or in response to $H_2O_2$ in control cells, while even in HPF1 KO cells serine ADPr narrowly stayed in the lead. The large majority of serine ADPr resided within KS motifs (Fig. 5b), even in the absence of HPF1, suggesting that the preferential targeting of ADPr to the KS motif does not rely solely on HPF1. In terms of amino acid distribution, the number of serine ADPr sites was by far the greatest (Fig. 5c), corresponding to abundance-based observations (Fig. 5a). Notably, in the absence of HPF1, where the total ADPr signal is greatly

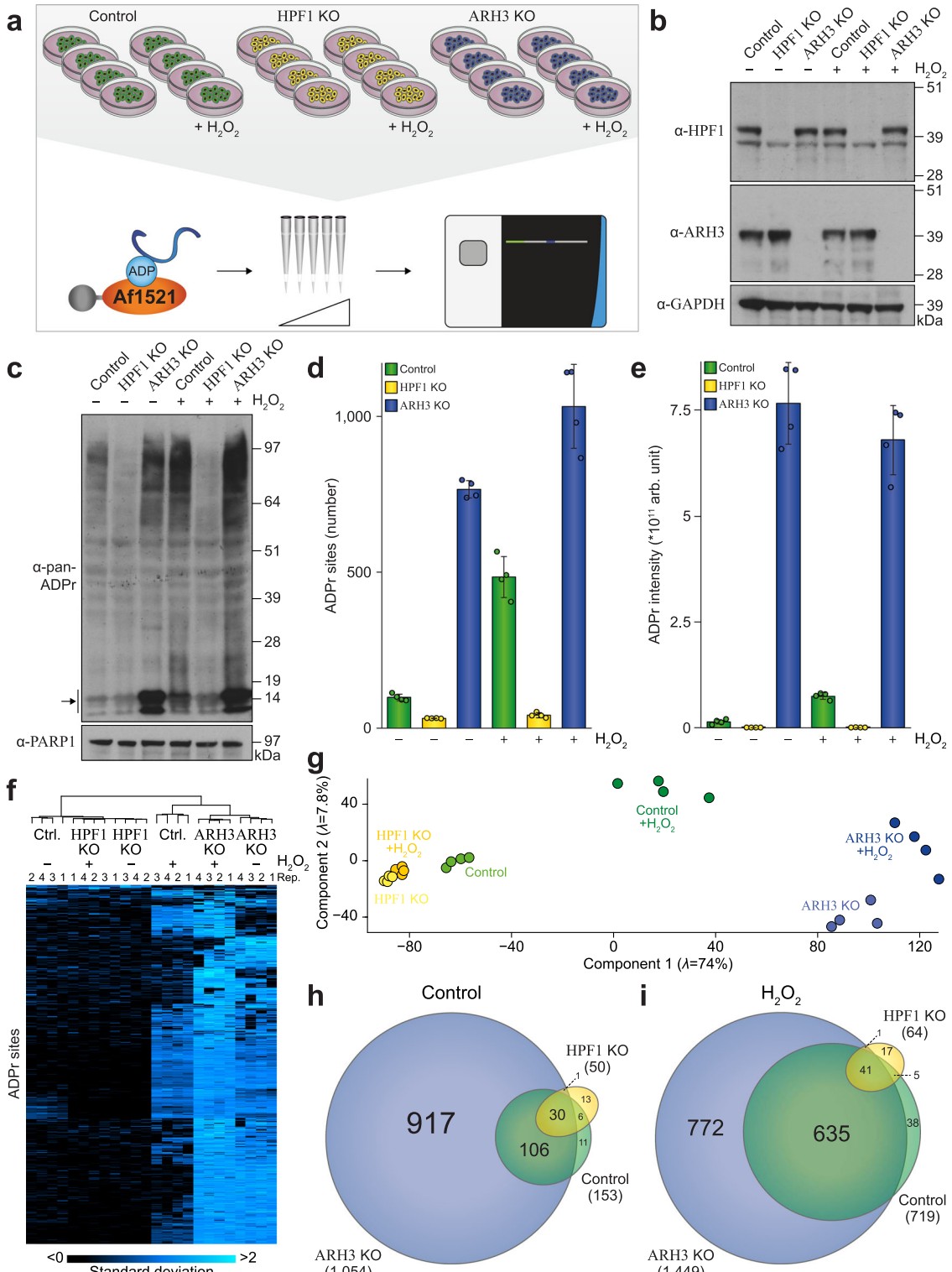

**Fig. 4 The ADP-ribosylome in HPF1 and ARH3 knockout cells. a** Overview of the experimental design. U2OS cells, either wild-type (control), HPF1 knockout (KO), or ARH3 KO, were cultured in quadruplicate ($n = 4$), and either mock- or H₂O₂-treated at 1 mM for 10 min. ADPr sites were enriched using the Af1521 methodology, fractionated, and analyzed using high-resolution MS. $n = 4$ cell culture replicates. **b** Immunoblot analysis validating the knockout of HPF1 and ARH3. This experiment was performed as two independent biological replicates, with similar results. The same set of samples was loaded on multiple membranes as technical replicates, to allow readout using different antibodies. Ponceau-S loading controls for each membrane are available in the Source Data. **c** As **b**, but highlighting the ADPr equilibrium in control, HPF1 KO, and ARH3 KO cells, ± H₂O₂ treatment at 1 mM for 10 min. The arrow indicates histone ADPr. **d** Overview of the number of identified and localized ADPr sites. $n = 4$ cell culture replicates, data are presented as mean values ± SD. **e** As **d**, showing ADPr abundance. **f** Hierarchical clustering analysis of z-scored ADPr site abundances, visualizing the relative presence of ADPr sites across the experimental conditions. **g** Principle component analysis indicating the highest degree of variance between sample conditions. **h** Scaled Venn diagram depicting the overlap between ADPr sites in untreated cells. **i** As **h**, but for H₂O₂-treated cells. Source data are provided as a Source Data file.

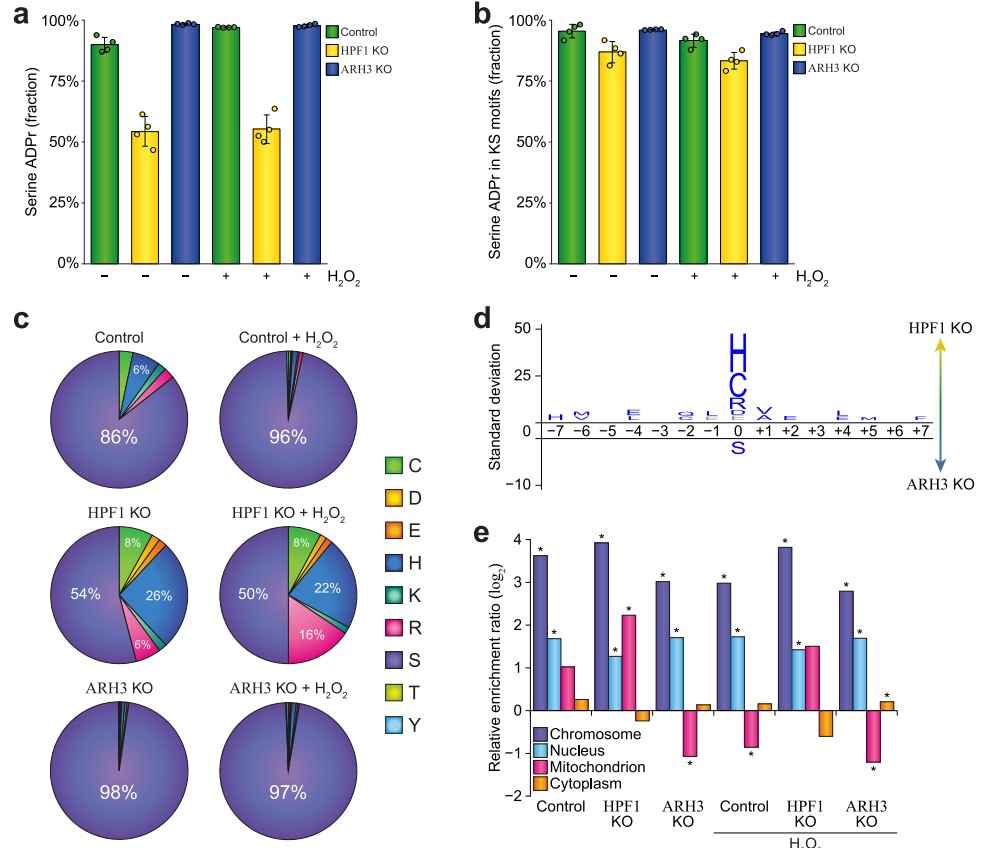

**Fig. 5 Site-specific properties of ADPr in HPF1 and ARH3 knockout cells. a** Overview of the abundance fraction of ADPr modifying serine residues. $n = 4$ cell culture replicates, data are presented as mean values ± SD. **b** As **a**, but visualizing the fraction of serine ADPr in (lysine–serine) KS motifs. **c** Pie-chart analysis showing the distribution of ADPr sites across different amino acid residue types. **d** IceLogo analysis visualizes relative preference for ADPr to be targeted to different amino acid residue types. Amino acid residues displayed above the line were enriched for HPF1 KO, and those displayed below were enriched for ARH3 KO. Displayed amino acids were determined to be significantly changed at $p < 0.05$ as determined by two-tailed Student's $t$ testing using iceLogo software[77], $n = 50$ and $n = 1054$ HPF1 KO and ARH3 KO ADP-ribosylation sites, respectively. $p$-Values for all amino acids were re-assessed using via two-tailed Fisher Exact testing with Benjamini–Hochberg correction, and are listed in the Source Data. Gray: not significantly different in the second statistical test. **e** Term enrichment analysis visualizing the Gene Ontology subcellular localization of ADPr target proteins across experimental conditions. $n = 99$ (control), $n = 35$ (HPF1 KO), $n = 575$ (ARH3 KO), $n = 429$ (control + H$_2$O$_2$), $n = 45$ (HPF1 KO + H$_2$O$_2$), $n = 756$ (ARH3 KO + H$_2$O$_2$) ADPr target proteins, tested versus a background of $n = 21,297$ proteins, with Gene Ontology subcellular localization annotated for $n = 402$ chromosomal, $n = 5298$ nuclear, $n = 1166$ mitochondrial, and $n = 5027$ cytoplasmic proteins. Significance was determined via two-tailed Fisher Exact testing with Benjamini–Hochberg correction for multiple hypotheses testing, *$p < 0.05$. Individual $p$-values are listed in the Source Data. Source data are provided as a Source Data file.

reduced (Fig. 4e), the remaining ADPr was observed to also target histidine (~24%), arginine (~11%), and cysteine (~8%) residues, with only weak evidence supporting trace ADPr targeting aspartic and glutamic acid residues (Fig. 5c). Directly comparing the sequence context of ADPr between HPF1 and ARH3 KO cells revealed no significant changes in amino acid distribution directly surrounding the ADPr sites (Fig. 5d), and highlighted a significant shift in modification away from serine residues and primarily toward histidine, cysteine, and arginine residues. Notably, this shift in preference away from serine was relative, as the absolute number of non-serine ADPr sites did not significantly change (Supplementary Data 6).

In terms of subcellular localization of proteins modified by ADPr, a strong enrichment was observed for chromosomal and nuclear localization across all experimental conditions (Fig. 5e). We did not note a particular preference for cytoplasmic ADPr target proteins; however, in the absence of HPF1 and in untreated control cells, we noted a relative preference for ADP-ribosylation of mitochondrial proteins (Fig. 5e), corroborating our previous observations of histidine ADPr targeting the mitochondria[16].

Taken together, the absence of HPF1 resulted only in reduction of serine ADPr, rendering non-serine ADPr relatively but not absolutely more prominent.

**Delayed dissociation of auto-ADPr-deficient PARP1 from DNA damage sites.** PARP1 is mainly modified on S-499, S-507, and S-519, residing in the auto-modification domain[13,14,18]. Here, we observed that the abundance of PARP1 auto-ADPr (Fig. 6a) followed the global trend for total ADPr with regard to ARH3 KO and H$_2$O$_2$ treatment (Fig. 4e). Under all conditions, including HPF1 KO, we noted predominant targeting of ADPr to the three primary serine residues (Fig. 6b).

To investigate the biological relevance of ADP-ribosylation of these three serine residues, we generated stable cell lines expressing either wild-type GFP-PARP1, or a mutant GFP-PARP1 in which these serine residues were substituted with alanine residues ("3SA mutant"). Notably, the cell lines were generated in PARP1 knockout U2OS cells[19], to eliminate interference from endogenous PARP1. In the case of the 3SA mutant, oxidative stress-induced PARP1 auto-modification was

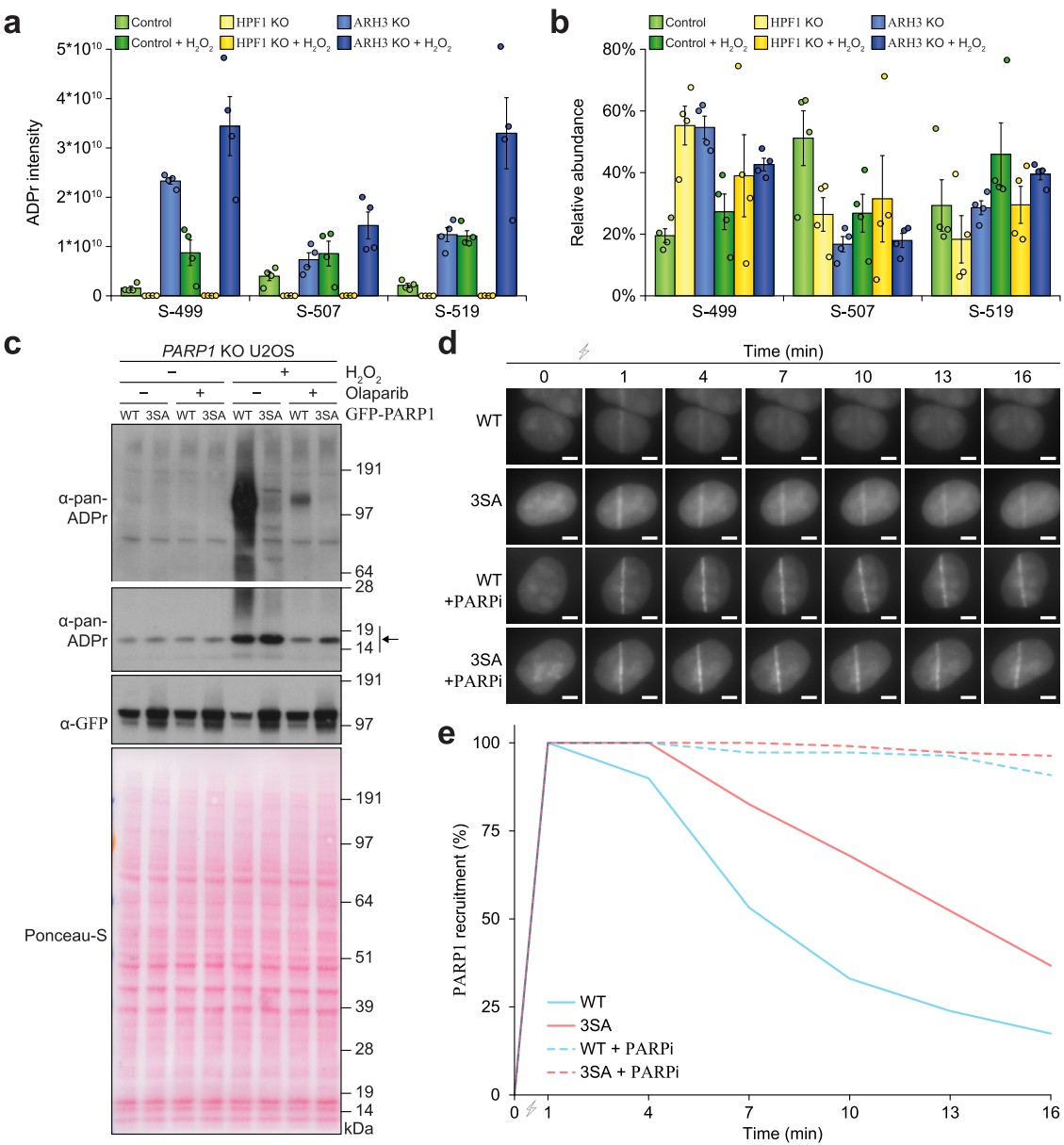

**Fig. 6 Mutational analysis of PARP1 serine auto-modification. a** PARP1 auto-modification analysis, showing absolute modification abundance. $n = 4$ cell culture replicates, data are presented as mean values ± SEM. **b** As **a**, but showing relative modification abundance. **c** Immunoblot analysis highlighting the ADPr equilibrium in PARP1 wildtype (WT) and 3SA mutant cells, ±Olaparib pre-treatment at 10 μM for 1 h ("PARPi"), ± H₂O₂ treatment at 5 mM for 30 min. The arrow indicates histone ADPr. This experiment was performed as two independent biological replicates, with similar results. **d** Representative live-cell microscopy images, visualizing GFP-PARP1 recruitment following laser microirradiation. The scale bar represents 5 μm. **e** Quantification of recruitment kinetics of GFP-PARP1 to damage stripes, as monitored via full visual dissipation of accumulated PARP1 signal. Stratification classes and all stratified data are available in Supplementary Fig. 5. PARP1 kinetics were significantly different between WT and 3SA mutant in the absence of PARPi, at $p = 1.0 \times 10^{-5}$ for $T = 1$, $p = 1.0 \times 10^{-5}$ for $T = 4$, $p = 7.9 \times 10^{-5}$ for $T = 7$, $p = 1.0 \times 10^{-5}$ for $T = 10$, $p = 2.4 \times 10^{-4}$ for $T = 13$, and $p = 0.017$ for $T = 16$, as determined via chi-squared testing on all stratified data (Supplementary Fig. 5B). $n = 109$ cells per condition. Source data are provided as a Source Data file.

greatly reduced compared to wild-type PARP1 (Fig. 6c, 1st panel), even though the 3SA mutant PARP1 was expressed at a modestly higher level (Fig. 6c, 3rd panel). Intriguingly, histone ADP-ribosylation was unaffected by the 3SA mutant (Fig. 6c, 2nd panel), and treatment with Olaparib abolished residual ADPr signal, demonstrating that the 3SA mutant retains in vivo catalytic activity. Next, we monitored recruitment kinetics of PARP1 WT and 3SA mutant to DNA damage, in the absence or presence of Olaparib, using a combination of live-cell microscopy and laser microirradiation (Fig. 6d and Supplementary Fig. 5A). Overall, treatment with Olaparib resulted in strong trapping of

PARP1, with a clear visible presence of PARP1 at sites of damage 15 min after microirradiation (Fig. 6e). Conversely, in the absence of Olaparib, PARP1 WT rapidly dissociated from damage stripes, with no PARP1 visible after 15 min in >75% of investigated cells. Intriguingly, the PARP1 3SA mutant resided significantly longer at DNA damage sites, with up to 50% of cells retaining visible PARP1 recruitment (Fig. 6e and Supplementary Fig. 5B). Taken together, we validate that auto-modification of PARP1 primarily occurs on S-499, S-507, and S-519, and find that mutation of these serine residues resulted in enhanced retention of PARP1 at sites of DNA damage.

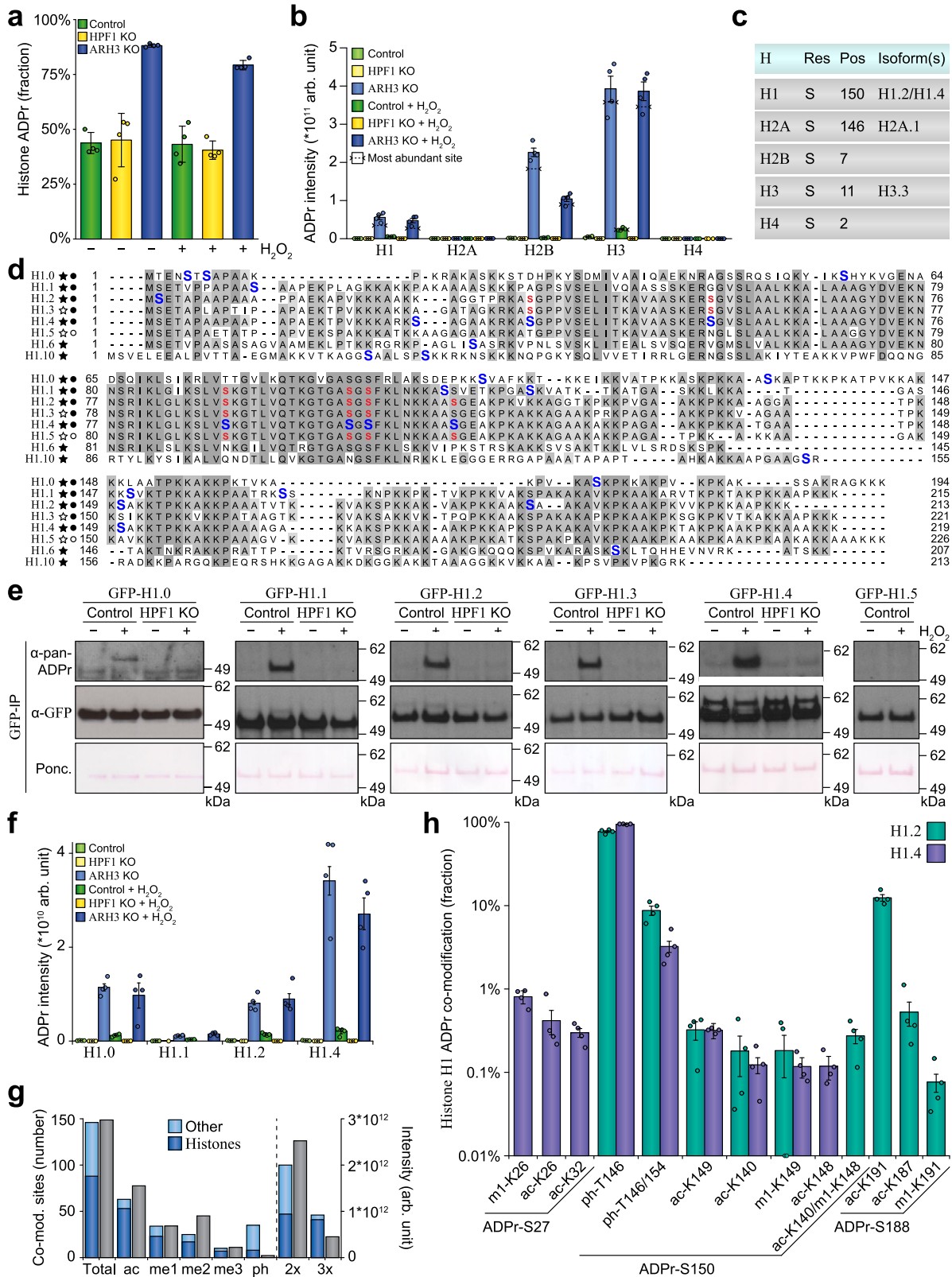

**Histone variants primarily contain serine ADPr.** The predisposition of histones to be ADP-ribosylated has been well-characterized[44–46]. In ARH3 KO cells, we found a striking prevalence of histone ADPr (Fig. 7a), whereas the relative fraction of histone ADPr was not notably altered in control and HPF1 KO cells, or in response to $H_2O_2$ treatment. Investigation of modification across all histone variants unveiled the highest degree of modification on H3 and H2B (Fig. 7b and Supplementary Fig. 6A), in accordance with previous in vivo observations[47–49]. On all histone variants, we noted that a single ADPr site accounted for the vast majority of total modification (Fig. 7b, c). Intriguingly, owing to the elevated levels of histone ADPr in the absence of ARH3, we found that Histone H1 constitutes the third-most ADPr-modified variant. Following this, we aligned the

**Fig. 7 A comprehensive analysis of the histone ADPr landscape. a** Overview of the fraction of ADPr residing on histones. $n = 4$ cell culture replicates, data are presented as mean values ± SD. **b** Histogram visualizing the distribution of ADPr abundance across histone family members. $n = 4$ cell culture replicates, data are presented as mean values ± SEM. **c** Overview of the most abundant ADPr site per histone variant. **d** Multiple sequence alignment of all Histone H1 isoforms detected to be ADP-ribosylated. ADPr-modified residues are highlighted in blue. Alternative non-unique assignments of the same ADPr peptide to different histone isoforms are highlighted in red. Black and white stars indicate unique and non-unique MS detection, respectively. Black and white circles indicate the presence or absence of histone ADPr via immunoblot analysis. **e** Immunoblot analysis accompanying. **d** Experiments were performed in wild-type or HPF1 KO HEK293T cells, transiently transfected with the indicated GFP-tagged histones for 24 h. $H_2O_2$ treatment was performed at 2 mM for 10 min, after which cells were lysed and GFP-IP was performed. This experiment was performed as two independent biological replicates, with similar results. **f** As **b**, but for Histone H1 isoforms. **g** Overview of the number of co-modified ADPr peptides (in blue) and the abundance of co-modified ADPr peptides (in gray). "ac"; acetylation, "me1"; mono-methylation, "me2"; di-methylation, "me3"; tri-methylation, "ph"; phosphorylation, "2×"; doubly modified (including ADPr), "3×"; triply-modified (including ADPr). **h** Visualization of the fractional abundance of ADPr co-modifications occurring on Histones H1.2 and H1.4. $n = 4$ cell culture replicates, data are presented as mean values ± SEM. Source data are provided as a Source Data file.

sequences of all Histone H1 isoforms and mapped all identified ADPr sites to the alignment (Fig. 7d). Our deep MS analysis facilitated direct identification of ADPr on Histones H1.0, H1.1, H1.2, H1.4, H1.6 (H1t), and H1.10 (H1x), with non-unique peptides also mapping to H1.3 and H1.5 (Fig. 7d). Immunoblot analysis was performed on GFP-tagged H1 isoforms, validating ADPr on H1.0, H1.1, H1.2, H1.3, and H1.4 (Fig. 7e and Supplementary Fig. 6B). With H1.5 not detected via unique peptides in the MS and not validated via IB, we suggest that H1.5 is not modified by ADPr. Whereas the absence of HPF1 abolished ADPr on Histone H1 (Fig. 7e), the absence of ARH3 conversely increased histone H1 ADPr levels (Supplementary Fig. 7A). Overall, in terms of modification abundance, H1.4 was the primary target of ADPr, followed by H1.0 and H1.2 (Fig. 7f). Site-directed mutagenesis of four main serine ADPr sites (>98% abundance as detected by MS) in Histone H1.2 eliminated detectable ADPr (Supplementary Fig. 7B), confirming our MS data and collectively highlighting that virtually all ADPr on histones occurs on serine residues.

**Multiple histone variants are co-modified by ADPr and other PTMs.** ADP-ribosylation can occur in the proximity of other post-translational modifications, such as phosphorylation[16,50], as well as methylation and acetylation in the context of histones[48,51–53], which could be indicative of crosstalk between the different PTMs. We investigated our data for ADPr peptides co-modified by phosphorylation, methylation, or acetylation, and in total, we were able to identify 240 unique co-modified peptides corresponding to 146 co-modified sites (Fig. 7g and Supplementary Data 8). Approximately, two-third of the peptides contained two PTMs, with the other one-third containing three PTMs. The majority of co-modification with methylation and acetylation was found to reside on histones, whereas co-modification with phosphorylation occurred predominantly on non-histone proteins (Fig. 7g). Crosstalk around Histone H3 S11-ADPr was previously described[48], which correlated well with our findings in the context of HPF1 and ARH3 knockout (Supplementary Fig. 7C), with acetylation of K15 overall being the major co-modification, followed by mono- and di-methylation of K10. We were able to visualize ADPr on Histone H1 to a great degree, allowing us to examine crosstalk with other PTMs. Histone H1.2 and H1.4 were the main targets of co-modification (Fig. 7h and Supplementary Data 8), with S27-ADPr and S188-ADPr flanked by methylation and acetylation for H1.4 and H1.2, respectively. Strikingly, S150-ADPr was co-modified for both H1.2 and H1.4, overall representing the highest degree of crosstalk, with frequent threonine phosphorylation at −4 or simultaneously at −4 and +4. Taken together, we demonstrate serine ADP-ribosylation targets a plethora of histone variants, particularly in the absence of ARH3, and we elucidate a high degree of co-modification of histones with serine ADPr and other PTMs.

## Discussion

Employing our Af1521 enrichment strategy for studying the global effects of HPF1 and ARH3 on the human ADP-ribosylome, we corroborate that ADPr is strongly induced upon depletion of ARH3 and conversely abrogated upon depletion of HPF1[18,19,26]. While HPF1 and ARH3 inversely regulate the serine ADP-ribosylome, the functional modularity is highly homogenous which contrasts previous observations related to ADPr of glutamate and aspartate[31].

In this study, we implemented and evaluated three biochemically distinct ADPr enrichment strategies (Figs. 2 and 3). Overall, the greatest number of ADPr-modified peptides was identified using the Af1521 macrodomain, with perceived peptide purity of >90%. Contrarily, enrichment using either antibodies or boronate affinity resulted in much lower sample purity; in the range of 5–20%. The main reason for this is specificity; the Af1521 macrodomain evolved for recognition of ADPr, whereas antibodies can have off-target specificity and boronic acid also interacts with nucleic acid components and carbohydrates[54]. Further, the amount of antibody used during enrichment was limiting, compared to Af1521 macrodomain that can be produced in-house in great quantities. In turn, the increased presence of unmodified and unrelated peptides in samples with low peptide purity results in reduced identification of ADPr-modified peptides, as mass spectrometric data acquisition is an intrinsically stochastic process[55]. Importantly, all three proteomics methods utilized here identified a predominance of serine ADPr, and identify a similar set of ADPr target proteins that is congruous with much other ADPr proteomics studies[10,14,16,29,31,36,42].

Intriguingly, while we find serine ADPr to be the major acceptor site across investigated cells, we did not observe a clear cellular switch in target residues from serine to glutamate or aspartate upon depletion of HPF1, although this was previously shown using immunoblot analyses[15]. Contrary, we were able to validate the HPF1-mediated PARP1 auto-modification switch from glutamate to serine residues in vitro, using ETD-based MS (Fig. 1). Still, under these in vitro conditions, the amount of glutamate ADPr observed was moderately lower compared to serine ADPr, which contrasts autoradiography observations[18]. However, whereas autoradiography and immunoblot detection generate signals for every ADPr moiety, our MS-based approach quantifies ADPr-modified peptides after conversion of PAR to MAR, and thus generates the same amount of signal regardless of the length of the ADPr polymer. MS-based approaches as outlined in this study are therefore suitable to elucidate the relative stoichiometry of the individual ADP-ribosylation acceptor sites. Glutamate-linked ADPr was only detectable after the addition of PARG in vitro, suggesting a predominant prevalence of PAR under these conditions, which could result in a diminished glutamate ADPr signal as perceived by the MS. This is in agreement with the observation that once mono-ADPr is

synthesized on the protein target, PARP1 very efficiently extends this ADPr primer into long PAR chains[56]. It is feasible that although we observed fewer glutamate and aspartate residues to be modified, these low-abundant acceptor sites could still harbor very long chains in vivo, which would in turn be able to regulate biological signaling.

Our observations could also reflect differences in experimental conditions and strategies for studying PARP enzymes. For example, it may be impractical to capture in vivo conditions in biochemical assays considering the existence of numerous PARP1 substrates and range of modulators[57–64]. Indeed, PARG inhibition of HeLa cells resulted in the accumulation of ADPr signal (Supplementary Fig. 2C), highlighting the dynamic nature of the PTM. It is plausible that this ADPr could be targeted to a multitude of residue types, including aspartate and glutamate, which would be congruous with previous proteomics observations that were exclusively carried out in the context of PARG knockout[10,31]. Whereas HA treatment of cellular lysates can give clues regarding the linkage specificity of ADPr, such treatment may reverse ADPr on residues other than glutamate and aspartate[65], and the exact treatment conditions during HA experiments can affect the observations. Thus, future unbiased MS-based proteomics studies would be beneficial for elucidation of the ADPr linkage architecture in the context of perturbation of potential ADPr writers, erasers, and co-factors.

Proteomic strategies used for mapping ADPr on glutamate and aspartate are based upon the derivatization of ADPr into a chemical mark ($+15.011$ Da) using HA[10]. While HA is a useful and established reagent to reverse ADPr ester bonds in biochemical assays, the use of HA can induce chemical artifacts ($+15.011$ Da) on acidic residues[37], which in proteomics experiments mimics the mark used for ADPr identification. Indeed, treatment of BSA with HA under native conditions revealed the same aberrant chemical marks ($+15.011$ Da), which could lead to erroneous identification of ADPr-modified aspartate and glutamate residues in the context of MS-based experiments (Supplementary Fig. 3). Overall, we find a larger fraction of hydroxamic acid modification in re-processed breast cancer cell data in which HA derivatization of ADPr was employed[31], compared to HA-treated BSA. We also observed a significant occurrence of asparagine and glutamine transamidation in the breast cancer cell line samples, although to a lesser extent compared to HA-treated BSA. Collectively, this could indicate that a part of the MS-resolved aspartate and glutamate hydroxamic acid modifications may be chemically induced rather than biologically. However, further investigations are necessary in order to detail the extent of these chemical marks in HA-based proteomics analysis of ADPr.

Contrary to this, our Af1521 methodology enriches peptides covalently modified with entire ADP-ribose moieties and is therefore not prone to similar chemical artifacts. Importantly, regardless of how ADPr site localization was performed, we find that boronate affinity chromatography enriches peptides derived from ADPr-modified proteins, and therefore the identified ADPr target proteins and their associated biological functions are highly likely to be precise.

We confirmed our proteomics data by orthogonal in vivo approaches, demonstrating that histone variants are predominantly modified with serine ADPr (Fig. 7e and Supplementary Figs. 6B and 7B). Although preferential H1 ADP-ribosylation on glutamate and aspartate residues was previously observed[66], these experiments were either performed in vitro in the absence of HPF1 and otherwise all in vivo experiments relied on MS readout of HA-derived hydroxamic acid residues which can intrinsically only affect glutamate and aspartate residues (*vide supra*).

To further our in vivo observations, we generated an auto-modification-deficient mutant PARP1 with S499A, S507A, and S519A substitutions, which displayed an increased retention time at sites of microirradiation-induced DNA damage (Fig. 6). This difference was abrogated in the context of PARP inhibition; further validating that auto-modification of PARP1 occurs on these three serine residues and is required for timely release from sites of DNA damage. During the revision of this manuscript, similar observations using a PARP1 3SA mutant were independently confirmed[67]. Crucially, Prokhorova et al. find that a PARP1 mutant with six glutamates to alanine substitutions in the auto-modification domain, does not exhibit any phenotype, thereby fortifying that serine-linked ADPr drives this PARP-trapping phenotype.

The enrichment of ADPr-modified peptides without downstream PARG treatment furthermore allowed us to infer the cellular MARylaton versus PARylation equilibrium, hereby uncovering that MARylation is a global and dominant phenomenon in human cells. While our observations capture the state of the cell upon lysis, and considering that ADPr is a very dynamic modification[68], the detected MARylation probably stems from substrates initially being modified with PAR, which is then rapidly degraded in vivo into MAR by high hydrolase activity of eraser enzymes[26,27,69], and PARP1 itself[70]. This dynamic regulation of ADPr has remained unnoticed for decades, which underscores the necessity for investigating ADPr dynamics under native conditions using unbiased proteomics strategies. Collectively, this work provides a powerful resource for researchers working on PARP1, ADPr, and PARP inhibitors; and the HPF1- and ARH3-dependent ADP-ribosylome presented here bolsters the global integrative view of cellular regulation.

## Methods

**Cell lines and cell culture**. HeLa cells (CCL-2, female), U-2 OS (U2OS) cells (HTB-96, female), and HEK293T cells (CRL-3216, female) were acquired via the American Type Culture Collection, and cultured at 37 °C and 5% $CO_2$ in Dulbecco's Modified Eagle's Medium (Invitrogen) supplemented with 10% fetal bovine serum and a penicillin/streptomycin mixture (100 U/mL; Gibco). U2OS cells with HPF1 knockout KO[19], U2OS cells with ARH3 KO[26], HEK293T cells with HPF1 KO[19], and HEK293T cells with ARH3 KO[69], were described previously. To generate cell lines stably expressing GFP-PARP1 wild type (WT) or 3SA mutant, U2OS PARP1 knockout cells[19] were transfected with pcDNA3.1-eGFP-PARP1 WT (UniProt ID P09874) or 3SA mutant (S499A, S507A, S519A; via GenScript). Transfection was performed using Lipofectamine 2000 (Thermo) following the manufacturer's instructions. Cells were selected using 0.6 mg/ml neomycin, after which single colonies were isolated for expansion. All cells were routinely tested for mycoplasma. Cells were not routinely authenticated.

**Plasmids and site-directed mutagenesis**. The mammalian expression H1.0 GFP-tagged pDEST47 (Invitrogen) construct was generated using LR Clonase II enzyme mix (Invitrogen) from H1.0 pDONR221 vector, acquired from DNASU Plasmid Repository. H1.1, H1.2, H1.3, H1.4, H1.5 GFP-tagged constructs were kind gifts from Gyula Timinszky. H1.2 GFP-tagged point mutants were made using Quik-Change Lightning Site-Directed Mutagenesis Kit (Agilent) following the manufacturer's protocol. A complete list of primers is available in Supplementary Table 1.

**Transfection and immunoprecipitation**. HEK293T cells were plated in 10-cm dishes and transiently transfected with an indicated plasmid for 24 h using Polyfect (Qiagen) following the manufacturer's instructions. The cells were washed with PBS and were lysed with Triton-X100 lysis buffer (50 mM Tris-HCl pH 8.0, 100 mM NaCl, 1% Triton X-100) supplemented with 5 mM $MgCl_2$, protease and phosphatase inhibitors (Roche), 2 μM Olaparib, 1 μM PARGi PDD00017273 at 4 °C. Protein concentrations were normalized using Bradford Protein Assay (Bio-Rad), and cell lysates were incubated with GFP-Trap MA magnetic agarose beads (ChromoTek) for 1 h while rotating at 4 °C. Beads were washed five times with Triton X-100 lysis buffer and eluted with 2× NuPAGE LDS sample buffer (Invitrogen) with TCEP (Sigma).

**Protein expression and purification**. PARP1 and HPF1 were purified as previously described[19]. PARP14CAT + WWE protein was purified as previously described in[71]. H1.0 was purchased from New England Biolabs (B2501S).

Expression vectors for PARP10CAT were described previously[72]. GST-tagged PARP10CAT(868-1025) was expressed in *E. coli* Rosetta (DE3) cells in LB media supplemented with 50 µg/µl ampicillin and 34 µg/µl chloramphenicol. Cultures were induced with 0.5 mM IPTG at 0.8–1.0 OD600 and protein was induced overnight at 16 °C. PARP10CAT(868–1025) cell pellet was resuspended in Lysis Buffer (PBS buffer supplemented with BugBuster protein extraction reagent, Benzonase, 10% glycerol, 1 mM DTT, and Complete Protease inhibitor cocktail), and lysed by the end over end mixing at 4 °C for 1 h. Then the cells were lysed thoroughly by Emulsi Flex-C5 homogenizer (Avestin) at 15,000 psi. The lysate was further centrifuged for 60 min at 35,000*g* and cleared lysate was applied to glutathione sepharose beads for 1 h at 4 °C. GST-tagged PARP10CAT(868–1025) was eluted using Lysis Buffer supplemented with 20 mM reduced glutathione. Eluted protein was further purified on a Superdex 200 column pre-equilibrated with 25 mM Tris-HCl pH 7.5, 150 mM NaCl, and 1 mM DTT.

**In vitro ADP-ribosylation experiments**. In vitro, ADPr experiments were essentially performed as described previously[21,71,72]. To obtain auto-modified PARP1, 2 µM His6-PARP1 was incubated with 1 µM activated DNA in 50 µl ADPr buffer (50 mM Tris-HCl, pH 8.0, 100 mM NaCl, 2 mM MgCl₂) with 100 µM NAD + for 2.5 h at 37 °C, and afterward frozen at −80 °C. For PARP10 and PARP14, 6.75 µM GST-PARP10CAT or 3.5 µM His6-PARP14(WWE + CAT) was auto-modified in 50 µl ADPr buffer with 100 µM NAD+ for 2.5 h at 37 °C and afterward frozen at −80 °C. For the PARP1-mediated H1.0 ADP-ribosylation, 2 µM His6-PARP1 was incubated with 5 µg H1.0 (New England Biolabs) in 50 µl ADPr buffer with 100 µM NAD+, and supplemented with 1 µM activated DNA in the presence or absence of 5 µM HPF1. Reactions were incubated at 37 °C for 2.5 h and afterward frozen at −80 °C. For subsequent MS analysis, samples were thawed, diluted 10-fold with 50 mM TRIS pH 8.5, and digested using 1:50 trypsin (w/w), overnight at room temperature. The resulting peptides were simultaneously reduced and alkylated via the addition of tris(2-carboxyethyl)phosphine (TCEP) and chloroacetamide (CAA) to a final concentration of 5 mM, at 37 °C for 1 h. Samples were split in two, with one-half mock-treated and the other half incubated with 1:500 PARG (w/w), overnight at room temperature shaking at 300 RPM. Finally, peptides were purified via C18 StageTip and analyzed by MS.

**HA treatment of BSA**. BSA (Sigma Aldrich, A7888) was dissolved in 200 mM HEPES pH 8.5, and sequentially digested using Lysyl Endopeptidase (Lys-C, 1:500 w/w; Wako Chemicals) for 1.5 h, and further digested overnight using modified sequencing grade Trypsin (1:500 w/w; Sigma Aldrich). Subsequently, the digest was either mock-treated or HA was added to a final concentration of 2 M, after which additional Lys-C and trypsin were added (1:1000 w/w). Samples were incubated overnight while mixing at room temperature, after which peptides were purified via C18 StageTip and analyzed by MS.

**Cell treatment**. For Af1521 and pan-ADPr antibody comparison experiments, ADP-ribosylation was induced in HeLa by treatment of the cells with 1 mM H₂O₂ (Sigma Aldrich) for 10 min in PBS at 37 °C. One batch of cells was prepared, totaling 2.7 billion cells, corresponding to approximately 75 million HeLa cells per technical replicate measured on the MS. For boronic acid enrichment experiments, HeLa cells were mock-treated or ADP-ribosylation was induced by treatment of the cells with 1 mM H₂O₂ for 10 min in PBS at 37 °C. For MS analysis, approximately 100 million HeLa cells were cultured per biological replicate. For HPF1 KO and ARH3 KO experiments, U2OS cells were mock-treated or ADP-ribosylation was induced by treatment of the cells with 1 mM H₂O₂ for 10 min in PBS at 37 °C. For MS analysis, approximately 150 million U2OS cells were cultured per biological replicate. For immunoblot analyses, HEK293T cells were mock-treated or ADPr was induced with 2 mM H₂O₂ in DPBS with calcium and magnesium (Gibco) for 10 min. All immunoblot analyses were performed in duplicate.

**Cell lysis and protein digestion**. The full procedure for proteolytic digestion of cells was done as described previously[14,16]. Briefly, cells were washed twice with ice-cold PBS and gently scraped at 4 °C in a minimal volume of PBS. Cells were pelleted by centrifugation at 500*g* and lysed in 10 pellet volumes of Lysis Buffer (6 M guanidine–HCl, 50 mM TRIS, pH 8.5). Complete lysis was achieved by alternating vigorous shaking with vigorous vortexing, for 30 s, prior to snap freezing of the lysates using liquid nitrogen. Frozen lysates were stored at −80 °C until further processing. Lysates were thawed and sonicated at 30 W, for 1 s per 1 mL of lysate, spread across 3 separate pulses. Tris(2-carboxyethyl)phosphine and chloroacetamide were added to a final concentration of 5 mM, and the lysate was incubated for 1 hour at room temperature. Proteins were digested using Lysyl Endopeptidase (Lys-C, 1:100 w/w; Wako Chemicals) for 3 h, and diluted with three volumes of 50 mM ammonium bicarbonate. Samples were further digested overnight using modified sequencing grade Trypsin (1:100 w/w; Sigma Aldrich). Digested samples were acidified by the addition of trifluoroacetic acid (TFA) to a final concentration of 0.5% (v/v), cleared by centrifugation, and purified using reversed-phase C18 cartridges (SepPak Classic, 360 mg sorbent, Waters) according to the manufacturer's instructions. Elution of peptides was performed with 30% ACN in 0.1% TFA, peptides were frozen overnight at −80 °C, and afterward lyophilized for 96 h.

**Purification of ADP-ribosylated peptides using Af1521 or antibodies**. Lyophilized peptides were dissolved in AP buffer (50 mM TRIS pH 8.0, 1 mM MgCl₂, 250 µM DTT, and 50 mM NaCl), after which either no enzyme, PARG enzyme (kind gift from Prof. Dr. Michael O. Hottiger), or ARH3 enzyme (kind gift from Prof. Dr. Bernhard Lüscher) were added in a 1:10,000 (w/w) ratio, overnight and at room temperature. For Af1521 and pan-ADPr antibody comparison experiments, no enzyme, PARG, or ARH3, were used as indicated in the figure legend. For HPF1/ARH3 KO experiments, PARG was added, theoretically in order to reduce ADPr polymers to monomers. GST-tagged Af1521 macrodomain was produced in-house using BL21(DE3) bacteria and coupled to glutathione Sepharose 4B beads (Sigma-Aldrich), essentially as described previously[14,16]. Pan-ADPr antibodies were a kind gift from CST, and 10 × 100 µL batches of E6F6A (AB1) and D9P7Z (AB2) were received pre-conjugated to agarose beads. For Af1521 and pan-ADPr antibody comparison experiments, 50 µL of sepharose beads with GST-tagged Af1521 or 50 µL of antibody beads were added to the samples and mixed at 4 °C for 3 h. For HPF1/ARH3 KO experiments, 150 µL of Af1521 beads were used. Beads were sequentially washed four times with ice-cold IP Buffer, two times with ice-cold PBS, and two times with ice-cold MQ water. On the first wash, beads were transferred to 1.5 mL LoBind tubes (Eppendorf), and LoBind tubes were exclusively used from this point on to minimize loss of peptide. Additional tube changes were performed every second washing step to minimize carryover of the background. ADP-ribosylated peptides were removed from the beads using two elution steps with two bead volumes ice-cold 0.15% TFA, and the pooled elutions were cleared through 0.45 µm spin filters (Ultrafree-MC, Millipore) and subsequently through pre-washed 100 kDa cut-off filters (Vivacon 500, Sartorius). The filtered ADP-ribosylated peptides were purified using C18 StageTips at high pH[16], and eluted as four or five fractions for comparison and HPF1/ARH3 KO experiments, respectively. Briefly, samples were basified by adding ammonium solution to a final concentration of 20 mM, and loaded onto StageTips carrying four layers of C18 disc material (punch-outs from 47 mm C18 3 M™ extraction discs, Empore). Elution was sequentially performed with 4%/10%/25% of ACN in 20 mM ammonium hydroxide for initial experiments, and 2%/4%/7%/25% of ACN in 20 mM ammonium hydroxide for HPF1/ARH3 KO experiments. The last fraction (F0) was prepared by performing StageTip purification at low pH on the flowthrough fraction from sample loading at high pH. All samples were completely dried using a SpeedVac at 60 °C and dissolved in a small volume of 0.1% formic acid. The final samples were frozen at −20 °C until measurement.

**Purification of ADP-ribosylated peptides using boronic acid**. Lyophilized peptides were dissolved in BA-AP buffer (50 mM TRIS pH 8.0, 1 mM MgCl₂, and 50 mM NaCl). m-Aminophenylboronic acid–Agarose resin was obtained from Sigma-Aldrich. Both saline suspension (A8530) and aqueous suspension (A8312) beads were acquired, and mixed in equal parts, due to ambiguity in the methods sections in published papers using boronate affinity chromatography in the context of ADPr purification[10,38]. A 50 µL of 4× pre-washed and equilibrated m-aminophenylboronic acid beads were added to the samples and mixed at room temperature for 1 h and at 4 °C for 2 h. Beads were sequentially washed four times with ice-cold BA-IP Buffer, two times with ice-cold PBS, two times with ice-cold high-salt PBS (NaCl at 300 mM), two times with ice-cold PBS, and once briefly with ice-cold MQ water. On the first wash, beads were transferred to 1.5 mL LoBind tubes (Eppendorf), and LoBind tubes were exclusively used from this point on to minimize loss of peptide. Additional tube changes were performed every fourth washing step to minimize carryover of the background. Peptides were removed from the beads using two elution steps with two bead volumes of ice-cold 0.15% TFA, and the pooled elutions were cleared through 0.45 µm spin filters (Ultrafree-MC, Millipore). The filtered peptides were purified using C18 StageTips[16] and eluted as seven fractions. All samples were completely dried using a SpeedVac at 60 °C and dissolved in a small volume of 0.1% formic acid. The final samples were frozen at −20 °C until measurement.

**MS dataset overview**. Dataset 1: In vitro experiments, related to Fig. 1, referred to as "DS1".

Dataset 2: Af1521 and antibody comparison, related to Fig. 2, referred to as "DS2".

Dataset 3: BSA and HA experiments, related to Supplementary Fig. 3, referred to as "DS3".

Dataset 4: Boronic acid experiments, related to Fig. 3, referred to as "DS4".

Dataset 5: HPF1 KO and ARH3 KO experiments, related to Fig. 4, referred to as "DS5".

**ETD-based mass spectrometric analysis**. Samples for DS1, DS2, DS4, and DS5, were measured using an Orbitrap Fusion™ Lumos™ Tribrid™ mass spectrometer (Thermo) and analyzed on 20-cm long analytical columns with an internal diameter of 75 µm, packed in-house using ReproSil-Pur 120 C18-AQ 1.9 µm beads (Dr. Maisch). On-line reversed-phase liquid chromatography to separate peptides was performed using an EASY-nLC™ 1200 system (Thermo), and the analytical column was heated to 40 °C using a column oven. Peptides were eluted from the column using a gradient of Buffer A (0.1% formic acid) and Buffer B (80% ACN in 0.1% formic acid). The primary gradient ranged from 3% buffer B to 24% buffer B

over 50 min, followed by an increase to 40% buffer B over 12 min to ensure elution of all peptides, followed by a washing block of 18 min. For DS1, gradient times were halved. ESI was achieved using a Nanospray Flex Ion Source (Thermo). Spray voltage was set to 2 kV, capillary temperature to 275 °C, and RF level to 30% (DS2,5) or 40% (DS1,4). Full scans were performed at a resolution of 60,000, with a scan range of 300–1750 m/z (DS2,5) or 375–1125 m/z (DS1,4), a maximum injection time of 60 ms, and an automatic gain control (AGC) target of 600,000 charges. Precursors were isolated from the entire full scan range (DS2,5) or a range of 400–1000 m/z (DS1,4). Isolation was performed at a width of 1.3 m/z, an AGC target of 200,000 charges, and precursor fragmentation was accomplished using electron transfer dissociation with supplemental higher-collisional disassociation (EThcD) at 20 NCE, using calibrated charge-dependent ETD parameters. Calibration of charge-dependent ETD parameters was essentially performed as described previously[73]. Charge-dependent ETD calibration resulted in ETD activation times of 48.39 ms for $z = 3$ precursors, 27.22 ms for $z = 4$ precursors, and 17.42 ms for $z = 5$ precursors. This equates to an ETD time constant ($\tau$) of 2.42 (DS2,5). Automatic calibration resulted in $\tau = 2.31$ for DS1 samples, and $\tau = 2.47$ for DS4 samples. Precursors with charge state 3–5 were isolated for MS/MS analysis and prioritized from charge 3 (highest) to charge 5 (lowest). Selected precursors were excluded from repeated sequencing by setting a dynamic exclusion of 60 s (DS2,4,5) or 30 s (DS1). MS/MS spectra were measured in the Orbitrap, with a loop count setting of 5 (DS2,5) or 3 (DS1,4), a maximum precursor injection time of 120 ms (DS2,5) or 120, 180, or 240 ms dependent on sample load (DS1,4), and a scan resolution of 60,000.

**HCD-type mass spectrometric analysis.** Samples for DS3 were measured using an Orbitrap Exploris™ 480 mass spectrometer (Thermo), using the same HPLC setup as outlined above. The primary elution gradient ranged from 5% buffer B to 30% buffer B over 15 min, followed by an increase to 90% buffer B over 10 min to ensure elution of all peptides, followed by a washing block of 15 min. ESI was achieved using a NanoSpray Flex™ NG ion source (Thermo). Spray voltage was set to 2 kV, capillary temperature to 275 °C, and RF level to 40%. Full scans were performed at a resolution of 60,000, with a scan range of 300–1750 m/z, a maximum injection time of 120 ms, and an AGC target of "100". Isolation was performed at a width of 1.3 m/z, an AGC target of "100", and precursor fragmentation was accomplished using HCD at 25 NCE. Precursors with charge state 2–7 were isolated for MS/MS analysis, and a dynamic exclusion of 2 s was used. MS/MS spectra were measured in the Orbitrap, with a loop count setting of 5, a maximum precursor injection time of 120 ms, and a scan resolution of 30,000.

**Data analysis.** Analysis of the MS raw data was performed using MaxQuant software[74,75], version 1.5.3.30. MaxQuant default settings were used, with exceptions outlined below. Five separate computational searches were performed, one for each separate MS dataset (DS1–5). For a generation of the theoretical spectral library (DS1,2,4,5,), a HUMAN.fasta database was downloaded from UniProt on the 24th of May, 2019. For DS1, recombinant protein sequences including epitope tags were co-searched along with the HUMAN.fasta. For DS3, only the BSA sequence (P02769) was searched. N-terminal acetylation, methionine oxidation, cysteine carbamidomethylation, and ADP-ribosylation on all amino acid residues known to potentially be modified (C, D, E, H, K, R, S, T, and Y), were included as variable modifications (DS1,2,4,5). For DS5, additionally, lysine acetylation, lysine mono-, di-, and tri-methylation, and serine, threonine, and tyrosine phosphorylation, were included as variable modifications. For DS3, N-terminal acetylation, oxidation (M,N,Q), deamidation (N,Q), and hydroxamic acid modification (D,E,M), were set as variable modifications. For the DS5 first search, which is only used for mass recalibration, phosphorylation was omitted as a variable modification. Up to 3 (DS3), 5 (DS5), 6 (DS1,2), or 10 (DS4) missed cleavages were allowed, a maximum allowance of 3 (DS4,5) or 4 (DS1,2,3) variable modifications per peptide were used, and maximum peptide mass was set to 3900 (DS5) or 4600 Da (DS1–4). Second peptide search was enabled (DS2,4,5). Matching between runs was enabled with an alignment time window of 10 min (DS1,3) or 20 min (DS2,4,5), and a match time window of 30 s (DS1,3), 42 s (DS2), or 60 s (DS4,5). A first search precursor mass tolerance of 15 ppm (DS1,4,5) or 20 ppm (DS2,3) was used, and the main search precursor mass tolerance of 3 ppm (DS1,4,5) or 4.5 ppm (DS2,3) was used. For fragment ion masses, a tolerance of 20 ppm was used. Modified peptides were filtered to have an Andromeda score of >40 (default), and a delta score of >20. Data were automatically filtered by posterior error probability to achieve a false discovery rate of <1% (default), at the peptide-spectrum match and the protein assignment levels. A site decoy fraction of 1% (DS2,5) or 2% (DS1,3,4) was applied. Re-processing of MS raw data from Zhen et al.[31] was performed with settings identical to those used for DS3, with the HUMAN.fasta database used for the search.

**Data filtering.** Beyond automatic filtering and FDR control as applied by MaxQuant, the DS2 and DS5 data were manually stringently filtered in order to ensure proper identification and localization of ADP-ribose, as this data constitutes the main Af1521 ADPr datasets. PSMs modified by more than one ADP-ribose was omitted. PSMs corresponding to unique peptides was only used for ADP-ribosylation site assignment if localization probability was >0.90, with localization

of >0.75 accepted only for purposes of intensity assignment of further evidence. Erroneous MaxQuant intensity assignments were manually corrected in the sites table, and based on localized PSMs only (>0.90 best-case, >0.75 for further evidence). For DS1 and DS4 data, multiply-modified peptides were accepted, and localization >0.75 was accepted. For DS3 and re-processed data from Zhen et al., the "evidence.txt" file from MaxQuant was filtered and used for quantification. For the ADP-ribosylation target proteins table derived from the HPF1/ARH3 KO data (DS5), the proteinGroups.txt file generated by MaxQuant was filtered to only contain those proteins with at least one ADP-ribosylation site detected and localized post-filtering as outlined above, with cumulative ADP-ribosylation site intensities based only on localized evidence.

**Immunoblot analysis.** U2OS cells were lysed in STBS buffer (2% sodium dodecyl sulfate, 150 mM NaCl, 50 mM TRIS-HCl pH 8.5) at room temperature. U2OS lysates were homogenized by shaking at 99 °C at 1400 RPM for 30 min. HEK293T cells were lysed with Triton-X100 lysis buffer (50 mM Tris-HCl pH 8.0, 100 mM NaCl, 1% Triton X-100) supplemented with 5 mM MgCl₂, protease and phosphatase inhibitors (Roche), 2 μM Olaparib, 1 μM PARGi PDD00017273 at 4 °C. HEK293T lysates were incubated with 0.1% Benzonase (Sigma) for 30 min at 4 °C, centrifuged at 16,900g for 15 min, and the supernatants were collected. Protein concentrations were analyzed by Bradford Protein Assay (Bio-Rad). Prior to loading, lysates were supplemented with 1× NuPAGE LDS sample buffer (Invitrogen) with DTT or TCEP, and size-separated on 4–12% Bis–Tris gels using MOPS running buffer. For U2OS samples, proteins were transferred to Amersham™ Protran® nitrocellulose membranes (GE Healthcare). For HEK293T samples, proteins were transferred onto nitrocellulose membranes (Bio-Rad) using Trans-Blot Turbo Transfer System (Bio-Rad). Equal total protein loading was ensured by Ponceau-S staining. Membranes were blocked using 5% BSA solution (for U2OS samples) or 5% non-fat dried milk (for HEK293T samples) in PBS supplemented with 0.1% Tween-20 (PBST) for 1 h. Subsequently, membranes were incubated with primary antibodies overnight at 4 °C and afterward washed three times with PBST. U2OS experiment membranes were incubated with Goat-anti-rabbit HRP conjugated secondary antibody (Jackson Immunoresearch, 111-036-045) or Goat-anti-mouse HRP conjugated secondary antibody (Jackson Immunoresearch, 115-036-062), at a concentration of 1:10,000 for 1 h at room temperature. HEK293T experiment membranes were incubated with peroxidase-conjugated secondary anti-rabbit antibody (Agilent, P0399), at a concentration of 1:3000 for 1 h at room temperature. Membranes were washed three times with PBST prior to detection using Novex ECL Chemiluminescent Substrate Reagent Kit (Invitrogen). The following primary antibodies were used for immunoblot analysis in this study and diluted at 1:1000 unless stated otherwise. For U2OS experiments: Poly/Mono-ADP Ribose (E6F6A) Rabbit mAb #83732 (CST), PARP1 (46D11) Rabbit mAb #9532 (CST), HPF1 Rabbit pAb HPA043467 (Atlas Antibodies), ADPRHL2 (ARH3) Rabbit pAb HPA027104 (Atlas Antibodies), GAPDH Rabbit pAb ab9485 (Abcam), GFP Mouse mAb clone 7.1 and 13.1 (Roche). For HEK293T experiments: Pan-ADPr (MABE1016) Rabbit (Millipore); at 1:1500, ADPRHL2 (ARH3) Rabbit pAb HPA027104 (Atlas Antibodies); at 1:2000, GFP (ab290) Rabbit (Abcam); at 1:5000, custom-made HPF1 antibody was previously described[19].

**Live-cell imaging and quantification.** Cells were seeded onto glass coverslips in a Lab-Tek borosilicate imaging chamber (Nunc), and pre-sensitized with fresh media containing 10 μM 5-bromo-2′-deoxyuridine (BrdU) 24 h prior to imaging. Before imaging the cells, the medium was replaced with a CO₂-independent medium without phenol red (Gibco), supplemented with 10% fetal bovine serum and penicillin/streptomycin mixture (100 U/mL; Gibco), and either containing 10 μM Olaparib or not. Cells were incubated with Olaparib for at least 1 h before imaging. During imaging, cells were kept in a heating chamber maintained at 37 °C. The live-cell time-lapses were imaged on a Zeiss-AxioObserver (PALM-Axiovision) fluorescence widefield microscope using a 40×/0.6 DIC II LD Plan Neofluar objective and recorded using a Hamamatsu ORCA Flash 4.0 LT camera. Laser microirradiation was performed using a CryLaS pulsed UV-A laser (355 nm), by drawing a straight path across the nuclei using the Zeiss PALMRobo Software (version 4.9) and using cut energy of 28, LPC energy Delta of 27, and a cutting speed of 10. To quantify the dissociation kinetics of PARP1 from sites of DNA damage, we categorized the extent of PARP1 in four classes (recruitment not visible, barely visible, moderately visible, or strongly visible). Cells that were out of focus, cells without a laser stripe at the earliest time point, mitotic cells, cells at the border of the image, cells with oversaturated nuclear signal, and cells with nuclear signal indiscernible from the background, were excluded from the analysis. An equal number of cells ($n = 109$) was randomly sampled from each experiment to generate graphs. Chi-squared testing was used to determine significance.

**Quantification and statistical analysis.** Details regarding the statistical analysis can be found in the respective figure legends. Statistical handling of the data was primarily performed using the freely available Perseus software[76] and includes term enrichment analysis through FDR-controlled Fisher Exact testing, density estimation for highly populated scatter plots, volcano plot analysis, hierarchical clustering, and principal component analysis. Protein Gene Ontology annotations

and UniProt keywords used for term enrichment analysis were concomitantly downloaded from UniProt with the HUMAN.fasta file used for searching the RAW data. Sequence context analysis was performed using iceLogo software, version 1.2[77]. Multiple sequence alignment was performed using Clustal Omega as integrated into UniProt, using the (default) transition matrix Gonnet, gap opening penalty of 6 bits, gap extension of 1 bit, and using the HHalign algorithm and its default settings as the core alignment engine[78].

**Reporting summary**. Further information on research design is available in the Nature Research Reporting Summary linked to this article.

## Data availability

The mass spectrometry proteomics data generated in this study have been deposited in the ProteomeXchange Consortium via the PRIDE[79] partner repository, under accession codes PXD023835 and PXD027504. All other data generated in this study are provided in the Supplementary Information/Source Data file. Source data are provided with this paper. The breast cancer cell line proteomics data[31] used in this study are available in the Chorus database under accession code 6153436701505083307. Source data are provided with this paper.

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

## Acknowledgements

The work carried out in this study was in part supported by the Novo Nordisk Foundation Center for Protein Research, the Novo Nordisk Foundation (grant agreement numbers NNF14CC0001 and NNF13OC0006477), Danish Council of Independent Research (grant agreement numbers 4002-00051, 4183-00322A, 8020-00220B, and 0135-00096B), and The Danish Cancer Society (grant agreement R146-A9159-16-S2). The proteomics technology applied was part of a project that has received funding from the European Union's Horizon 2020 research and innovation program under grant agreement EPIC-XS-823839. Work in I.A.'s laboratory was supported by Wellcome Trust (101794 and 210634); Biotechnology and Biological Sciences Research Council (BB/R007195/1); Ovarian Cancer Research Alliance (813369); and Cancer Research United Kingdom (C35050/A22284). We would like to thank members of the NNF-CPR Mass Spectrometry Platform for instrument support and technical assistance, and Marcin Suskiewicz and Johannes Rack for providing reagents. Microscopy-based work was carried out with the support of the CPR Protein Imaging Platform.

## Author contributions

S.C.B.-L., I.A.H. and A.K.L.F.S.R. prepared MS samples. I.A.H. and S.C.B.-L. measured all samples on the mass spectrometer, processed all MS raw data, and performed the data analysis. S.C.B.-L., E.P., J.D.E. and I.A.H. performed the IB analysis. K.Z. and D.A. performed in vitro ADP-ribosylation assays. J.D.E. performed microscopy experiments and data analysis in consultation with C.L. M.L.N. conceived the project. M.L.N. and I.A. supervised the project. I.A.H., S.C.B.-L. and M.L.N. wrote the paper with input from all authors.

## Competing interests

The authors declare no competing interests.
