## [Peer Review File · Nature Communications]

REVIEWER COMMENTS

Reviewer #1 (Remarks to the Author):

General comments

In this manuscript, Ivo A. Hendriks and colleagues explored the ADP-ribosylome of human cells depleted of HPF1 and ARH3. Major conclusions are that mono-ADP-ribosylation predominates in cells and that SER-ADP-ribosylation is independent of HPF1. Michael Nielsen's group has accustomed us to very high-level papers and his contribution to the field of ADP-ribosylation remains unmatched in terms of mass spectrometry development. Once again, this paper is entirely appropriate and is part of a scientific approach which aims to clarify certain concepts actively debated in the PARP community. The development of its experimental approach coupled with Macro-AF1521/PARG and ETD fragmentation was at the origin of a major change of course concerning the type of amino acid targeted by the activity of PARP-1. Dr. Nielsen's group published a series of papers which attempted to demonstrate the absence of experimental bias of their method with respect to the type of amino acid acceptor of the modification, using both biochemical and bioinformatic approaches. There is no doubt that the results presented in the present study are of very high quality, but a fundamental problem remains : an alternative MS-based approach developed by the group of Yonghao Yu shows that glutamate and aspartate ADP-ribosylation is a very significant modification following PARP-1 activation by genotoxic insults. Their study does not cast doubt on the existence of serine ADP-ribosylation as ASP/GLU/SER ADP-ribosylation is likely to co-occur in cells. The problem is that the Macro-AF1521/PARG/ETD approach is basically excluding the existence of ASP/GLU ADP-ribosylation, a conclusion that goes against the work of Yu's group and others. Nielsen's group did a fantastic exploratory job to clear up this inconsistency but a final and conclusive demonstration is still awaited by the PARP community. If we cannot be sure that D/E ADP-ribosylation can be detected by the current Macro-AF1521/PARG/ETD approach, how can we be sure that ADP-ribosylation is mostly limited to SER-ADP-ribosylation and how can we conclude anything regarding a switch from ASP/GLU to SER ADP-ribosylation? Although the use of ETD fragmentation has been shown to be better than CID fragmentation for site-specific assignment of particularly labile post-translational modifications, doubts remain as to its effectiveness in locating ASP/GLU ADP-ribosylation. As previously mentioned, Nielsen's group used a variety of approaches to demonstrate the absence of bias in their experimental pipeline. These demonstrations are strong and very convincing from a biochemical point of view but the recurring problem is that it's always the same MS technique, developed in their own laboratory, which is used to draw the final conclusions. Since alternative approaches exist and directly conflict with their method, it is difficult to understand why the demonstration of the absence of experimental bias does not involve a parallel confrontation with the boronate/hydroxylamine versus Macro-AF1521/PARG approaches. On several occasions, Nielsen's group discusses the possibility that the derivatization of ASP/GLU ADP-ribosylation is only an experimental artifact. The problem with this claim is that it doesn't hold up if you look at the protein profile isolated by the boronate/hydroxylamine approach and the quantitative profiling of ASP/GLU ADP-ribosylation in the presence of specific PARP-1 inhibitors. Boronate affinity purification has been broadly used for decades for the selective isolation and enrichment of cis-diol-

containing molecules such as PAR. Technically, boronate-diol complexation is not dependent on the ADP-ribosylation amino acid linkage and therefore can be used for unbiased isolation of all types of ADP-ribosylated peptides. I think the only way to truly establish one approach against another would be to take the protocol of each method and subject it to the two alternative methods of derivatization.

Specific comments

In the first part of their manuscript, the authors performed a series of in vitro biochemical assays to evaluate if their Macro-AF1521 enrichment strategy is unbiased with regard to identification of amino acid acceptor sites. This is absolutely critical to draw any conclusions about the regulatory landscape of the human HPF1- and ARH3-dependent ADP-ribosylome. The strategy developed by the group of ML Nielsen in the last years is based on the use of PARG to convert heterogeneous PAR polymers into mono-ADP-ribose remnants with a unique +541 Da mass increment signature. In the absence of PAR hydrolysis, PARylated peptides cannot be analyzed by mass spectrometry because of their complex chemical structures. In this study, the authors show that the number of SER-ADP-ribosylation sites is basically the same whether PARG is used or not to convert PAR into MAR signatures in cell extracts in which PARP-1 was hyperactivated. The authors therefore come to the conclusion that mono-ADP-ribosylation is predominant in human cells. This observation also implies that the ADP-ribosylation landscape is not fundamentally modified following PARP-1 activation. This is a shocking statement for many reasons. I will get back to that.

The authors used two anti-MAR/PAR antibodies as complementary approaches to isolate ADP-ribosylated peptides to ensure that their MS-based proteomics strategy using Macro-AF1521 enrichment preserves ADP-ribosylation. The authors mentioned that a residual MAR hydrolase activity of Macro-AF1521 could have been responsible for a bias towards certain types of ADP-ribosylation linkages. Since anti-MAR/PAR antibodies do not possess MAR hydrolase activity and that SER-ADP-ribosylation was the only significant modification found, the authors concluded that their approach is unbiased. There are a number of problems with this conclusion. In theory, if anti-MAR/PAR antibodies are targeting the same MAR/PAR modifications as Macro-AF1521, one would expect to identify a similar number of ADP-ribosylation sites. However, the profiles are very different with AB1/AB2 versus Macro-AF1521 in Fig-1B. Is it very common for anti-PAR antibodies to have much more affinity for long and complex PAR than MAR. The affinity for PAR by an anti-MAR/PAR antibody is likely introducing a bias for PARylation versus MARYlation. The main concern regarding the bias for SER-ADP-ribosylation do not come from the potential selective MAR hydrolysis activity of Macro-AF1521 but the capacity of the ETD fragmentation approach (or the selected experimental strategy) to identify ASP/GLU ADP-ribosylation. If PARylation occurs on D/E residues and that this specific linkage is unstable under the authors experimental conditions, we would expect to see a decrease in the number of SER-ADP-ribosylation sites following PARP-1 activation when peptides are isolated with antibodies. This is what is reported in Fig-1B. If the peptide fragmentation strategy is biased for site-specific localization of SER-ADP-ribosylation, we would also expect that any other type of ADP-ribosylation chemical linkages, including D/E ADP-ribosylation, would be insignificant when analyzing the total number of ADP-ribosylation sites. So, even if PARG is used to generate D/E MARYlated peptides in complex samples, these peptides won't be identified and SER-

ADP-ribosylation will appear unchanged with or without PARG treatment, as can be observed in Fig-1B. The shocking proclamation that PARylation predominates in cells, even after extensive PARP-1 activation with hydrogen peroxide needs more evidence. Obviously, all of these assumptions hold true to the extent that we consider that there is indeed a bias for SER-ADP-ribosylation.

In contrast to what is cited in the manuscript and in the supplementary note, it has been demonstrated that hydroxylamine do not react nonspecifically with free aspartic acid and glutamic acid residues (at least in the experimental conditions described by the group of Yonghao Yu in *Nature Methods*, 2013). The authors should use the m-aminophenyl boronate affinity-purification of PARylated peptides developed by Yu's group coupled to their MS-based approach to see if there is really a bias for specific ADP-ribosylation sites. In theory, they could be able to easily identify D/E ADP-ribosylated peptides with ETD fragmentation. The intensity of basically all D/E ADP-ribosylated peptides is strongly decreased after treatment with a PARP inhibitor (even reaches a 90-99% decrease for PARP-1 automodification sites). If hydroxylamine derivatization was just an artifact, such PARPi-specific modulation of D/E ADP-ribosylation would not occur. If D/E ADP-ribosylation is not detectable using Yu's approach coupled to ETD that could indicate a bias for non ASP/GLU linkages, such as SER-ADP-ribosylation. Conversely, hydroxylamine could be used in peptide extracts isolated with Macro-AF1521 versus m-aminophenylboronate for bias analysis. I think that the validation of the unbiased approach described in the present study by Hendriks IA and colleagues should be based on MS rather than on biochemical assays targeting selective hydrolysis of specific ADP-ribosylation linkages.

The authors mentioned in the manuscript that treatment of cell lysates with hydroxylamine did not change the distribution of ADP-ribosylation, suggesting that ASP/GLU ADP-ribosylation is not significant (Fig. S2C). The conditions used to hydrolyze PAR with hydroxylamine in cell extracts is not described in the manuscript. Automodified PARP-1 has been shown to be extremely sensitive to hydroxylamine. A control is missing when using hydroxylamine for Western blot analysis of residual PARylation. Lysis buffer supplemented with automodified PARP-1 treated with/without hydroxylamine should be used to visualize the efficiency of the reaction.

For protein digestion, the authors used a two-step approach. Proteins are first digested with LysC in 6M guanidine hydrochloride buffer, followed by trypsin after a dilution of the extract in ammonium bicarbonate to lower guanidine concentration to a level compatible with trypsin activity. This is a typical procedure that is normally very effective to digest proteins to completion (especially a two-step method with a strong chaotropic/denaturing agent such as guanidine hydrochloride). However, peptides with a surprisingly high number of missed cleavages are identified with site-specific ADP-ribosylation sites (i.e. on serine for the vast majority of peptides). Systematically, very large peptides with an unusual high number of missed cleavages are listed (e.g. Supplementary Table-1). Under these conditions, we can practically speak of partial trypsin/LysC digestion. However, the digestion conditions described in the Materials section is not presented as such. For example, the peptide AEPVEVVAPRGKSGAALSCKK(ADPr)KGQVK (5 missed cleavages) and several other similar peptides of varying length are almost always missed cleaved after the arginine. However, the peptide AEPVEVVAPR is consistently identified in complex mixtures of PARylated

peptides. This is a very efficient cleavage site that generates a typical PARP-1 peptide in basically all MS-based studies. The authors should explain why such a high number of missed cleavages are observed in their data sets. Trypsin is known to be sensitive to post-translationally modified amino acids located next to the K/R cleavage site so we would expect the KS(ADPr) to be skipped but not at unmodified cleavage sites to such extent. Following the same idea, a large number of peptides starting with S(ADPr) in N-terminus are listed in Supplementary Table-1 (>400 peptides) suggesting that trypsin cuts at K/R residues that precede major SER ADP-ribosylation sites. This is surprising because, for example, K/R residues are often not cleaved by trypsin when a phosphorylated serine is located adjacent or in close proximity. Considering the charge density of PAR and its steric hindrance, efficient cleavage of KS(ADPr) sites is unexpected. This should be discussed because this unusual peptide features are not found with other MS-based techniques used to localize site-specific ADP-ribosylation sites. Peptides with numerous missed cleavages are generally more hydrophobic and have a higher overall charge. Can it introduce a bias in the present MS analysis pipeline?

In the last part of the manuscript, the authors analyzed the co-occurrence of other histone PTMs with ADP-ribosylation. The authors did not mention that ASP/GLU ADP-ribosylation of histones was reported by the group of BA Garcia (Mol Biosyst, 2017). This group used a quantitative approach to show that the abundance of ASP/GLU ADP-ribosylation sites increases in abundance over time after PARP-1 activation. They also showed that histone H1 is a major ADP-ribosylation target. This study cannot be ignored and should be discussed. Garcia BA and colleagues also report that treatment with hydroxylamine demonstrated a near complete depletion of signal, indicating that a vast majority of histone ADP-ribosylation sites are occurring on ASP/GLU residues. To ensure that the hydroxylamine is not chemically modifying histone proteins, they performed a control experiment where they incubated recombinant histone proteins with hydroxylamine overnight. They did not identify any derivitized sites, indicating that the hydroxylamine does not chemically modify ASP/GLU residues on histones. It should be noted that other papers are also reporting the specific ASP/GLU ADP-ribosylation of histone proteins (e.g. site-specific ADP-ribosylation of histone H2B at GLU-18 and GLU-19 in response to DNA double strand breaks, Rakhimova A. et al. Scientific Reports, 2017).

As long as a formal demonstration regarding the preferential targeting of serine residues in the ADP-ribosylation process is not fully established, it would be preferable to acknowledge the fact that, perhaps, both ASP/GLU and SER ADP-ribosylation exists in cells. There is a very significant overlap in the protein profiles of ADP-ribosylated proteins identified by the boronate/hydroxylamine and Macro-AF1521/PARG approaches. As seen in Fig-S2B of this manuscript, RNA-binding proteins, chromatin regulators and DNA damage response factors are predominant in data sets of proteins identified with the Macro-AF1521/PARG approach, just as it can be observed with the boronate/hydroxylamine approach. A similar over representation of PARP-1 targets has also been shown using a NAD⁺ analog-sensitive approach for PARPs coupled to hydroxylamine derivitization (Gibson BA et al. Science 2016). These converging results provide strong evidence that there is a real relationship between the identification of ADP-ribosylation sites, by both approaches, so that one does not exclude the other.

Reviewer #2 (Remarks to the Author):

Hendriks and colleagues have successfully developed a quantitative proteomics method to explore the ADP-ribosylated proteome in the presence and absence of the regulatory proteins HPF1 and ARH3. Utilization of this method has revealed a set of ~1,500 ADP-ribose sites that are differentially modified across the investigated knockout cell lines. Importantly, their technique has helped corroborate the roles for ARH3 and HPF1 in the ADP-ribosylation of serine residues. Further, they demonstrate that the addition of PARG used in prior iterations of their technique is not strictly required as most of the identified ADP-ribose sites already exist as mono-ADP-ribose sites prior to exogenous PARG treatment. This is a very useful and interesting article that will boost various proteomic and biochemical research efforts in the field, while highlighting the need for further mechanistic characterization of the ADP-ribose modification pathway. The following are a few specific comments regarding the manuscript:

1. One of the main arguments that the authors propose from their work is that there is little to no modification of glutamate or aspartate residues in the experimental conditions that they are evaluating. This point would be in contrast to multiple published reports of acidic residue targeting within the PARP family of enzymes. An alternate explanation that might be consistent with their observations is that the removal of ADP-ribose from Ser and Glu/Asp residues are governed by significantly different kinetics. In this alternate explanation, the absence of Glu/Asp modifications in their survey could be explained by a prior, and fast, removal of ADP-ribose before they generate their trypsinized peptide libraries. This removal activity might be due to the endogenous activity of PARG itself or another member of the glycohydrolase family (i.e. TARG or MARG). It might bolster their case if they performed an additional experiment wherein endogenous PARG activity was halted (through KO or inhibitor) prior to lysis and a similar plot to figure S2C was prepared in non-stressed and H₂O₂ treated cells. Such an experiment would not definitively rule out the activity of another Glu/Asp specific eraser, but would provide useful data for their primary argument. Regardless, I think it would be appropriate to engage with this possible alternate explanation within the context of the discussion.

2. The authors' observation of extensive cross-talk between multiple PTMs and ADP-ribose is a particularly intriguing result. I was curious if the system they already have in place might be able to address whether ADP-ribose is necessary and/or sufficient to drive PTM cross-talk. When they generated their GFP-H1.2 S150A mutant, did they attempt to determine whether the mutant form was still phosphorylated on the neighboring threonine? Further, was the 4xSA mutant still alkylated to a similar extent as the wild-type H1.2? Ultimately, these experiments might be beyond the scope of the current work, but their inclusion would bolster the impact of this report even further and would provide an exciting inroad into the potential mechanisms governing ADP-ribose at the histone.

The statistical analysis performed herein is well-detailed, valid, and appropriate for the methods employed by the authors. Based on the included data I am convinced that the research presented could be reproduced successfully in an independent laboratory.

Reviewer #3 (Remarks to the Author):

The manuscript by Hendriks et al. „The regulatory landscape of the human HPF1- and ARH3-dependent ADP-ribosylome” presents a proteomic analysis of the ADP-ribosylation in human cell lines with the KO of essential PARP1 regulatory proteins (HPF1 and ARH3). The work is mostly descriptive, cataloging the modification sites in these cell lines and comparing the site occupancies between the mutants and the wild type cells. The study is technically sound and provides a resource of interest to the specialized field of ADP-ribosylation. It doesn't go beyond the resource and therefore it may be too preliminary and descriptive, lacking significance matching the standard of Nature Communications. The major suggestion is to expand on the biological significance of the specific sites, especially those regulated by the enzymes depleted in the examined cell lines.

Other comments:

1. The authors state that their finding of mostly serine residues being modified support the notion that their methodology is unbiased (page 5). However, the statement that the serine residues are the major target of ADP-ribosylation, comes from their own work using the same methodology, so this seems to be an unfounded statement. It would be advisable to see the comparison with other purification methodologies using the same cell lines.
2. The manuscript seems to be hastily assembled. Some references are missing (for example, page 17, MaxQuant software). It also requires significant copyediting as it is full of errors that are not typos (for example, on page 5, incorrect use of the verb “support” which takes a noun, so the two sentences containing this verb are completely wrong), but there are also numerous typos rendering the manuscript difficult to read.

Overview of manuscript changes

Previous version	Revised version		Previous version	Revised version	
	Fig. 1A Fig. 1B Fig. 1C Fig. 1D	New New New New	Fig. S1A Fig. S1B	Fig. S1A Fig. S1B Fig. S1C	New
Fig. 1	Fig. 2		S.N. A S.N. B S.N. C S.N. D S.N. E S.N. F S.N. G S.N. H S.N. I S.N. J	Fig. S2A Fig. S2B Fig. S2C Fig. S2D Fig. S2E Fig. S2F Fig. S2G Fig. S2H Fig. S2I Fig. S2J	Revised Revised Revised Revised Revised
Fig. 2	Fig. 4				
Fig. 3A Fig. 3B Fig. 3C Fig. 3D Fig. 3G	Fig. 5A Fig. 5B Fig. 5C Fig. 5D Fig. 5E		Fig. S2A Fig. S2B Fig. S2C	Fig. S3A Fig. S3B	Removed
Fig. 3E Fig. 3F	Fig. 6A Fig. 6B Fig. 6C Fig. 6D Fig. 6E	Moved Moved New New New		Fig. S4A Fig. S4B	New New
Fig. 4	Fig. 7		Fig. S3 Fig. S4	Fig. S5 Fig. S6	
			Table S1	Table S1 Table S2 Table S3 Table S4 Table S5	New New New New
			Table S2 Table S3 Table S4	Table S6 Table S7 Table S8	

Point-by-point response to Reviewer comments

Reviewer #1 (Remarks to the Author):

General comments

In this manuscript, Ivo A. Hendriks and colleagues explored the ADP-ribosylome of human cells depleted of HPF1 and ARH3. Major conclusions are that mono-ADP-ribosylation predominates in cells and that SER-ADP-ribosylation is independent of HPF1. Michael Nielsen's group has accustomed us to very high-level papers and his contribution to the field of ADP-ribosylation remains unmatched in terms of mass spectrometry development. Once again, this paper is entirely appropriate and is part of a scientific approach which aims to clarify certain concepts actively debated in the PARP community.

Reply: We thank the Reviewer for the kind words, and we are delighted that the Reviewer finds our manuscript appropriate for publication in Nature Communications.

The development of its experimental approach coupled with Macro-AF1521/PARG and ETD fragmentation was at the origin of a major change of course concerning the type of amino acid targeted by the activity of PARP-1. Dr. Nielsen's group published a series of papers which attempted to demonstrate the absence of experimental bias of their method with respect to the type of amino acid acceptor of the modification, using both biochemical and bioinformatic approaches. There is no doubt that the results presented in the present study are of very high quality, but a fundamental problem remains : an alternative MS-based approach developed by the group of Yonghao Yu shows that glutamate and aspartate ADP-ribosylation is a very significant modification following PARP-1 activation by genotoxic insults. Their study does not cast doubt on the existence of serine ADP-ribosylation as ASP/GLU/SER ADP-ribosylation is likely to co-occur in cells.

Reply: We are grateful that the Reviewer acknowledges the high quality of our results, and our efforts towards elucidating exactly which amino acids are modified by ADPr in an unbiased manner. To this end, we would like to point out that the Af1521 methodology we devised inherently does not exclude any type of ADP-ribosylated amino acid residue from detection. To demonstrate this in the initial manuscript, in addition to the Af1521 approach, we also included an antibody-based approach. In the revised manuscript we have added additional experiments to further this (*vide infra*).

We agree with the Reviewer that the work by Prof. Yonghao Yu does in principle not exclude the existence of serine ADPr, however, it should be noted that by design their method exclusively detects

ADPr on glutamate and aspartate residues. Thus, while they may not cast doubt on the existence of serine ADPr, their method also has no way to detect (or fail to detect) it.

The problem is that the Macro-AF1521/PARG/ETD approach is basically excluding the existence of ASP/GLU ADP-ribosylation, a conclusion that goes against the work of Yu's group and others. Nielsen's group did a fantastic exploratory job to clear up this inconsistency but a final and conclusive demonstration is still awaited by the PARP community. If we cannot be sure that D/E ADP-ribosylation can be detected by the current Macro-AF1521/PARG/ETD approach, how can we be sure that ADP-ribosylation is mostly limited to SER-ADP-ribosylation and how can we conclude anything regarding a switch from ASP/GLU to SER ADP-ribosylation?

Reply: While we have not detected a large number of aspartate and glutamate ADPr modifications using our methodology, we do detect a few in the current manuscript and similarly in our previous proteomics analyses (Hendriks et al., 2019; Larsen et al., 2018). For example, our ETD screens identified E488-ADPr and E491-ADPr on PARP1, which are some of the most abundantly modified sites detected by the Yu lab. This demonstrates that our MS methodology is fully able to detect and pinpoint ADP-ribosylation on acidic residues. However, the extent of glutamate and aspartate ADPr we observe is far from what is achieved using the methodology developed by Prof. Yonghao Yu. We believe that the observations made in our revised manuscript provide some clarification of this discrepancy.

Although the use of ETD fragmentation has been shown to be better than CID fragmentation for site-specific assignment of particularly labile post-translational modifications, doubts remain as to its effectiveness in locating ASP/GLU ADP-ribosylation.

Reply: To the best of our knowledge, we are not aware of any reduced effectiveness in ETD fragmentation being able to localize ADPr to aspartate and glutamate residues (or any other PTM for that matter). Nor are we aware of any published research demonstrating this effect. In fact, ETD is a non-ergodic fragmentation technique capable of elucidating the amino acid localization of labile PTMs, including histidine phosphorylation, O-glycosylation, sulfonation, S-nitrosylation and others (Mikesh et al., 2006). Furthering this, S-nitrosylation is so labile in cells that the community is questioning whether it constitutes a ubiquitous PTM at all (Wolhuter et al., 2018). Despite this, ETD fragmentation is fully able to pinpoint S-nitrosylation amino acid acceptor sites, and similarly ETD is able to localize ADP-ribosylation on acidic residues as already demonstrated in our current and previous MS datasets.

Nonetheless, to ascertain that ETD-based detection of ADPr-modified peptides is not biased towards modification occurring on serine residues compared to any other residue types, we have now included an additional proteomics experiment based on *in vitro* ADP-ribosylation. In this new experiment (new Figure 1 in the revised manuscript), recombinant PARP1, PARP10, or PARP14 enzymes were incubated with NAD⁺. Additionally, the PARP1 reaction was performed in the presence or absence of HPF1 and/or H1.0 (a known PARP1 target). Subsequently we trypsinized the reactions, split them in half, and either mock-treated or PARG-treated the resulting peptides. Finally, we analyzed the samples using ETD fragmentation.

We found that in the absence of HPF1, ADP-ribosylation appears to be exclusively directed towards glutamic acid residues *in vitro* (Fig. 1B-C in the revised manuscript). In this analysis, we obtained very clear fragmentation spectra of glutamic acid ADP-ribosylation, using ETD fragmentation, thereby demonstrating that ETD is perfectly capable of detecting this type of peptide if it is sufficiently present in samples (Fig. 1D). Notably, glutamic acid ADPr modification was only observed when the reactions were also incubated with PARG, suggesting that ADPr chain formation seems to be favored under these conditions (Fig. 1C). When HPF1 was present in the *in vitro* reaction, we observed a dramatic shift from exclusive glutamic acid to exclusive serine modification. Collectively, the observed HPF1-dependency is in agreement with previous reports (Bonfiglio et al., 2017; Palazzo et al., 2018).

In case of the PARP10 and PARP14 *in vitro* reactions, we observed a mixture of various reactive residue types to be modified, including predominant arginine and lysine ADP-ribosylation for PARP10 and PARP14, respectively (Fig. 1B). We did not observe any serine ADPr in the PARP10 and PARP14 reactions, whereas some aspartate and glutamate ADPr was found. Notably, these *in vitro* reactions did not contain any potential co-factors, so the observed *in vitro* reactivity might differ from what would be observed *in vivo*, as is the case for PARP1.

Overall, these additional *in vitro* experiments demonstrate that ETD fragmentation is not biased in favor of detecting serine ADPr, nor is it biased against detection of glutamate or aspartate ADPr.

As previously mentioned, Nielsen's group used a variety of approaches to demonstrate the absence of bias in their experimental pipeline. These demonstrations are strong and very convincing from a biochemical point of view but the recurring problem is that it's always the same MS technique, developed in their own laboratory, which is used to draw the final conclusions. Since alternative

approaches exist and directly conflict with their method, it is difficult to understand why the demonstration of the absence of experimental bias does not involve a parallel confrontation with the boronate/hydroxylamine versus Macro-AF1521/PARG approaches.

Reply: We are delighted that the Reviewer finds our approaches and associated findings to be strong and convincing. With regard to using the same MS technique, this is true as far as detection of modified peptides using ETD-driven fragmentation, although we have employed various different fragmentation techniques in our prior studies. For example, we have previously performed a detailed comparison to HCD fragmentation (Larsen et al., 2018), where we determined that HCD fragmentation cannot faithfully localize the full ADPr moiety to any location in the peptide. We have also investigated the application of AI-ETD fragmentation (Buch-Larsen et al., 2020), which further enhanced the ability to localize ADPr in an unbiased manner. Otherwise, in the current manuscript we utilized both the established Af1521 methodology, and a novel antibody-based enrichment method using two unique antibodies. All three of these approaches allow for an unbiased readout using ETD fragmentation, where ADPr is allowed to be assigned to nine previously reported amino acid residues. Contrary to this, and the primary reason we do not use the boronate/hydroxylamine methodology, is that this method by design can only detect ADPr on glutamate and aspartate residues. Thus, it is a biased methodology that cannot address ADPr modification on any other residue type, and therefore can only provide a limited vision.

On several occasions, Nielsen's group discusses the possibility that the derivatization of ASP/GLU ADP-ribosylation is only an experimental artifact. The problem with this claim is that it doesn't hold up if you look at the protein profile isolated by the boronate/hydroxylamine approach and the quantitative profiling of ASP/GLU ADP-ribosylation in the presence of specific PARP-1 inhibitors. Boronate affinity purification has been broadly used for decades for the selective isolation and enrichment of cis-diol-containing molecules such as PAR. Technically, boronate-diol complexation is not dependent on the ADP-ribosylation amino acid linkage and therefore can be used for unbiased isolation of all types of ADP-ribosylated peptides.

Reply: In our manuscript, we raise the significant probability that treatment of samples with hydroxylamine can chemically induce hydroxamic acid modification on unrelated aspartate and glutamate residues, which in some cases could lead to erroneous assignment of ADPr to such residues. The overarching challenge with using hydroxylamine is that the chemical artefact induced is identical to the derivatization mark used for identification of ADP-ribosylation acceptor sites. As a result, mass spectrometry is unable to ascertain which mass adducts on acidic residues are due to

chemical artefacts, and which constitute true biological ADP-ribosylation sites. Still, the Reviewer raises a valid point, and we apologize for not having explained this conundrum in more detail in our initial manuscript. We fully understand the thought of the Reviewer, and would like to emphasize that the boronate/hydroxylamine approach consist of a two-step process; 1) enrichment of ADP-ribosylated peptides/proteins, and 2) derivatization of ADP-ribosylation sites using hydroxylamine. Notwithstanding, the boronate affinity purification is indeed fully capable of enriching ADPr-modified peptides and proteins, and we do not dispute this in our manuscript. The backbones of the boronic acid-enriched peptides can be successfully sequenced by MS, and afterwards the correct proteins will be inferred from these sequences. Furthering this, we previously demonstrated that at the protein target level our data (Larsen et al., 2018) overlaps very well with the data from the Yu lab (Zhang et al., 2013).

However – beyond identifying the correct peptide sequence, we provide evidence that the second step in the boronate/hydroxylamine approach; the assignment of ADPr to an aspartate or glutamate within these peptides might be erroneous in certain cases because of the ability of hydroxylamine to induce non-specific hydroxamic acid modification of aspartate/glutamate residues. We have more clearly elaborated this in the discussion of the revised manuscript, where we have provided further explanation of our observations. We have also moved the hydroxylamine-treated BSA data from a supplementary note to a supplementary figure, to better discuss the points raised by the Reviewer.

In our original manuscript, we demonstrated the occurrence of hydroxylamine-induced chemical artefacts on bovine serum albumin (BSA). We now additionally reprocessed publically available data from the Yu lab where ADP-ribosylation in breast cancer cell lines was profiled using the boronate/hydroxylamine approach (Zhen et al., 2017). Here (previously Supplemental Note 1, now Figure S2, updated panels C-E-G-I), we investigated the prevalence of hydroxamic acid modifications on aspartate and glutamate residues (which is presumed to correlate with ADP-ribosylation) in addition to transamidation on asparagine and glutamine residues (which correlates with the chemical artefact we proposed). Overall, we find a larger fraction of hydroxamic acid modification in the breast cancer cell line samples compared to HA-treated BSA, and a smaller fraction of transamidation in the breast cancer cell line samples both suggesting an elevated presence of 'true' ADPr-modified peptides. Nonetheless, we did observe a significant occurrence of asparagine and glutamine transamidation in the breast cancer cell line samples (compared to untreated BSA control), indicating that at least a part of the observed aspartate and glutamate hydroxamic acid modification could be chemically induced rather than biologically. We would also like to bring to the Reviewer's attention that the non-specific reaction between hydroxylamine and

(non-ADP-ribosylated) glutamic and aspartic residues constitutes one of the most important reactions in organic chemistry; the amide bond formation (Pattabiraman and Bode, 2011). We would also like to mention, that due to the experimental design of the boronate/hydroxylamine approach, it is not feasible to decipher the true extent of this chemical artefact in a straightforward manner.

I think the only way to truly establish one approach against another would be to take the protocol of each method and subject it to the two alternative methods of derivatization.

Reply: We agree with the Reviewer, in the sense that the specificity of the boronate affinity purification should be compared to our Af1521 (and antibody) methodology. However, we respectfully disagree with the notion of comparing alternative methods of derivatization. We believe the most unbiased way is to detect the full intact ADPr moiety modifying an amino acid residue (as achieved using the Af1521 and antibody enrichment strategies) – rather than substituting the ADPr moiety on peptides with a chemical mark (as in the hydroxylamine approach). Especially if such derivatization, as in the case of hydroxylamine treatment, precludes the vast majority of amino acid residues from being detectable as ADP-ribosylated, and can induce chemical artefacts that mimic derivatized ADP-ribosylation sites.

Still, in our revised manuscript, we have included a boronate affinity purification of ADP-ribosylated peptide from untreated and H₂O₂-treated HeLa cells. However, rather than derivatizing the peptides using hydroxylamine, we used a previously reported strategy of eluting the peptides with acid (Haag and Buck, 2015), thereby retaining the complete ADPr moiety. We then subjected these peptides to MS detection via ETD, essentially as done for our other approaches.

Overall, we successfully enriched ADPr using the boronate affinity purification followed by acid elution (new Figure 3), although the peptide purity we observed was only around ~5%, which is substantially lower than our Af1521-based enrichment (>90%) but comparable to our antibody-based enrichments (~5-20%). Intriguingly, in the samples enriched using boronic acid, we still mainly detect serine residues to be modified, demonstrating that neither the Af1521- or antibody-based enrichments are biased. Furthermore, the ADPr-modified proteins we identify overlap highly significantly with other ADPr proteomics studies. Thus, when considering the consistency of the three distinct purification methods we utilized in this manuscript, along with the ability of ETD to unbiasedly identify ADPr on every amino acid type (new Figure 1), we believe these data demonstrate that the reduced presence of glutamate and aspartate ADPr in our analyses accurately represents cell biology.

As to why other labs primarily observe ADP-ribosylation on glutamic and aspartic acids using boronic acid enrichment; this strongly relates to the use of hydroxylamine.

Specific comments

In the first part of their manuscript, the authors performed a series of in vitro biochemical assays to evaluate if their Macro-AF1521 enrichment strategy is unbiased with regard to identification of amino acid acceptor sites. This is absolutely critical to draw any conclusions about the regulatory landscape of the human HPF1- and ARH3-dependent ADP-ribosylome. The strategy developed by the group of ML Nielsen in the last years is based on the use of PARG to convert heterogeneous PAR polymers into mono-ADP-ribose remnants with a unique +541 Da mass increment signature. In the absence of PAR hydrolysis, PARylated peptides cannot be analyzed by mass spectrometry because of their complex chemical structures. In this study, the authors show that the number of SER-ADP-ribosylation sites is basically the same whether PARG is used or not to convert PAR into MAR signatures in cell extracts in which PARP-1 was hyperactivated. The authors therefore come to the conclusion that mono-ADP-ribosylation is predominant in human cells. This observation also implies that the ADP-ribosylation landscape is not fundamentally modified following PARP-1 activation. This is a shocking statement for many reasons. I will get back to that.

Reply: The Reviewer correctly states that we observe that the majority of ADPr exists as mono-ADPr within HeLa cells, under the conditions we investigated. However, we would like to stress that we do not at all imply that the ADPr landscape is not fundamentally modified following PARP1 activation. In fact, in response to H₂O₂ treatment (which we have previously demonstrated to activate PARP1), we observe a dramatic increase in the number of ADPr sites. Similarly, when treating cells with the PARP inhibitor Olaparib, we observe a complete loss of ADP-ribosylation in our analyses (Larsen et al., 2018). Hence, the regulated ADP-ribosylation observed in the current study is likely due to strong PARP1 activity. Whereas H₂O₂ treatment could increase the amount of poly-ADPr, from our observations the overall equilibrium remains predominantly weighted towards mono-ADPr. Moreover, we would like to point out that our observations align well with a recently published paper demonstrating the prevalence of mono-ADPr in comparable cell systems (Bonfiglio et al., 2020). Additionally, we would like to note that our observations are a snapshot of the cells, which we capture upon lysis, and only cover the conditions we investigated (i.e. 10 min H₂O₂ treatment). It is possible that long poly-ADPr chains are rapidly generated upon PARP1 activation (Figure S1, new panel C), but these could be highly transient and consequently difficult to detect because of high eraser enzyme activity. We have emphasized this aspect in more detail in the revised manuscript.

The authors used two anti-MAR/PAR antibodies as complementary approaches to isolate ADP-ribosylated peptides to ensure that their MS-based proteomics strategy using Macro-AF1521 enrichment preserves ADP-ribosylation. The authors mentioned that a residual MAR hydrolase activity of Macro-AF1521 could have been responsible for a bias towards certain types of ADP-ribosylation linkages. Since anti-MAR/PAR antibodies do not possess MAR hydrolase activity and that SER-ADP-ribosylation was the only significant modification found, the authors concluded that their approach is unbiased. There are a number of problems with this conclusion. In theory, if anti-MAR/PAR antibodies are targeting the same MAR/PAR modifications as Macro-AF1521, one would expect to identify a similar number of ADP-ribosylation sites. However, the profiles are very different with AB1/AB2 versus Macro-AF1521 in Fig-1B.

Reply: Indeed, we investigated the antibodies to serve as an additional technical control for our Af1521 methodology. Further, we had hoped that the antibody-based enrichment would be more competitive with Af1521 for MS-based proteomics identification of ADPr-modified peptides.

There are several technical reasons that can explain why the AB1/AB2 profiles are very different from the Af1521 profile:

1) Prior to the large-scale Af1521 versus antibody comparison we included in this manuscript, we performed a small-scale pilot experiment using three different antibodies (see figure below).

In this pilot experiment, we found that two antibodies identified ~60% of the number of sites compared to Af1521, with ADPr primarily (~85%) targeting serine residues across all of these experiments.

Following this, we scaled up the experiment approximately 20-fold in order to increase sequencing depth and to profile a larger number of ADPr sites. For inclusion in the initial manuscript, we only included antibodies labeled “AB1” and “AB3” as they performed the best. Contrary to the antibodies that are only commercially available, we can easily produce milligrams of Af1521 in-house, making it highly feasible for scaling up experiments. However, we could not similarly scale up the amount of antibody used, due to a limited supply. Thus, considering the purification was performed with milligrams of Af1521 on beads versus micrograms of antibody on beads, we reason that the limited number of ADPr sites identified using the antibody-based approach is mainly because of a lack of binding capacity.

2) We observed a considerably lower amount of total ion current generated from the AB samples compared to the Af1521 samples; approximately 5-10 times less. This corroborates what we outlined above; an overall lower yield from the AB-based purification. Nonetheless, the fraction of MS/MS spectra identified from the AB samples was 35.9% for AB1 and 39.0% for AB2, compared to 22.4% for Af1521. Thus, ETD fragmentation was quite successful at identifying the contents of the AB samples, arguing against the possibility of a higher presence of ETD-unidentifiable ADPr linkages in the AB samples.

3) The sample purity of AB samples is considerably lower than Af1521 samples (new Figure 3B), which means that there are many more unmodified peptides within the AB samples. The relatively high presence of unmodified peptides can obfuscate detection of ADPr-modified peptides, as selection of precursors by the mass spectrometer is a stochastic process. In turn, this can partially explain why lower numbers of ADPr sites were identified from the AB samples.

In conclusion, the comparison between antibodies and Af1521 was included to demonstrate that Af1521 is not biased towards serine ADPr which we believe our data fully demonstrates. The actual number of sites identified across the different strategies is therefore not very relevant for the comparison. Moreover, in the revised manuscript we have included additional experiments using boronic-acid enrichment of ADP-ribosylated peptides. Here we also find serine to be the major ADPr acceptor site, supporting our findings from our Af1521 and antibody-based experiments and demonstrating that both of the outlined methodologies are unbiased.

Is it very common for anti-PAR antibodies to have much more affinity for long and complex PAR than MAR. The affinity for PAR by an anti-MAR/PAR antibody is likely introducing a bias for PARylation versus MARylation.

Reply: The Reviewer is correct, and we have observed the same phenomenon for the antibodies we used in the context of western blot analyses. To a certain degree, this selectivity may also be present during enrichment of peptides for MS analysis, especially because we used a limiting amount of antibody (see #1 above). Indeed, this is visible in the manuscript in (previously Figure 1, now Figure 2, panel B) where there is a slight increase in the number of sites identified using the AB-based enrichment after PARG treatment of the peptides. In case samples were not first PARG-treated, the antibodies might preferably enrich ADPr chains, which are then unidentifiable by the MS (although these would only be a very small fraction of the total sample). In case samples were PARG-treated, the selectivity for PAR would no longer play a role, and the antibodies can then exert their full binding capacity towards MAR, resulting in a small increase in the total number of ADPr sites identified.

We believe that such extensive technical discussion goes beyond the scope of the manuscript, especially considering the numerous technical variables involved. Nonetheless, we have now clarified in the discussion that the amount of antibody used was limiting.

Again, we would like to emphasize the observations made in our boronic-acid-based enrichment experiment. While boronic-acid beads are more likely to bind PAR than MAR, we still primarily observe serine ADP-ribosylation as the major acceptor site in our data. Which fully supports our initial observations made using our other enrichment strategies.

The main concern regarding the bias for SER-ADP-ribosylation do not come from the potential selective MAR hydrolysis activity of Macro-AF1521 but the capacity of the ETD fragmentation approach (or the selected experimental strategy) to identify ASP/GLU ADP-ribosylation.

Reply: As outlined above we have now demonstrated, using *in vitro* essays, that ETD is fully capable of detecting ADPr on glutamate and aspartate (new Figure 1).

If PARylation occurs on D/E residues and that this specific linkage is unstable under the authors experimental conditions, we would expect to see a decrease in the number of SER-ADP-ribosylation

sites following PARP-1 activation when peptides are isolated with antibodies. This is what is reported in Fig-1B.

Reply: We have explained above why (for technical reasons) we see a decreased number of identified ADPr sites in the AB samples.

If the peptide fragmentation strategy is biased for site-specific localization of SER-ADP-ribosylation, we would also expect that any other type of ADP-ribosylation chemical linkages, including D/E ADP-ribosylation, would be insignificant when analyzing the total number of ADP-ribosylation sites. So, even if PARG is used to generate D/E MARylated peptides in complex samples, these peptides won't be identified and SER-ADPr-ribosylation will appear unchanged with or without PARG treatment, as can be observed in Fig-1B.

Reply: We have now demonstrated, using *in vitro* assays, that ETD is fully capable of detecting ADPr on glutamate and aspartate, and is not biased in favor of detecting ADPr on serine residues (new Figure 1).

The shocking proclamation that MARylation predominates in cells, even after extensive PARP-1 activation with hydrogen peroxide needs more evidence. Obviously, all of these assumptions hold true to the extent that we consider that there is indeed a bias for SER-ADP-ribosylation.

Reply: Our observation of a predominant amount of serine MAR does not preclude the existence of glutamate/aspartate PAR. However, glutamate/aspartate ADPr may be more transient than other types of ADPr. They could be rapidly reversed by eraser enzymes, or rapidly hydrolyzed, and would thus require specialized experimental design (such as perturbation of eraser enzymes) to be more readily visualized. Further, it is possible that glutamate/aspartate PAR signaling could be >50-fold lower abundant compared to the serine MAR signaling, at least in the context of MS identification of the modified amino acid residues within proteins. It is entirely feasible that although fewer glutamate/aspartate residues are modified, these could harbor very long chains *in vivo*, which would in turn still be able to regulate biological signaling. We have now expanded on this in the discussion of the revised manuscript.

Finally, our observation of predominant MAR is corroborated by a recent study published in Cell (Bonfiglio et al., 2020).

In contrast to what is cited in the manuscript and in the supplementary note, it has been demonstrated that hydroxylamine do not react nonspecifically with free aspartic acid and glutamic acid residues (at least in the experimental conditions described by the group of Yonghao Yu in Nature Methods, 2013).

Reply: We are aware of the claim that hydroxylamine does not cause chemical artefacts (Zhang et al., 2013).

However, using similar experimental conditions and in the context of BSA, we do find a significant artificial hydroxamic acid modification. Additionally, there is evidence in the literature that hydroxylamine treatment causes this artificial modification, for example in the context of TMT experiments (Geiszler et al., 2020). Furthermore, in the revised manuscript (previously supplemental note, now Figure S2, updated panels C-E-G-I) we reprocessed data from the Yu lab (Zhen et al., 2017), and find a significant presence of transamidation on asparagine and glutamine, which is a chemical co-artefact caused by use of hydroxylamine. The existence of transamidation of asparagine and glutamine residues therefore serves as a positive control for the existence of similar chemical artefacts on glutamic and aspartic acid residues. Hence, our observations provide strong evidence that usage of hydroxylamine in proteomics-based analysis may not be analytically suitable for studying ADP-ribosylation, as it remains impossible to discriminate true derivatized ADP-ribosylation sites on glutamic/aspartic acid residues from the chemical artefacts introduced by hydroxylamine.

We are not fully aware of all experimental parameters affecting the control experiment as performed by the Yu lab (Zhang et al., 2013). However, we reason that the extremely high number of proteoforms in a total lysate (which they used as their control system) may render detection of hydroxamic acid modifications far more challenging, compared to detection of these modifications in a much simpler sample (such as BSA, or ADPr samples enriched via boronate-affinity). This relates to the stochastic nature of precursor selection for MS/MS analysis, and the very high abundance of non-related peptides in total proteome samples. The inherent challenge of detecting modified peptides from a total lysate is further compounded by the fact that they used a Lys-C digest (which generates much longer and thus more difficult-to-identify peptides).

Overall, we believe that the control experiment provided by the Yu lab is not exhaustive, and that we provide sufficient counter-evidence, which is moreover supported by additional literature (Geiszler et al., 2020).

The authors should use the m-aminophenyl boronate affinity-purification of PARylated peptides developed by Yu's group coupled to their MS-based approach to see if there is really a bias for specific ADP-ribosylation sites. In theory, they could be able to easily identify D/E ADP-ribosylated peptides with ETD fragmentation.

Reply: Following the Reviewer's suggestion, the revised manuscript now includes a boronate affinity-purification, followed by acid elution, and ETD analysis of the intact ADPr moiety. We primarily observe serine ADPr in these samples (new Figure 3), even though ETD is fully capable of detecting D/E ADPr (new Figure 1).

The intensity of basically all D/E ADP-ribosylated peptides is strongly decreased after treatment with a PARP inhibitor (even reaches a 90-99% decrease for PARP-1 automodification sites). If hydroxylamine derivatization was just an artifact, such PARPi-specific modulation of D/E ADP-ribosylation would not occur. If D/E ADP-ribosylation is not detectable using Yu's approach coupled to ETD that could indicate a bias for non ASP/GLU linkages, such as SER-ADP-ribosylation. Conversely, hydroxylamine could be used in peptide extracts isolated with Macro-AF1521 versus m-aminophenylboronate for bias analysis. I think that the validation of the unbiased approach described in the present study by Hendriks IA and colleagues should be based on MS rather than on biochemical assays targeting selective hydrolysis of specific ADP-ribosylation linkages.

Reply: As outlined above, we have now demonstrated that:

- 1) ETD is fully capable of detecting exclusive glutamate ADPr in PARP1 *in vitro* reactions (lacking HPF1).
- 2) ETD is fully capable of detecting glutamate/aspartate ADPr in PARP10/PARP14 *in vitro* reactions.
- 3) Using boronate affinity purification followed by ETD detection, we only observe a low amount of glutamate/aspartate ADPr.

Rather than our approach being biased, we believe our results represent cell biology. Further, our observations do not preclude that aspartate/glutamate ADPr exists, albeit at ~50-1,000 fold lower abundance compared to serine ADPr (in the experimental context we investigated). Notably, our *in vitro* and *in vivo* analysis provides evidence for such abundance difference (Fig. 1B & Fig. 5C).

Whereas we point out that hydroxylamine treatment can induce artificial hydroxamic acid modification of aspartate and glutamate residues, we did not claim that all ADPr sites identified by the Yu lab are artificial. In our revised manuscript, we have included reprocessed data from the Yu lab which suggests that a part of their hydroxamic acid residues could correspond to displaced ADPr, and we have updated the discussion.

Additionally, we would like to point out that inherently it is not possible with the hydroxylamine strategy to discriminate which glutamic/aspartic acids constitute true derivatized ADP-ribosylation sites, and which could be chemically induced.

The authors mentioned in the manuscript that treatment of cell lysates with hydroxylamine did not change the distribution of ADP-ribosylation, suggesting that ASP/GLU ADP-ribosylation is not significant (Fig. S2C). The conditions used to hydrolyze PAR with hydroxylamine in cell extracts is not described in the manuscript. Automodified PARP-1 has been shown to be extremely sensitive to hydroxylamine. A control is missing when using hydroxylamine for Western blot analysis of residual PARylation. Lysis buffer supplemented with automodified PARP-1 treated with/without hydroxylamine should be used to visualize the efficiency of the reaction.

Reply: We apologize for not detailing the hydroxylamine treatment of cell lysates in the methods. We used the same treatment procedure as described previously (Gagné et al., 2015; Palazzo et al., 2018), which entails treatment using 1 M hydroxylamine for 3 hours, either in TRIS or HEPES buffer at pH 8.0. We have now validated (see figure below), using auto-modified full-length PARP1, that this treatment is highly efficient at removing the auto-modification of PARP1.

We concur with the Reviewer (*vide supra*), that since our study is primarily based on MS, that additional validation using biochemical assays does not necessarily add a lot of value. Further, various labs report different methodologies for hydroxylamine treatment, and use a wide range of lysis buffers and procedures (Gagné et al., 2015; Moss et al., 1983; Palazzo et al., 2018; Rodriguez et al., 2021). Moreover, we believe that hydroxylamine treatment of lysates does not necessarily give as detailed of an insight as MS data, hydroxylamine treatment may reverse ADPr not on glutamate and aspartate residues (Hsia et al., 1985), and the exact treatment conditions during hydroxylamine experiments may skew the data. For these reasons, and following the recommendation of the Reviewer, we have decided to omit the hydroxylamine treatment western blot (previously supplementary Fig. S2C) from the manuscript.

For protein digestion, the authors used a two-step approach. Proteins are first digested with LysC in 6M guanidine hydrochloride buffer, followed by trypsin after a dilution of the extract in ammonium bicarbonate to lower guanidine concentration to a level compatible with trypsin activity. This is a typical procedure that is normally very effective to digest proteins to completion (especially a two-step method with a strong chaotropic/denaturing agent such as guanidine hydrochloride).

Reply: The digestion strategy we utilized was identical to what we have previously described (Buch-Larsen et al., 2020; Hendriks et al., 2019; Larsen et al., 2018). This is indeed a very effective digestion strategy and usually the final average missed cleavage rate is ~0.5 when considering the total lysate digest and read-out of the peptide mixture via HCD fragmentation.

However, peptides with a surprisingly high number of missed cleavages are identified with site-specific ADP-ribosylation sites (i.e. on serine for the vast majority of peptides). Systematically, very large peptides with an unusual high number of missed cleavages are listed (e.g. Supplementary Table-1). Under these conditions, we can practically speak of partial trypsin/LysC digestion. However, the digestion conditions described in the Materials section is not presented as such.

Reply: Overall, the average number of missed cleavages we observe in this study is 0.67 for unmodified peptides, which indicates a highly efficient digestion. The average number of missed cleavage for ADPr-modified peptides is 1.66, or about ~1 missed cleavage higher compared to the unmodified peptides. Average charge of the peptides was 3.17 for both unmodified and ADPr-modified peptides, i.e. there was no discrepancy at the charge state level. We have previously observed the exact same phenomena in all our ADPr studies (Buch-Larsen et al., 2020; Hendriks et al., 2019; Larsen et al., 2018).

We reason that the presence of an intact ADPr moiety incurs a negative charge upon the peptide, which must be compensated for by another charge-bearing residue type (such as lysine or arginine) in order for the peptide to be resolvable via MS/MS analysis.

Furthermore, this could also indicate that ADPr preferentially resides in protein regions that are very rich in arginine or lysine. Indeed, when considering ADPr-modified peptides in histone proteins, the average missed cleavage number rises to 2.23. When excluding histones, the average number of missed cleavages for ADPr-modified peptides drops to 1.32.

Finally, we previously observed that there is also a discrepancy between HCD and ETD in terms of peptide resolvability (Larsen et al., 2018). There, we compared HCD versus ETD for their applicability in detecting and localizing the intact ADPr moiety on peptides. Beyond the fact that HCD was dramatically worse in being able to localize ADPr, we also observed the following:

	PSMs	Mis.cl.	Charge	Mass	Length
HCD-all	6,431	1.21	3.28	2,104	17.14
ETD-all	2,686	1.83	3.36	1,801	12.98
Both-all	6,696	1.25	3.23	1,801	14.35
HCD-AR	1,580	1.88	3.51	2,665	19.73
ETD-AR	1,585	2.3	3.45	1,958	12.82
Both-AR	1,648	1.99	3.33	2,161	15.02

“Both” refers to PSMs identified by both ETD and HCD, and “mis.cl” to average missed cleavage count. Charge, mass, and length all depict the average properties of all identified peptides. “All” refers to all peptides identified, and “AR” to ADP-ribosylated peptides. We observed that peptides preferentially identified by ETD contained more missed cleavages, i.e. more lysines and arginines, although the overall charge state in the identified peptides was not notably different between ETD and HCD. We also observed a considerable difference in peptide size and mass, with HCD identifying larger peptides compared to ETD. However, these observed peptide property trends (missed cleavages, charge, mass, length) held true for ETD and HCD when considering either unmodified peptides or modified ADP-ribosylated peptides, demonstrating fundamental differences in the technologies.

Overall, we conclude that digestion was performed successfully, and that the elevated number of missed cleavages observed for ADPr-modified peptides are due to a mixture of technical reasons as well as the preference for ADPr to target lysine-rich proteins (such as histones).

For example, the peptide AEPVEVVAPRGKSGAALSKKS(ADPr)KGQVK (5 missed cleavages) and several other similar peptides of varying length are almost always missed cleaved after the arginine.

Reply: Within the primary MS/MS dataset of this study, and disregarding protein C-terminal peptides, we observed:

65,097 total peptides ending with a K (~59.8%)

43,759 total peptides ending with an R (~40.2%)

20,589 ADPr-modified peptides ending with a K (~67.4%)

9,980 ADPr-modified peptides ending with an R (~32.6%)

The prevalence of K residues in the total proteome is 7.2%

The prevalence of R residues in the total proteome is 4.2%

Thus, one would expect peptides to end with a lysine residue ~63.2% of the time, and end with an arginine residue ~36.8% of the time. The distribution of ADP-ribosylated peptides ending with a K or R in our dataset therefore follows what would be expected.

However, the peptide AEPVEVVAPR is consistently identified in complex mixtures of PARylated peptides. This is a very efficient cleavage site that generates a typical PARP-1 peptide in basically all MS-based studies. The authors should explain why such a high number of missed cleavages are observed in their data sets. Trypsin is known to be sensitive to post-translationally modified amino acids located next to the K/R cleavage site so we would expect the KS(ADPr) to be skipped but not at unmodified cleavage sites to such extent.

Reply: As outlined above, we conclude that digestion was performed successfully, and that the elevated number of missed cleavages observed for ADPr-modified peptides are due to a mixture of technical reasons as well as the preference for ADPr to target lysine-rich proteins (such as histones).

Following the same idea, a large number of peptides starting with S(ADPr) in N-terminus are listed in Supplementary Table-1 (>400 peptides) suggesting that trypsin cuts at K/R residues that precede major SER ADP-ribosylation sites. This is surprising because, for example, K/R residues are often not cleaved by trypsin when a phosphorylated serine is located adjacent or in close proximity. Considering the charge density of PAR and its steric hindrance, efficient cleavage of KS(ADPr) sites is unexpected. This should be discussed because this unusual peptide features are not found with other MS-based techniques used to localize site-specific ADP-ribosylation sites.

Reply: We agree that some steric hindrance from a bulky PTM such as ADP-ribosylation could limit proteolytic cleavage, and may result in somewhat elevated occurrence of missed cleavages. Further, PTM-modified residues themselves are often not good substrates for proteolytic enzymes, especially in the case of bulky PTMs. This is also why we do not allow assignment of ADPr to the C-terminus of any peptide (unless it is protein C-terminal), as modification of the K/R would have prevented cleavage by trypsin and Lys-C.

However, most common enzymes with C-terminal cleavage specificity (such as trypsin and Lys-C) are able to cleave just N-terminal of PTM-modified residues, albeit potentially at a lower efficiency.

Similarly, enzymes with N-terminal cleavage specificity (such as Asp-N) are able to cleave just C-terminal of bulky-PTM-modified residues, for example as described for SUMOylation (Hendriks et al., 2018).

Thus, because of the high prevalence of ADPr occurring on serine residues in KS motifs, and despite a potentially reduced activity of proteolytic enzymes in the proximity of ADPr, we reason that the observation of many peptides starting with serine-ADPr falls entirely within expectations.

Peptides with numerous missed cleavages are generally more hydrophobic and have a higher overall charge. Can it introduce a bias in the present MS analysis pipeline?

Reply: As outlined above, we do not observe a significant deviation between charge state of unmodified and ADPr-modified peptides within the samples we have analyzed. The overall elevated charge state of these peptides is beneficial for ETD analysis, which excels at analysis of peptides with charge state of three or higher. Therefore, we do not believe there is any bias within our MS analysis pipeline.

In the last part of the manuscript, the authors analyzed the co-occurrence of other histone PTMs with ADP-ribosylation. The authors did not mention that ASP/GLU ADP-ribosylation of histones was reported by the group of BA Garcia (Mol Biosyst, 2017). This group used a quantitative approach to show that the abundance of ASP/GLU ADP-ribosylation sites increases in abundance over time after PARP-1 activation. They also showed that histone H1 is a major ADP-ribosylation target. This study cannot be ignored and should be discussed. Garcia BA and colleagues also report that treatment with hydroxylamine demonstrated a near complete depletion of signal, indicating that a vast majority of histone ADP-ribosylation sites are occurring on ASP/GLU residues.

Reply: In the paper from the Garcia lab, ADP-ribosylation of histones was performed *in vitro* in the absence of HPF1, which could lead to preferential modification on glutamate and aspartate residues (similarly to what we described in Figure 1 of the revised manuscript). In their *in vivo* experiment, the authors used biotinylated NAD⁺ to allow visualization of ADP-ribosylation, which was then shown to be sensitive to hydroxylamine treatment. However, in the mentioned manuscript from the Garcia lab, they did not show endogenous and native ADPr, and therefore it is hard to conclude whether the phenomenon they observe may be specific to the biotinylated NAD⁺ analogs. Most importantly, they used hydroxylamine derivatization in order to identify ADPr sites via MS, and therefore precluded any non-E/D ADPr from being detectable (*vide supra*).

We do agree with the Reviewer that this paper is important to discuss, and we have now included this in the discussion.

To ensure that the hydroxylamine is not chemically modifying histone proteins, they performed a control experiment where they incubated recombinant histone proteins with hydroxylamine overnight. They did not identify any derivitized sites, indicating that the hydroxylamine does not chemically modify ASP/GLU residues on histones.

Reply: To cite the paper from the Garcia lab: "*We did not identify any derivitized sites, indicating that the hydroxylamine does not chemically modify Asp and Glu residues on histones (data not shown). Therefore, the hydroxylamine derivitization is a suitable method to comprehensively analyze histone ADP-ribosylation sites.*" Notably, the authors mention data not shown, and therefore we cannot ascertain the validity of this specific claim.

It should be noted that other papers are also reporting the specific ASP/GLU ADP-ribosylation of histone proteins (e.g. site-specific ADP-ribosylation of histone H2B at GLU-18 and GLU-19 in response to DNA double strand breaks, Rakhimova A. et al. Scientific Reports, 2017).

Reply: We would like to thank the Reviewer for bringing this study to our attention. However, we would like to point out that the study by Rakhimova et al. (Figure 3F in their paper) reports that separate mutation of either E18 or E19 does not seem to decrease ADPr, and in case of E19 even increases it. A combined mutation of both of these residues appears to only slightly decrease observed ADPr signal, and the observed signal also does not entirely line up with the presented input control.

As long as a formal demonstration regarding the preferential targeting of serine residues in the ADP-ribosylation process is not fully established, it would be preferable to acknowledge the fact that, perhaps, both ASP/GLU and SER ADP-ribosylation exists in cells.

Reply: We agree with the Reviewer, and we do not preclude glutamate/aspartate ADPr from existing in cells. In the absence of HPF1, our *in vitro* experiments confirm that the majority of modification occurs on glutamate/aspartate. Conversely, we believe that we present strong evidence in our manuscript for preferential serine ADP-ribosylation of histone H1 (90-99%) *in vitro*, which is entirely mediated via HPF1.

There is a very significant overlap in the protein profiles of ADP-ribosylated proteins identified by the boronate/hydroxylamine and Macro-AF1521/PARG approaches. As seen in Fig-S2B of this manuscript, RNA-binding proteins, chromatin regulators and DNA damage response factors are predominant in data sets of proteins identified with the Macro-AF1521/PARG approach, just as it can be observed with the boronate/hydroxylamine approach. A similar over representation of PARP-1 targets has also been shown using a NAD⁺ analog–sensitive approach for PARPs coupled to hydroxylamine derivatization (Gibson BA et al. Science 2016). These converging results provide strong evidence that there is a real relationship between the identification of ADP-ribosylation sites, by both approaches, so that one does not exclude the other.

Reply: We previously observed ADPr on similar proteins when comparing our Af1521 data (Larsen et al., 2018) and boronate/hydroxylamine data (Zhang et al., 2013). In our revised manuscript, we also find a significant overlap to previous studies when considering ADPr-modified proteins in the context of boronate affinity purification followed by acid elution and ETD detection (new Figure 3). Any approach that enriches ADPr-modified proteins, regardless of which residue type is modified, will yield a similar profile of identified proteins (via identification of the target peptide backbones). We believe that the main discrepancy occurs after enrichment of these peptides, where either the entire ADPr moiety is identified (using our methodology), or there is derivatization of ADPr (using hydroxylamine elution) which prohibits detection of ADPr on residue types other than glutamate/aspartate.

We put additional focus on this discrepancy in the discussion, and suggest that aspartate/glutamate ADPr can co-occur with serine ADPr, although aspartate/glutamate ADPr is much less abundant within the cellular systems we have investigated.

Reviewer #2 (Remarks to the Author):

Hendriks and colleagues have successfully developed a quantitative proteomics method to explore the ADP-ribosylated proteome in the presence and absence of the regulatory proteins HPF1 and ARH3. Utilization of this method has revealed a set of ~1,500 ADP-ribose sites that are differentially modified across the investigated knockout cell lines. Importantly, their technique has helped corroborate the roles for ARH3 and HPF1 in the ADP-ribosylation of serine residues. Further, they demonstrate that the addition of PARG used in prior iterations of their technique is not strictly required as most of the identified ADP-ribose sites already exist as mono-ADP-ribose sites prior to exogenous PARG treatment. This is a very useful and interesting article that will boost various proteomic and biochemical research efforts in the field, while highlighting the need for further mechanistic characterization of the ADP-ribose modification pathway.

Reply: We are delighted that the Reviewer appreciates the value of our proteomics efforts towards elucidating the ADP-ribosylome as mediated by HPF1 and ARH3, and recognizes the significance of our finding that ADPr could exist primarily as MAR in the context of oxidative stress.

The following are a few specific comments regarding the manuscript:

1. One of the main arguments that the authors propose from their work is that there is little to no modification of glutamate or aspartate residues in the experimental conditions that they are evaluating. This point would be in contrast to multiple published reports of acidic residue targeting within the PARP family of enzymes. An alternate explanation that might be consistent with their observations is that the removal of ADP-ribose from Ser and Glu/Asp residues are governed by significantly different kinetics. In this alternate explanation, the absence of Glu/Asp modifications in their survey could be explained by a prior, and fast, removal of ADP-ribose before they generate their trypsinized peptide libraries. This removal activity might be due to the endogenous activity of PARG itself or another member of the glycohydrolase family (i.e. TARG or MARG). It might bolster their case if they performed an additional experiment wherein endogenous PARG activity was halted (through KO or inhibitor) prior to lysis and a similar plot to figure S2C was prepared in non-stressed and H₂O₂ treated cells. Such an experiment would not definitively rule out the activity of another Glu/Asp specific eraser, but would provide useful data for their primary argument. Regardless, I think it would be appropriate to engage with this possible alternate explanation within the context of the discussion.

Reply: We fully agree with the Reviewer that the relative absence of glutamate and aspartate ADP-ribosylation in our data is somewhat surprising. To minimize the possibility of technical bias in our observations, as detailed in our response to Reviewer #1, we now demonstrate in the revised manuscript that:

- 1) ETD is fully capable of detecting exclusive glutamate ADPr in PARP1 *in vitro* reactions lacking HPF1 (new Figure 1).
- 2) ETD is fully capable of detecting glutamate/aspartate ADPr in PARP10/PARP14 *in vitro* reactions (new Figure 1).
- 3) Using boronate affinity purification followed by ETD detection, we do not observe a lot of glutamate/aspartate ADPr (new Figure 3).

Indeed, as pointed out by the Reviewer, the dynamics of glutamate and aspartate ADPr could be significantly different from serine (and other types of amino acid acceptor sites that we frequently observe in our datasets) ADPr. Although other labs commonly investigate ADPr after 10 minutes of treatment with hydrogen peroxide, including other proteomics-based analyses, the observations of predominant glutamate and aspartate ADPr by the Yu lab were exclusively made in the context of PARG knockdown/knockout (Zhang et al., 2013; Zhen et al., 2017). Thus, it is feasible that PARG, or other eraser enzymes, might rapidly reverse glutamate and aspartate ADPr in living cells.

Furthering this, we have now performed the experiment as suggested by the Reviewer, wherein we investigated the ADP-ribosylation signal +/- H₂O₂ and +/- PARGi (see figure below). We observed an increase of signal when combining H₂O₂ with PARGi, which suggests that PARG does actively remove ADPr signal during the 10 minutes of H₂O₂ treatment.

We also treated these lysates with hydroxylamine, and observed no notable difference (also not on shorter exposures), which could suggest that the additional ADPr cannot significantly be attributed to glutamate or aspartate ADPr. However, we believe that hydroxylamine treatment of lysates does not necessarily give as detailed of an insight as MS data, hydroxylamine treatment may partially reverse ADPr on residue types other than glutamate and aspartate (Hsia et al., 1985), and the exact treatment conditions during hydroxylamine experiments may skew the data. Moreover, as alluded to in our reply to Reviewer #1, our study is primarily based on mass spectrometric observations, where we pinpoint the exact amino acid residue modified by ADPr in specific proteins.

Therefore, we are reluctant to draw conclusions based upon western blot experiments in this context, and have decided to only include the samples not treated with hydroxylamine in the revised manuscript (new Figure S1C). This serves to clarify that high eraser enzyme activity may rapidly reverse ADPr in our experimental context, and that this ADPr could include aspartate and glutamate ADPr. We have also further expanded upon this subject in the discussion, and overall we reason that glutamate and aspartate ADPr can co-exist, but overall seems lower abundant compared to serine ADPr, which may stem from acidic acceptor sites being too transient to readily visualize under our experimental conditions.

We concur with the overall notion raised by the Reviewer, and agree that further investigation into glutamate and aspartate ADPr, in the context of PARGi (or perturbation of other eraser enzymes) would be very intriguing. However, we would like to do so using unbiased mass spectrometry, and profiling the dynamic nature of glutamate and aspartate ADPr using MS would likely require adjustment of our experimental setup, which is highly time-consuming and would convolute the current story.

2. The authors' observation of extensive cross-talk between multiple PTMs and ADP-ribose is a particularly intriguing result. I was curious if the system they already have in place might be able to address whether ADP-ribose is necessary and/or sufficient to drive PTM cross-talk. When they generated their GFP-H1.2 S150A mutant, did they attempt to determine whether the mutant form was still phosphorylated on the neighboring threonine? Further, was the 4xSA mutant still alkylated to a similar extent as the wild-type H1.2? Ultimately, these experiments might be beyond the scope of the current work, but their inclusion would bolster the impact of this report even further and would provide an exciting inroad into the potential mechanisms governing ADP-ribose at the histone.

Reply: We fully agree with the Reviewer that crosstalk between ADPr and various other PTMs is highly intriguing and would warrant further investigation. We considered exploration of T146-phosphorylation in the context of the H1.2 S150A mutant, but the only validated antibody that we are aware of (#ab3596 from Abcam) is currently not available. An alternative approach could be to perform additional ADPr proteomics experiments following ectopic expression of either the S150A or 4xSA mutants, but transient expression of these mutants in the context of proteomics is not feasible due to the large scale of the experiment. Moreover, in experiments entailing transient expression of H1.2 the endogenous counterpart would still be expressed, rendering it challenging to decipher by bottom-up MS whether a neighboring phosphorylation would reside on the endogenous or transiently expressed H1.2 isoform. Hence, one would ideally have to establish stable cell lines where endogenous H1.2 is replaced with mutant H1.2, which is highly time-consuming. Another possibility would be to perform phosphorylation/methylation/acetylation enrichment proteomics in the context of ARH3 KO and HPF1 KO cell lines, and investigate the effect on these PTMs when histone ADPr is either highly stimulated or diminished. Still, such experiments would demand a lot of time and complicate the overall story. Overall, we concur with the Reviewer that these experiments are beyond the scope of this manuscript.

The statistical analysis performed herein is well-detailed, valid, and appropriate for the methods employed by the authors. Based on the included data I am convinced that the research presented could be reproduced successfully in an independent laboratory.

Reply: We thank the Reviewer for their kind words.

Reviewer #3 (Remarks to the Author):

The manuscript by Hendriks et al. „The regulatory landscape of the human HPF1- and ARH3-dependent ADP-ribosylome” presents a proteomic analysis of the ADP-ribosylation in human cell lines with the KO of essential PARP1 regulatory proteins (HPF1 and ARH3). The work is mostly descriptive, cataloging the modification sites in these cell lines and comparing the site occupancies between the mutants and the wild type cells. The study is technically sound and provides a resource of interest to the specialized field of ADP-ribosylation. It doesn't go beyond the resource and therefore it may be too preliminary and descriptive, lacking significance matching the standard of Nature Communications. The major suggestion is to expand on the biological significance of the specific sites, especially those regulated by the enzymes depleted in the examined cell lines.

Reply: We are delighted to see that the Reviewer finds our study to be technically sound, and that it presents a significant resource for the ADP-ribosylation field. We concur that our initial submission could be further improved, and following the constructive suggestions by all Reviewers, we trust that we have now substantially improved the quality of our manuscript. We believe that the systems-wide analyses we present here, supported by over half a dozen large-scale proteomics experiments, already provide a wealth of contextual advance.

Although ADP-ribosylation was discovered more than 50 years ago, a detailed understanding on the regulatory aspects of the protein modification remain unexplored for technical reasons, with our current manuscript filling this knowledge gap. As the Reviewer points out, our manuscript is technically sound, provides significant insights for the ADP-ribosylation field, and will undoubtedly be of great interest to the readers of Nature Communications.

Nonetheless, we have now included additional functional experiments, wherein we generated stable cell lines expressing a PARP1 3SA auto-modification mutant (new Figure 6C-E). This mutant displays greatly abolished auto-modification, and moreover resides significantly longer at sites of DNA damage compared to wildtype PARP1, highlighting a biologically relevant *in vivo* function for serine auto-modification of PARP1. Moreover, this accentuates the relevance of our proteomics data, which can be mined for a plethora of ADPr target proteins for functional follow-up.

Overall, we now provide a first example that explains the physiological relevance of site-specific serine ADPr (Figure 6C-E), which was recently independently observed by the Ahel lab using transient overexpression of PARP1 mutants (Prokhorova et al., 2021).

Other comments:

1. The authors state that their finding of mostly serine residues being modified support the notion that their methodology is unbiased (page 5). However, the statement that the serine residues are the major target of ADP-ribosylation, comes from their own work using the same methodology, so this seems to be an unfounded statement. It would be advisable to see the comparison with other purification methodologies using the same cell lines.

Reply: We agree with the Reviewer, and we would like to point out that Reviewer #1 made a similar request. As a result, in the revised manuscript we have included new data that further validates the unbiased nature of our methodology:

- Predominant serine ADPr in cells, via two distinct antibodies (previously Figure 1, now Figure 2)
- Predominant serine ADPr in cells, via boronate affinity purification followed by acid elution (new Figure 3)
- The ability of ETD fragmentation to identify and localization glutamate and aspartate ADPr from *in vitro* samples where such linkages are predominant (new Figure 1)

2. The manuscript seems to be hastily assembled. Some references are missing (for example, page 17, MaxQuant software). It also requires significant copyediting as it is full of errors that are not typos (for example, on page 5, incorrect use of the verb “support” which takes a noun, so the two sentences containing this verb are completely wrong), but there are also numerous typos rendering the manuscript difficult to read.

Reply: We apologize for the missing reference as pointed out by the Reviewer. We have rephrased the sentence pointed out by the Reviewer for clarity, and we have generally improved the clarity of the manuscript text.

REFERENCES

- Bonfiglio, J.J., Fontana, P., Zhang, Q., Colby, T., Gibbs-Seymour, I., Atanassov, I., Bartlett, E., Zaja, R., Ahel, I., and Matic, I. (2017). Serine ADP-Ribosylation Depends on HPF1. *Mol. Cell* **65**, 932-940.
- Bonfiglio, J.J., Leidecker, O., Dauben, H., Longarini, E.J., Colby, T., San Segundo-Acosta, P., Perez, K.A., and Matic, I. (2020). An HPF1/PARP1-Based Chemical Biology Strategy for Exploring ADP-Ribosylation. *Cell* **183**, 1086-1102 e1023.
- Buch-Larsen, S.C., Hendriks, I.A., Lodge, J.M., Rykær, M., Furtwängler, B., Shishkova, E., Westphall, M.S., Coon, J.J., and Nielsen, M.L. (2020). Mapping Physiological ADP-Ribosylation Using Activated Ion Electron Transfer Dissociation. *Cell Rep* **32**, 108176.
- Gagné, J.P., Ethier, C., Defoy, D., Bourassa, S., Langelier, M.F., Riccio, A.A., Pascal, J.M., Moon, K.M., Foster, L.J., Ning, Z., *et al.* (2015). Quantitative site-specific ADP-ribosylation profiling of DNA-dependent PARPs. *DNA Repair (Amst)* **30**, 68-79.
- Geiszler, D.J., Kong, A.T., Avtonomov, D.M., Yu, F., da Veiga Leprevost, F., and Nesvizhskii, A.I. (2020). PTM-Shepherd: analysis and summarization of post-translational and chemical modifications from open search results. *Mol Cell Proteomics*.
- Haag, F., and Buck, F. (2015). Identification and analysis of ADP-ribosylated proteins. *Curr Top Microbiol Immunol* **384**, 33-50.
- Hendriks, I.A., Larsen, S.C., and Nielsen, M.L. (2019). An Advanced Strategy for Comprehensive Profiling of ADP-ribosylation Sites Using Mass Spectrometry-based Proteomics. *Mol Cell Proteomics* **18**, 1010-1026.
- Hendriks, I.A., Lyon, D., Su, D., Skotte, N.H., Daniel, J.A., Jensen, L.J., and Nielsen, M.L. (2018). Site-specific characterization of endogenous SUMOylation across species and organs. *Nat Commun* **9**, 2456.
- Hsia, J.A., Tsai, S.C., Adamik, R., Yost, D.A., Hewlett, E.L., and Moss, J. (1985). Amino acid-specific ADP-ribosylation. Sensitivity to hydroxylamine of [cysteine(ADP-ribose)]protein and [arginine(ADP-ribose)]protein linkages. *J Biol Chem* **260**, 16187-16191.
- Larsen, S.C., Hendriks, I.A., Lyon, D., Jensen, L.J., and Nielsen, M.L. (2018). Systems-wide Analysis of Serine ADP-Ribosylation Reveals Widespread Occurrence and Site-Specific Overlap with Phosphorylation. *Cell Rep* **24**, 2493-2505 e2494.
- Mikesh, L.M., Ueberheide, B., Chi, A., Coon, J.J., Syka, J.E.P., Shabanowitz, J., and Hunt, D.F. (2006). The utility of ETD mass spectrometry in proteomic analysis. *Biochimica et biophysica acta* **1764**, 1811-1822.
- Moss, J., Yost, D.A., and Stanley, S.J. (1983). Amino acid-specific ADP-ribosylation. *J. Biol. Chem* **258**, 6466-6470.

Palazzo, L., Leidecker, O., Prokhorova, E., Dauben, H., Matic, I., and Ahel, I. (2018). Serine is the major residue for ADP-ribosylation upon DNA damage. *Elife* 7.

Pattabiraman, V.R., and Bode, J.W. (2011). Rethinking amide bond synthesis. *Nature* 480, 471-479.

Prokhorova, E., Zobel, F., Smith, R., Zentout, S., Gibbs-Seymour, I., Schutzenhofer, K., Peters, A., Gros Lambert, J., Zorzini, V., Agnew, T., *et al.* (2021). Serine-linked PARP1 auto-modification controls PARP inhibitor response. *Nat Commun* 12, 4055.

Rodriguez, K.M., Buch-Larsen, S.C., Kirby, I.T., Siordia, I.R., Hutin, D., Rasmussen, M., Grant, D.M., David, L.L., Matthews, J., Nielsen, M.L., *et al.* (2021). Chemical genetics and proteome-wide site mapping reveal cysteine MARYlation by PARP-7 on immune-relevant protein targets. *Elife* 10.

Wolhuter, K., Whitwell, H.J., Switzer, C.H., Burgoyne, J.R., Timms, J.F., and Eaton, P. (2018). Evidence against Stable Protein S-Nitrosylation as a Widespread Mechanism of Post-translational Regulation. *Molecular Cell* 69, 438-450.e435.

Zhang, Y., Wang, J., Ding, M., and Yu, Y. (2013). Site-specific characterization of the Asp- and Glu-ADP-ribosylated proteome. *Nat. Methods* 10, 981-984.

Zhen, Y., Zhang, Y., and Yu, Y. (2017). A Cell-Line-Specific Atlas of PARP-Mediated Protein Asp/Glu-ADP-Ribosylation in Breast Cancer. *Cell Rep* 21, 2326-2337.

REVIEWERS' COMMENTS

Reviewer #1 (Remarks to the Author):

In the study presented by Hendriks and colleagues, there are basically two fundamental points that needed to be clarified in order to have a valid interpretation of the cellular ADP-ribosylome. First, the MacroAF1521-based affinity-purification of ADP-ribosylated peptides following PARG treatment must be unbiased with respect to the type of ADP-ribose linkages on the amino acid target. Second, the mass spectrometry must also be unbiased to be able to assign site-specific ADP-ribosylation modifications to the ADP-ribosylated peptides isolated with the MacroAF1521 approach. The authors provided evidence which demonstrates that indeed, the MacroAF1521 approach appears to be universally applicable to the isolation of ADP-ribosylated peptides regardless of their type of chemical bond to ADP-ribose. The problem still lies with the second premise. Contrary to what is mentioned in the response to the reviewers letter, there is a very significant bias for unusual tryptic peptides listed with site-specific ADP-ribose modifications. The authors performed statistical analysis on the entire dataset of peptides, including the unmodified peptides, to conclude that K/R missed cleavages are in the range of what would be expected based on the occurrence of K and R residues in the human proteome. What I want to emphasize is that the number of missed cleavages is absolutely not common for the vast majority of ADP-ribosylated peptides analyzed with their ETD-based peptide fragmentation approach. Lets take a look at all the ADP-ribosylated PARP-1 peptides listed in Supplementary Table-2:

AEPVEVVAPR(ADPr)GK

AEPVEVVAPRGKS(ADPr)GAALSK

AEPVEVVAPRGKS(ADPr)GAALSCK

AEPVEVVAPRGKS(ADPr)GAALSKKSK

AEPVEVVAPRGKS(ADPr)GAALSKKSKGQVK

AEPVEVVAPRGKS(ADPr)GAALSKKSKGQVK

AEPVEVVAPRGKS(ADPr)GAALSKKSKGQVK

AEPVEVVAPRGKS(ADPr)GAALSKKSKGQVK

AEPVEVVAPRGKS(ADPr)GAALSKKSKGQVK

AEPVEVVAPRGKS(ADPr)GAALSKKSKGQVK

AEPVEVVAPRGKS(ADPr)GAALSKKSKGQVK

AEPVEVVAPRGKS(ADPr)GAALSKKSKGQVK

AEPVEVVAPRGKS(ADPr)GAALSKKSKGQVK

AEPVEVVAPRGKS(ADPr)GAALSKKSKGQVK

AEPVEVVAPRGKSGAALS(ADPr)K

AEPVEVVAPRGKSGAALS(ADPr)KK

AEPVEVVAPRGKSGAALSKKS(ADPr)KGQVK

EEGINKS(ADPr)EK

EEGINKS(ADPr)EKR

EEGINKS(ADPr)EKRM(ox)K

EEGINKS(ADPr)EKRM(ox)KLTLK

EEGINKS(ADPr)EKRMK

EEGINKS(ADPr)EKRMKLTLK

EFREIS(ADPr)YLK

EFREIS(ADPr)YLKK

GGAAVDPDSGLEH(ADPr)SAHVLEK

GKS(ADPr)GAALSK

GKS(ADPr)GAALSKK

GKS(ADPr)GAALSKK

GKS(ADPr)GAALSKKSK

GKSGAALS(ADPr)K

GKSGAALS(ADPr)KK

GKSGAALSKKS(ADPr)K

GKSGAALSKKS(ADPr)KGQVK

GKSGAALSKKS(ADPr)KGQVKEEGINK

GQDGIGS(ADPr)KAEK

GQVKEEGINKS(ADPr)EK

GQVKEEGINKS(ADPr)EK

GQVKEEGINKS(ADPr)EKR

GQVKEEGINKS(ADPr)EKRM(ox)K

GQVKEEGINKS(ADPr)EKRM(ox)KLTLK
GQVKEEGINKS(ADPr)EKRMK
GQVKEEGINKS(ADPr)EKRMKLTLK
GQVRLS(ADPr)K
GQVRLS(ADPr)KK
HASHIS(ADPr)KLPK
KC(ca)SES(ADPr)IPKDSLRL
KQLPGVKS(ADPr)EGK
KQLPGVKS(ADPr)EGKR
KS(ADPr)KGQVKEEGINK
KS(ADPr)KGQVKEEGINKSEK
KSK(ADPr)GQVKEEGINKSEK
KSKGQVKEE(ADPr)GINKSEK
KSKGQVKEEGINKS(ADPr)EK
KSKGQVKEEGINKS(ADPr)EKR
KTAEAGGVTGKGQDGIGS(ADPr)KAEK
M(ox)PLGKLS(ADPr)KR
MPLGKLS(ADPr)K
MPLGKLS(ADPr)KR
NFTKY(ADPr)PK
QLPGVKS(ADPr)EGK
QLPGVKS(ADPr)EGKR
QLPGVKS(ADPr)EGKRKGDEVDGVDEVAK
RKGDEVDGVDEVAKKKS(ADPr)KK
S(ADPr)GAALSK
S(ADPr)GAALSKK
S(ADPr)GAALSKK

S(ADPr)GAALSKK
S(ADPr)GAALSKK
S(ADPr)GAALSKK
S(ADPr)GAALSKK
S(ADPr)GAALSKK
S(ADPr)GAALSKK
S(ADPr)GAALSKK
S(ADPr)GAALSKK
S(ADPr)GAALSKK
S(ADPr)GAALSKK
S(ADPr)GAALSKK
S(ADPr)GAALSKK
S(ADPr)GAALSKK
S(ADPr)GAALSKK
S(ADPr)GAALSKK
S(ADPr)GAALSKK
S(ADPr)GAALSKK
S(ADPr)GAALSKK
S(ADPr)GAALSKK
S(ADPr)GAALSKK
S(ADPr)GAALSKK
S(ADPr)GAALSKK
S(ADPr)GAALSKK
S(ADPr)GAALSKK
S(ADPr)GAALSKK
S(ADPr)GAALSKK
S(ADPr)GAALSKKSKGQVK
S(ADPr)GAALSKKSKGQVKEEGINK
S(ADPr)KGQVKEEGINK
S(ADPr)KGQVKEEGINKSEK
S(ADPr)KGQVKEEGINKSEKR
S(ADPr)KKEKDKDSKLEK

SGAALS(ADPr)KK
SGAALSKKS(ADPr)K
SGAALSKKS(ADPr)KGQVK
SGAALSKKS(ADPr)KGQVKEEGINK
SGAALSKKS(ADPr)KGQVKEEGINKSEK
SK(ADPr)GQVKEEGINK
SK(ADPr)GQVKEEGINKSEK
SKGQVK(ADPr)EEGINKSEK
SKGQVKEE(ADPr)GINKSEK
SKGQVKEEGINK(ADPr)SEK
SKGQVKEEGINKS(ADPr)EK
SKGQVKEEGINKS(ADPr)EKR
TAEAGGVTGKGQDGIGS(ADPr)K
TAEAGGVTGKGQDGIGS(ADPr)KA EK

Among these 108 PARP-1 peptides, only 2 ADP-ribosylation sites were assigned from fully tryptic peptides. In other words, 98% of all ADP-ribosylation sites are identified on peptides with at least one missed cleavage. Peptides with three, four or even five missed cleavages are very common in the ADP-ribosylome dataset. This pattern is not specific to PARP-1. The vast majority of ADP-ribosylated protein substrates identified in this study follow the same trend for large and unusually positively charged peptides. The authors provided the following explanation in the response to reviewers letter : « We reason that the presence of an intact ADPr moiety incurs a negative charge upon the peptide, which must be compensated for by another charge-bearing residue type (such as lysine or arginine) in order for the peptide to be resolvable via MS/MS analysis ». Based on their own explanation, it is understood that their method is particularly unsuitable for identifying negatively charged peptides such as those containing D/E residues. The authors strongly defend their approach by demonstrating that the ETD approach can identify ADP-ribosylation on D/E residues (new Fig-1). D/E-ADP-ribosylation is so abundant when PARP-1 is automodified in vitro in the absence of HPF1 that its identification with ETD is not surprising even if ETD appears to be sub-optimal for this type of linkage. The final datasets based on complex peptides mixtures shows that D/E ADP-ribosylation is under-represented, likely by the intrinsic nature of the ETD approach. Thus, to rule that PARylation is physiologically predominant over serines appears biased. Even though there is a technological bias in favor of S-ADP-ribosylation, the importance of HPF1 in the catalytic mechanism of PARP-1 is well demonstrated and it is probably a predominant target. However, the authors underestimate the possible contribution of D/E-ADP-ribosylation and instead attempt to present it as an artifact. In the response to reviewers letter, the authors treated automodified PARP-1 with hydroxylamine and showed its extreme sensitivity to hydroxylamine hydrolysis. I agree that artefacts induced by

hydroxylamine treatment are likely to occur but decades of research clearly established the targeting of carboxylic esters by hydroxylamine. Based on hydroxylamine hydrolysis, Yu and colleagues reported more than 400 MS/MS spectra with ADP-ribosylation signatures on endogenous PARP-1. Some of the sites (e.g. E491) are assigned with more than 100 peptide spectrum matches, outnumbering the total amount of PARP-1 spectra for the entire dataset listed here. It would be much more prudent and fair to speak of complementarity in MS/MS approaches (with their strengths and limitations) rather than to attempt to demonstrate that alternative approaches are artifacts. Under these conditions, it is difficult to challenge the current dogma concerning the targets of PARP-1 activity as stipulated by the authors. The study presented here is of excellent quality and the comments of the co-authors following the points raised by the reviewers answer most of the questions. An obvious effort has been made to improve the study, but the authors would benefit to temper their assertions and leave a door open to the diversity and complexity of PARylation.

Notes :

Fig. 1-D : Diagnostic ions corresponding to ADP-ribose fragmentation (i.e. 136.06, 250.09, 348.07, and 428.03) are not annotated on the spectra. Are they there? If not, why?

Fig. 6-E. Data points and error bars are missing.

Reviewer #2 (Remarks to the Author):

I think that the authors have responded to the reviewer comments appropriately and the manuscript is ready for publication.

Point-by-point response to Reviewer comments

Reviewer #1 (Remarks to the Author):

In the study presented by Hendriks and colleagues, there are basically two fundamental points that needed to be clarified in order to have a valid interpretation of the cellular ADP-ribosylome. First, the MacroAF1521-based affinity-purification of ADP-ribosylated peptides following PARG treatment must be unbiased with respect to the type of ADP-ribose linkages on the amino acid target. Second, the mass spectrometry must also be unbiased to be able to assign site-specific ADP-ribosylation modifications to the ADP-ribosylated peptides isolated with the MacroAF1521 approach. The authors provided evidence which demonstrates that indeed, the MacroAF1521 approach appears to be universally applicable to the isolation of ADP-ribosylated peptides regardless of their type of chemical bond to ADP-ribose.

Reply: We would like to thank the Reviewer for the time and effort they have put towards reviewing our manuscript, and for the constructive criticism provided on our original and revised manuscript. We agree that the entire sample-processing pipeline for ADPr proteomics experiments, including ADPr enrichment and mass spectrometric analysis of the final samples, should occur in a controlled and unbiased manner. This is exactly what we have aimed to demonstrate with the current and previous manuscripts centered on our Af1521 enrichment strategy, and we are delighted that the Reviewer finds the Af1521 methodology to be universally applicable, and able to detect ADPr on every residue type.

The problem still lies with the second premise. Contrary to what is mentioned in the response to the reviewers letter, there is a very significant bias for unusual tryptic peptides listed with site-specific ADP-ribose modifications. The authors performed statistical analysis on the entire dataset of peptides, including the unmodified peptides, to conclude that K/R missed cleavages are in the range of what would be expected based on the occurrence of K and R residues in the human proteome.

Reply: We would like to clarify that our approximation based on occurrence of K/R within the human proteome, only relates to the expected amino acid at the C-terminus of the peptides, which indeed follows expectations in our data (i.e. having 2/3rd of peptides end with K and 1/3rd of peptides end with R). Notably, this approximation did not apply to the overall number of observed missed cleavages.

What I want to emphasize is that the number of missed cleavages is absolutely not common for the vast majority of ADP-ribosylated peptides analyzed with their ETD-based peptide fragmentation approach. Lets take a look at all the ADP-ribosylated PARP-1 peptides listed in Supplementary Table-2:

AEPVEVVAPR(ADPr)GK
AEPVEVVAPRGKS(ADPr)GAALSK
AEPVEVVAPRGKS(ADPr)GAALSKK
AEPVEVVAPRGKS(ADPr)GAALSKKSK
AEPVEVVAPRGKS(ADPr)GAALSKKSKGQVK
AEPVEVVAPRGKS(ADPr)GAALSKKSKGQVK
AEPVEVVAPRGKS(ADPr)GAALSKKSKGQVK
AEPVEVVAPRGKS(ADPr)GAALSKKSKGQVK
AEPVEVVAPRGKS(ADPr)GAALSKKSKGQVK
AEPVEVVAPRGKS(ADPr)GAALSKKSKGQVK
AEPVEVVAPRGKS(ADPr)GAALSKKSKGQVK
AEPVEVVAPRGKS(ADPr)GAALSKKSKGQVK
AEPVEVVAPRGKS(ADPr)GAALSKKSKGQVK
AEPVEVVAPRGKSGAALS(ADPr)K
AEPVEVVAPRGKSGAALS(ADPr)KK
AEPVEVVAPRGKSGAALSKKS(ADPr)KGQVK
EEGINKS(ADPr)EK
EEGINKS(ADPr)EKR
EEGINKS(ADPr)EKRM(ox)K
EEGINKS(ADPr)EKRM(ox)KLTLK
EEGINKS(ADPr)EKRMK
EEGINKS(ADPr)EKRMKLTLK
EFREIS(ADPr)YLK
EFREIS(ADPr)YLKK
GGAAVDPDSGLEH(ADPr)SAHVLEK
GKS(ADPr)GAALSK
GKS(ADPr)GAALSKK
GKS(ADPr)GAALSKK
GKS(ADPr)GAALSKKSK
GKSGAALS(ADPr)K
GKSGAALS(ADPr)KK
GKSGAALSKKS(ADPr)K
GKSGAALSKKS(ADPr)KGQVK
GKSGAALSKKS(ADPr)KGQVKEEGINK
GQDGIGS(ADPr)KA EK
GQVKEEGINKS(ADPr)EK
GQVKEEGINKS(ADPr)EK
GQVKEEGINKS(ADPr)EKR
GQVKEEGINKS(ADPr)EKRM(ox)K
GQVKEEGINKS(ADPr)EKRM(ox)KLTLK
GQVKEEGINKS(ADPr)EKRMK
GQVKEEGINKS(ADPr)EKRMKLTLK
GQVRLS(ADPr)K
GQVRLS(ADPr)KK
HASHIS(ADPr)KLPK
KC(ca)SES(ADPr)IPKDSLRL
KQLPGVKS(ADPr)EGK
KQLPGVKS(ADPr)EGKR
KS(ADPr)KGQVKEEGINK
KS(ADPr)KGQVKEEGINKSEK
KSK(ADPr)GQVKEEGINKSEK
KSKGQVKEE(ADPr)GINKSEK
KSKGQVKEEGINKS(ADPr)EK
KSKGQVKEEGINKS(ADPr)EKR
KTAEAGGVTKGQDGIGS(ADPr)KA EK
M(ox)PLGKLS(ADPr)KR

Reply: We would like to thank the reviewer for bringing this observation to our attention. Following this, we investigated the specific examples raised by the Reviewer, and we were unfortunately not able to reproduce the exact same list of 108 PARP1 peptides from our provided Table S2. We would also like to point out that several of the mentioned entries, such as _AEPVEVVAPRGKS(ADPr)GAALSKKSKGQVK_ and _S(ADPr)GAALSKK_, seem to be duplicated many times in the above list.

Nonetheless, we noticed that among the 108 peptides highlighted by the reviewer, the majority of these actually harbor a glutamic acid, and several sequences harbor multiple glutamic acid residues. In fact, from the 1,520 amino acids entailed in the 108 peptides, 151 correspond to glutamic acid residues, corresponding to 10% of all detected amino acids in these sequences, which is larger than what is expected in the human proteome (~6% of total amino acids are glutamic acids). Hence, these selected peptides further that our analytical strategy is not biased against detection of glutamic acid residues, even in peptides harboring missed cleavages.

With regard to the missed cleavages, we would like to bring to the Reviewer's attention that while we in the supplemental material provide all modified peptide sequences we detected for each individual ADPr site, there is no direct quantitative information for the individual contribution of each of these peptides to the overall observed abundance.

Therefore, to make such quantitative comparison available, we analyzed all the raw evidences that went into construction of Table S2, with regard to PARP1 peptides. Overall, we identified 1,876 ADPr-modified PARP1 peptides via MS/MS (in the experiment corresponding to Table S2), which entailed on average 2.51 missed cleavages:

However, the intensity contribution from the various peptide sequences will vary. When weighing the observed peptides by their actual abundance in the samples, the average number of missed cleavages adjusts down to 1.97, because peptides with 1 and 2 missed cleavages are far more abundant than those with 3+ missed cleavages.

Whereas we agree with the Reviewer that 1.97 (or 2.51) missed cleavages is on the high side, this observation falls in line with what we and others have observed in previous studies, and in the previous “Point-by-point response to Reviewer comments” we provided various explanations as to why this may occur. For example, analysis of ADPr-enriched samples resulted in highly frequent detection of peptides with one or more missed cleavages, when using either HCD or ETD, and for both fragmentation methods the number of missed cleavages observed was higher in ADPr-modified peptides compared to unmodified peptides (Larsen et al., 2018).

We would also like to point out, that the same phenomenon has been observed for phosphopeptides, which also contain a higher number of missed cleavages compared to unmodified peptides (Molina et al., 2007). Here, the authors outline that the increase in missed cleavages primarily is due to structural interference of phosphorylation with trypsin, and the same may indeed happen with ADP-ribosylation, which after all is a much larger PTM than phosphorylation. Whether ADP-ribosylation to some extent interferes with trypsin will require detailed investigations, which lies outside the scope of this story. Still, the observations made with phosphorylated peptides align with the notions raised

by the Reviewer, and further that the slight increase in missed cleavages does not relate to any analytical biases against detection of glutamic or aspartic acid residues.

We would also like to mention, that while ETD fragmentation did result in a higher number of observed missed cleavages, this trend applied to all peptides in the sample – including unmodified peptides. This follows what is well known in the literature; that ETD analysis of $z=2$ precursors is less efficient, and that ETD thrives on analysis of highly protonated precursors (Liu and McLuckey, 2012). Thus, additional lysine and arginine residues in the peptide sequence will allow more protonation to happen during the electrospray process, and consequently render these peptides more amenable for detection via ETD. Along these lines, we would also like to clarify that in proteomics experiments (commonly performed at low pH) there is a distinction between the protonation required for MS analysis (and peptide fragmentation), and the term related to ‘charged’ amino acids. In proteomics, basic amino acids (primarily lysine and arginine, along with the peptide N-terminus) constitute the most frequently targeted residues for protonation during the electrospray process. This renders the analyzed peptides ‘charged’ via the protonation, and not based upon their content of glutamic or aspartic acids in the peptide sequence.

Collectively, our obtained data follows the fragmentation propensity related to ETD and what is reported in other PTM-based proteomics studies. While ETD fragmentation is more efficient on multiply charged (protonated) peptides, this does not affect the ability of our analytical strategy to identify ADPr acceptor sites in an unbiased manner.

The authors provided the following explanation in the response to reviewers letter : « We reason that the presence of an intact ADPr moiety incurs a negative charge upon the peptide, which must be compensated for by another charge-bearing residue type (such as lysine or arginine) in order for the peptide to be resolvable via MS/MS analysis ». Based on their own explanation, it is understood that their method is particularly unsuitable for identifying negatively charged peptides such as those containing D/E residues.

Reply: We would like to apologize for the confusing explanation we offered previously, as outlined by the Reviewer. To clarify this: We have previously observed that the ADPr moiety is able to prevent the peptide from fragmentation after absorbing one electron via the ETD reaction (Buch-Larsen et al., 2020). This phenomenon is referred to as “ETnoD” (Ledvina et al., 2010), and appears to be much more prevalent for ADPr-modified peptides as compared to unmodified peptides. In our previous works, we investigated the ability of both EThcD (Hendriks et al., 2019) and AI-ETD (Buch-

Larsen et al., 2020) to bypass this property of the ADPr modification on peptides, and although both methods improve fragmentation of ADPr-modified peptides via ETD, the dissociation often remains incomplete. However, this phenomenon remains a general property of ADP- and other PTM-modified peptides and is not dependent, or biased against, certain acceptor sites.

Importantly, because the ADPr moiety appears able to stabilize the peptide after absorbing one electron during the ETD process, we have observed that the peptide needs to absorb two electrons before it will readily dissociate. This is due to the non-ergodic fragmentation propensity of the ETD dissociation technique. Hence, if the precursor would be $z=2$, then transfer of two electrons during the ETD process would neutralize the two electrospray-generated protons and consequently result in a net $z=0$ precursor, which thus becomes 'invisible' to any mass spectrometric detector. If the precursor would be $z=3$ (or higher), then neutralization of two protons by two electrons would allow one positive charge (i.e. proton) to remain on the peptide fragments, thereby making them detectable via the Orbitrap and allowing us to sequence the peptide backbone. This, in addition to what we described above (and in our previous point-by-point response), is why ETD more efficiently detects higher-charged precursors, which in turn can be expected to harbor a larger number of missed cleavages. As mentioned above, this phenomenon is not unique to ADP-ribosylated peptides but also observed for e.g. phosphorylated peptides, and this phenomenon does not affect detection of peptides harboring acidic residues – as supported by the peptide list selected by the Reviewer.

Beyond this, we would also like to clarify that the chromatography we perform in front of the mass spectrometer is performed at low pH, using 0.1% formic acid. At this pH, glutamate and aspartate residues will remain neutral and not be negatively charged.

The authors strongly defend their approach by demonstrating that the ETD approach can identify ADP-ribosylation on D/E residues (new Fig-1). D/E-ADP-ribosylation is so abundant when PARP-1 is automodified *in vitro* in the absence of HPF1 that its identification with ETD is not surprising even if ETD appears to be sub-optimal for this type of linkage.

Reply: We are delighted that the Reviewer appreciates our demonstration of ETD being able to detect the intact ADPr moiety on glutamate residues in PARP1.

With regard to the relatively low abundance of glutamate ADPr we observed *in vitro* in the absence of HPF1, we reasoned that this is likely due to the fact that we need PARG to reduce the very long *in vitro* ADPr polymers to monomers in order to facilitate their detection (Figure 1C). Our analysis

measures the site-specific abundance of glutamic mono-ADPr compared to alternative methods. For example, immunoblot (and similar) analyses usually visualize each ADPr moiety entailed in a poly-ADPr chain, and thus generate a far stronger signal for long chains. Since our MS method only generates signal for the amino-acid bound ADPr moiety (i.e. the mono-ADPr attached to the acceptor site), our MS analysis provides details on the actual stoichiometry of individual ADPr-modified residues, whereas immunoblot analyses may overestimate this in cases where modified residues harbor a poly-chain (as the latter visualizes each individual moiety in the chain). Hence, MS analyses generally provide a better visualization of the amount of protein modified with ADPr at individual acceptor sites, whereas the measured signal from immunoblotting may be biased towards visualizing longer chains.

The final datasets based on complex peptides mixtures shows that D/E ADP-ribosylation is under-represented, likely by the intrinsic nature of the ETD approach. Thus, to rule that PARylation is physiologically predominant over serines appears biased. Even though there is a technological bias in favor of S-ADP-ribosylation, the importance of HPF1 in the catalytic mechanism of PARP-1 is well demonstrated and it is probably a predominant target.

Reply: We respectfully disagree with the Reviewer on the notion that ETD is unsuitable for detection of E/D residues. As stated by the Reviewer (*vide supra*): “*Based on their own explanation, it is understood that their method is particularly unsuitable for identifying negatively charged peptides such as those containing D/E residues.*” As noted above, our initial explanation regarding ADPr adding a ‘negative charge’ to the peptide was insufficiently detailed. Rather, the ADPr moiety appears to stabilize the peptide from electron-mediated dissociation, and requires two electrons in order to dissociate, necessitating a base charge of $z=3$ (or higher).

There is, to the best of our knowledge, no published scientific evidence indicating that ETD-type fragmentation is unable to resolve D/E residues. In fact, we would like to point out that the peptide sequences highlighted by the reviewer do contain a large fraction of glutamic acid residues, which confirms that our analytical strategy is not biased.

Nonetheless, we performed an additional investigation of HCD vs. ETD vs. EThcD fragmentation, using HeLa cell lysates. Three replicate analyses were performed using each fragmentation method, and we investigated the amino acid content of the peptides directly MS/MS-identified via each of the fragmentation techniques.

The graph above displays the amino acid content of the peptides identified in these runs, and then specifically for the ADPr acceptor amino acids we investigated as part of our main study. The box bounds represent 1st and 3rd quantile, the upper whisker 95th percentile, and the plus symbol represents the average value. The number of peptides observed is listed below, and was 100k for HCD, >40k for ETD, and >60k for EThcD.

Overall, we observed a slightly higher prevalence for positively charged residues (H,K,R) for pure ETD fragmentation, which makes sense as these amino acids are the primary targets for protonation during the electrospray, and a higher charge state ($z=3+$) is often required for resolving peptides via ETD. The number of D/E observed was only slightly lower for ETD. Importantly, serine residues were also slightly underrepresented via ETD fragmentation. Moreover, these subtle differences were alleviated when performing EThcD fragmentation, which is the fragmentation mode used for all ETD-type experiments entailed in this study.

The graph above is a more simplified overview, and also visualizes the extremely low variance (CV=0.3%) between the three runs we performed. It corroborates that K/R residues are slightly overrepresented by ETD, and E/D and S residues are slightly underrepresented. The difference, though, is minor (less than 10% relatively) and largely overcome via application of EThcD.

Finally, we would like to re-iterate that these types of systemic observations (amino acid distribution, charge state, etc.) hold true for HCD vs. ETD across all of the sample types we have investigated in recent years. Hence, we conclude that what is truly variable between our studies is the content of the samples, and that ETD is not biased in favor of serine residues or against D/E residues.

However, the authors underestimate the possible contribution of D/E-ADP-ribosylation and instead attempt to present it as an artifact. In the response to reviewers letter, the authors treated automodified PARP-1 with hydroxylamine and showed its extreme sensitivity to hydroxylamine hydrolysis. I agree that artefacts induced by hydroxylamine treatment are likely to occur but decades of research clearly established the targeting of carboxylic esters by hydroxylamine.

Reply: We are happy that the Reviewer agrees that hydroxylamine treatment can introduce artificial chemical marks, which may be falsely interpreted as true ADPr sites. We do not exclude that glutamate/aspartate ADPr exists, and we previously already toned down underestimation of D/E ADP-ribosylation in the manuscript, and now added additional discussion to explain our observations in detail. We have now further toned down some of the statements relating to D/E ADPr contribution; in the abstract, introduction, and discussion.

We would also like to mention that the caution related to hydroxylamine primarily relates to its use in proteomics experiments, as mass spectrometric analysis cannot discriminate between artefacts introduced by hydroxylamine and true ADPr-modified acidic residues targeted by hydroxylamine. However, we are aware that hydroxylamine is successfully used in biochemical methods (i.e. immunoblot analysis) to reverse ADPr ester bonds. Hence, our observations related to chemical artifacts induced by hydroxylamine only reflect its usage in proteomics experiments, and we have emphasized this in the latest iteration of our manuscript.

Based on hydroxylamine hydrolysis, Yu and colleagues reported more than 400 MS/MS spectra with ADP-ribosylation signatures on endogenous PARP-1. Some of the sites (e.g. E491) are assigned with more than 100 peptide spectrum matches, outnumbering the total amount of PARP-1 spectra for the entire dataset listed here.

Reply: We respectfully disagree with the observations made by the Reviewer in this case.

For the in vitro experiments we performed here, we operated our instrument in a manner that prevents it from sequencing the same precursor repeatedly. This facilitates a more in-depth sequencing of the sample. As a result, indeed, often we will make exactly one PSM per unique peptide, per MS run.

We presume the Reviewer is referring to the MS experiments performed by Yu and colleagues (Zhang et al., 2013; Zhen et al., 2017). These experiments were performed on an older type of mass spectrometer (LTQ Orbitrap Velos), which has the technological limitation of a very short precursor exclusion list. This in practice means that the instrument can only exclude few precursors from being sequenced repeatedly, and thus overall tends to re-identify the same precursor over and over.

Furthering this, our re-analyses of the data from Zhen et al. indicated a repeated sequencing of isotopic patterns of >30%, including some of the auto-modified PARP1 peptides being sequenced repeatedly (sometimes up to 10 times in a single MS run). On the contrary, our runs exhibit <10% repeated sequencing, and most of the modified PARP1 peptides were sequenced exactly one time per MS run. Nonetheless, we observed 47 modified PARP1 PSMs over 6 of our in vitro runs, whereas 115 modified PARP1 PSMs were observed in 9 MS runs (based on breast cancer cells) by Zhen et al.

Moreover, we would like to point out that we observed 1,880 modified PARP1 PSMs in the MS data associated with Figure 2 (antibody comparison), and 1,189 modified PARP1 PSMs in the MS data associated with Figure 4 (HPF1/ARH3 KO).

Finally, we would like to emphasize that some of the sites reported by Yu et al may certainly constitute true ADPr acceptor sites. The challenge, however, is to discriminate chemical artifacts from true ADPr acceptor sites when using hydroxylamine in MS-based proteomics analyses.

We have further clarified this in the discussion of the revised manuscript.

It would be much more prudent and fair to speak of complementarity in MS/MS approaches (with their strengths and limitations) rather than to attempt to demonstrate that alternative approaches are artifacts. Under these conditions, it is difficult to challenge the current dogma concerning the targets of PARP-1 activity as stipulated by the authors.

The study presented here is of excellent quality and the comments of the co-authors following the points raised by the reviewers answer most of the questions. An obvious effort has been made to improve the study, but the authors would benefit to temper their assertions and leave a door open to the diversity and complexity of PARylation.

Reply: We are grateful that the Reviewer finds our study of excellent quality, and appreciates the effort we made towards improving the manuscript. We agree with the Reviewer that the different MS/MS approaches offer complementarity. However, we still believe that we should accurately document our observations regarding the distribution of ADPr-modified amino acid types (and the relative absence of D/E ADPr within our dataset), and that we should caution the field against potential artefacts that may occur through usage of hydroxylamine in proteomics experiments.

We have tempered our harshest conclusions, and we fully agree that contemporary MS methodology may not yet encompass the full diversity and complexity of ADP-ribosylation.

Notes :

Fig. 1-D : Diagnostic ions corresponding to ADP-ribose fragmentation (i.e. 136.06, 250.09, 348.07, and 428.03) are not annotated on the spectra. Are they there? If not, why?

Reply: We do not observe ADPr diagnostic fragments because we do not use collisional-type dissociation (HCD or CID). These types of high-energy fragmentation cause the ADPr moiety to

shatter, and forcefully remove ADPr from the target peptide. These ADPr fragments take positive charge with them and are thus visible as diagnostic fragments. This fragmentation and removal of ADPr is why HCD or CID cannot accurately localize ADPr, which we have previously detailed (Larsen et al., 2018).

As we use ETD-type fragmentation, which leaves the ADPr moiety intact and attached to the peptide, no ADPr fragments (i.e. diagnostic ions) are observed.

Fig. 6-E. Data points and error bars are missing.

Reply: We apologize for the missing information, and have now provided all data points as part of the Source Data included with the manuscript. Error bars are not applicable in this figure because we display a summed total of observations. Significance was determined via Chi-squared testing.

Once again, we thank the reviewer for the kind comments and constructive suggestions. We really appreciate the reviewer's great effort in reviewing our manuscript and improving the quality of our work.

Reviewer #2 (Remarks to the Author):

I think that the authors have responded to the reviewer comments appropriately and the manuscript is ready for publication.

Reply: We are delighted that the Reviewer finds our manuscript suitable for publication.

REFERENCES

Buch-Larsen, S.C., Hendriks, I.A., Lodge, J.M., Rykær, M., Furtwängler, B., Shishkova, E., Westphall, M.S., Coon, J.J., and Nielsen, M.L. (2020). Mapping Physiological ADP-Ribosylation Using Activated Ion Electron Transfer Dissociation. *Cell Rep* 32, 108176.

Hendriks, I.A., Larsen, S.C., and Nielsen, M.L. (2019). An Advanced Strategy for Comprehensive Profiling of ADP-ribosylation Sites Using Mass Spectrometry-based Proteomics. *Mol Cell Proteomics* 18, 1010-1026.

Larsen, S.C., Hendriks, I.A., Lyon, D., Jensen, L.J., and Nielsen, M.L. (2018). Systems-wide Analysis of Serine ADP-Ribosylation Reveals Widespread Occurrence and Site-Specific Overlap with Phosphorylation. *Cell Rep* 24, 2493-2505 e2494.

Ledvina, A.R., Beauchene, N.A., McAlister, G.C., Syka, J.E., Schwartz, J.C., Griep-Raming, J., Westphall, M.S., and Coon, J.J. (2010). Activated-ion electron transfer dissociation improves the ability of electron transfer dissociation to identify peptides in a complex mixture. *Anal Chem* 82, 10068-10074.

Liu, J., and McLuckey, S.A. (2012). Electron Transfer Dissociation: Effects of Cation Charge State on Product Partitioning in Ion/Ion Electron Transfer to Multiply Protonated Polypeptides. *Int J Mass Spectrom* 330-332, 174-181.

Molina, H., Horn, D.M., Tang, N., Mathivanan, S., and Pandey, A. (2007). Global proteomic profiling of phosphopeptides using electron transfer dissociation tandem mass spectrometry. *Proc Natl Acad Sci U S A* 104, 2199-2204.

Zhang, Y., Wang, J., Ding, M., and Yu, Y. (2013). Site-specific characterization of the Asp- and Glu-ADP-ribosylated proteome. *Nat. Methods* 10, 981-984.

Zhen, Y., Zhang, Y., and Yu, Y. (2017). A Cell-Line-Specific Atlas of PARP-Mediated Protein Asp/Glu-ADP-Ribosylation in Breast Cancer. *Cell Rep* 21, 2326-2337.